# ABIN1 is a negative regulator of effector functions in cytotoxic T cells

Sarka Janusova [1,2], Darina Paprckova [1], Juraj Michalik [1], Valeria Uleri [1,2], Ales Drobek[1], Eva Salyova [1], Louise Chorfi[1], Ales Neuwirth [1], Arina Andreyeva [1], Jan Prochazka [3], Radislav Sedlacek[3], Peter Draber [1,4] & Ondrej Stepanek[1✉]

## Abstract

**T cells are pivotal in the adaptive immune defense, necessitating a delicate balance between robust response against infections and self-tolerance. Their activation involves intricate cross-talk among signaling pathways triggered by the T-cell antigen receptors (TCR) and co-stimulatory or inhibitory receptors. The molecular regulation of these complex signaling networks is still incompletely understood. Here, we identify the adaptor protein ABIN1 as a component of the signaling complexes of GITR and OX40 co-stimulation receptors. T cells lacking ABIN1 are hyper-responsive ex vivo, exhibit enhanced responses to cognate infections, and superior ability to induce experimental autoimmune diabetes in mice. ABIN1 negatively regulates p38 kinase activation and late NF-κB target genes. P38 is at least partially responsible for the upregulation of the key effector proteins IFNG and GZMB in ABIN1-deficient T cells after TCR stimulation. Our findings reveal the intricate role of ABIN1 in T-cell regulation.**

**Keywords** T Cells; Co-stimulation; Antigen Receptor; ABIN1; p38
**Subject Categories** Immunology; Post-translational Modifications & Proteolysis; Signal Transduction

## Introduction

T cells use their variable T-cell antigen receptors (TCR) to recognize cognate antigens to initiate adaptive immune responses to infections and cancer. To prevent overt T-cell responses, which might lead to autoimmune pathology, a plethora of mechanisms regulating T-cell activation has evolved, including the co-stimulatory and inhibitory surface receptors and intracellular regulators of signaling pathways. Understanding the individual regulatory steps of T-cell activation is critical for developing novel immunotherapeutic strategies to treat immune-related diseases such as cancer, infection, and autoimmunity. Several co-stimulatory receptors from the TNFR superfamily (e.g., GITR, OX40, CD137) are upregulated upon the initial antigen encounter. As these receptors enhance T-cell responses (Wortzman et al, 2013), some of them are used as targets of emerging anti-tumor immunotherapies (Chan et al, 2022; Van Beek et al, 2019; Duhen et al, 2021; Geuijen et al, 2021). However, despite the recent progress in uncovering the CD137 signalosome (Glez-Vaz et al, 2023), the signal transduction mechanisms of these co-stimulation TNFR superfamily receptors are largely unknown.

A20-binding inhibitor of NF-κB 1 (ABIN1 alias TNIP1) is an adaptor protein interacting with polyubiquitin signaling chains, NF-kappa-B essential modulator (NEMO), and A20, a zinc-finger protein with deubiquitinase activity (G'Sell et al, 2015). It has been shown that ABIN1 negatively regulates several signaling pathways, which use polyubiquitin chains for signal propagation, such as tumor necrosis factor receptor 1 (TNFR1), toll-like receptors, and autophagy signaling (G'Sell et al, 2015; Oshima et al, 2009; Callahan et al, 2013; Nanda et al, 2019; Nanda et al, 2011; Le Guerroue et al, 2023). Polymorphisms in *ABIN1* are associated with systemic lupus erythematosus in humans (Liu et al, 2018). Mice deficient in *Abin1* and mice expressing a severely compromised variant of *Abin1* develop TNF-triggered embryonal lethality (Oshima et al, 2009; Li et al, 2022) or a severe autoimmune disorder (Nanda et al, 2011), depending on the character of the *Abin1* modification and/or the genetic background of the mice. Whereas the importance of ABIN1 for the homeostasis and tolerance of both myeloid and B cells is well documented (Nanda et al, 2019; Nanda et al, 2011), much less is known about the role of ABIN1 in T cells. A recent study identified ABIN1 as a component of the CARD11-BCL10-MALT1 (CBM) complex (Yin et al, 2022), which is formed upon TCR signaling in T cells. Subsequent examination of the role of ABIN1 in a T-cell line Jurkat indicated its role in regulating the TCR-induced NF-κB pathway (Yin et al, 2022). However, the role of ABIN1 in CD8⁺ T cells has not been elucidated.

In this study, we characterize ABIN1 as a component of proximal signaling complexes of co-stimulatory receptors GITR and OX40, suggesting that ABIN1 negatively regulates both the antigenic signaling and T-cell co-stimulation. Using ABIN1-deficient primary T cells, we show that ABIN1 limits the effector responses of cytotoxic T cells during infection and autoimmunity.

[1]Laboratory of Adaptive Immunity, Institute of Molecular Genetics of the Czech Academy of Sciences, Prague, Czech Republic. [2]Faculty of Science, Charles University in Prague, Prague, Czech Republic. [3]Czech Centre for Phenogenomics, Institute of Molecular Genetics of the Czech Academy of Sciences, Prague, Czech Republic. [4]Laboratory of Immunity & Cell Communication, Division BIOCEV, First Faculty of Medicine, Charles University, Vestec, Czech Republic. ✉E-mail: ondrej.stepanek@img.cas.cz

# Results

## ABIN1 is a component of the proximal signaling complexes of GITR and OX40 receptors

Because the signal transduction pathway of co-stimulatory T-cell receptors GITR and OX40 was not completely understood, we analyzed the composition of GITR and OX40 signaling complexes (SC) using a proteomic approach previously used for the characterization of the TNFR1 and IL-17R SCs (Lafont et al, 2018; Knizkova et al, 2022; Draberova et al, 2020). First, we produced a tagged recombinant GITRL (Fig. 1A), which binds to PMA/ionomycin-pre-activated T cells (Fig. EV1A). Next, we used the GITRL to trigger the GITR signaling pathway in these cells, followed by cell lysis, purification of the ligand-receptor-SC by tandem affinity purification, and finally by mass spectrometry analysis to identify the SC composition (Fig. EV1B,C). The samples in which the ligand was added post lysis, served as negative controls to reveal contaminating proteins (Fig. EV1B,C). We identified known members of the GITR-SC, including TRAF1, TRAF2, CIAP2 (Wortzman et al, 2013) together with novel components, namely NF-kappa-B essential modulator (NEMO); subunits of the linear ubiquitin chain assembly complex (LUBAC): HOIP, HOIL1 and SHARPIN; A20 deubiquitinase; and adaptor protein ABIN1 (Fig. 1B,C). We confirmed the recruitment of ABIN1 and A20 into the GITR-SC by immunoprecipitation in DO11.10 hybridoma cells (Fig. 1D) and in primary murine OT-I $Rag2^{KO/KO}$ T cells (Fig. 1E). In the next step, we applied the same approach to another co-stimulatory TNFR superfamily receptor, OX40, to identify ABIN1 and A20 as novel components of the OX40-SC (Fig. EV1D–G). Overall, these experiments showed that A20 and ABIN1 are parts of the SC of GITR and OX40, and likely other co-stimulatory T-cell receptors from the TNFR superfamily and are thus, candidate negative regulators of T-cell activation through co-stimulation.

## Mouse models to study the role of ABIN1 in T cells

Based on the previous results, we hypothesized that ABIN1 is a negative regulator of T-cell activation. To study the role of ABIN1 in T cells, we aimed to establish a mouse model of ABIN1 deficiency. First, we used a mouse bearing a "knock-out first" gene trap (GT) $Abin1$ allele (Fig. EV2A). $Abin1^{GT/GT}$ mice lacked the full-length form ABIN1 but still expressed its low molecular weight form in T cells (Fig. EV2B). By sequential crossing of the $Abin1^{GT/GT}$ mice to transgenic CAG-Flp and then $Act$-CRE mice expressing specific recombinases, we generated exon 5 lacking (dE5) whole-body $Abin1^{dE5/dE5}$ mice (Fig. EV2A). Contrary to our predictions, the $Abin1^{dE5}$ allele was expressed as a truncated form of ABIN1 in T cells (Fig. EV2B). Both $Abin1^{GT/GT}$ and $Abin1^{dE5/dE5}$ mice were weaned at a sub-mendelian ratio (Fig. EV2C), suggesting pre-weaning lethality with incomplete penetrance.

We immunophenotyped $Abin1^{GT/GT}$ and $Abin1^{dE5/dE5}$ mice by flow cytometry (Appendix Figs. S1 and 2). Whereas the $Abin1^{dE5/dE5}$ mice did not show any apparent phenotype in T-cell and B-cell compartments (Appendix Fig. S3), we observed elevated CD4⁺ and CD8⁺ T-cell numbers and altered subset frequencies in the $Abin1^{GT/GT}$ mice (Appendix Fig. S4). To generate fully ABIN1-deficient mice, we crossed the $Abin1^{GT/GT}$ mice with the $Act$-CRE mice to generate the $Abin1^{GTKO}$ allele lacking exon 5 and still carrying part of the GT cassette (Fig. EV2A). $Abin1^{GTKO/GTKO}$ mice lacked the natural and truncated forms of ABIN1 (Fig. EV2B), were born in a sub-mendelian

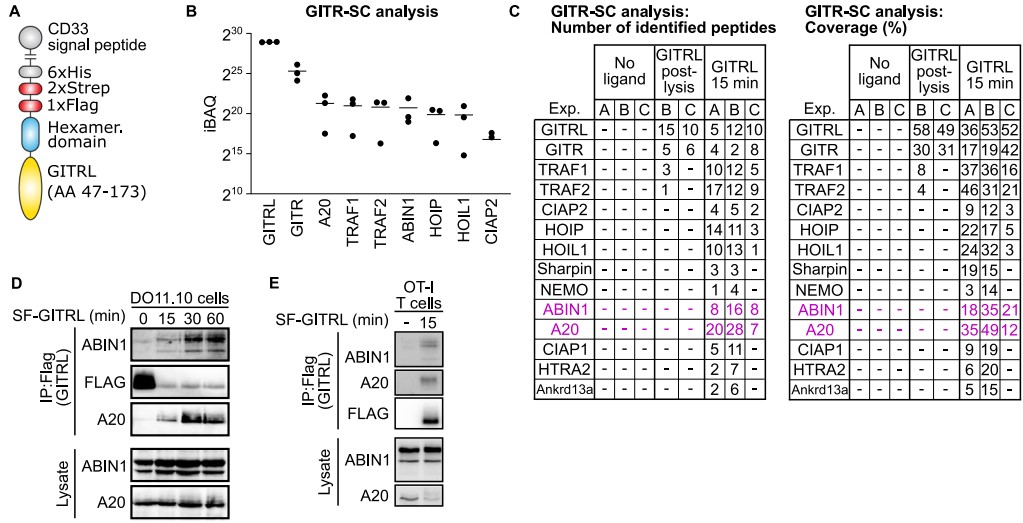

**Figure 1.  Analysis of the proximal GITR signaling complex (SC).**

(A) A schematic depiction of the recombinant GITRL for affinity purification. (B, C) Primary murine T cells were pre-activated with PMA/ionomycin for 72 h and stimulated with the recombinant GITRL for 15 min. The cells were lysed and GITR-SC was isolated via tandem affinity purification and analyzed by mass spectrometry. iBAQ score of GITR-SC proteins identified in all three biological replicates (B) and the number of peptides and coverage of GITR-SC proteins identified in at least two experiments (C) are shown. (D, E) DO11.10 cells (D) or primary mouse OT-I $Rag2^{KO/KO}$ T cells pre-activated with PMA/ionomycin (E) were activated with the SF-GITRL followed by immunoprecipitation and immunoblotting using indicated antibodies. Representative results out of two biological replicates in total are shown. The strong anti-FLAG band in the non-activated control (first lane) (D) is caused by the addition of the SF-GITRL directly to the cell lysate as a control for non-specific post-lysis interactions. Data information: In (B) data are presented as mean. Source data are available online for this figure.

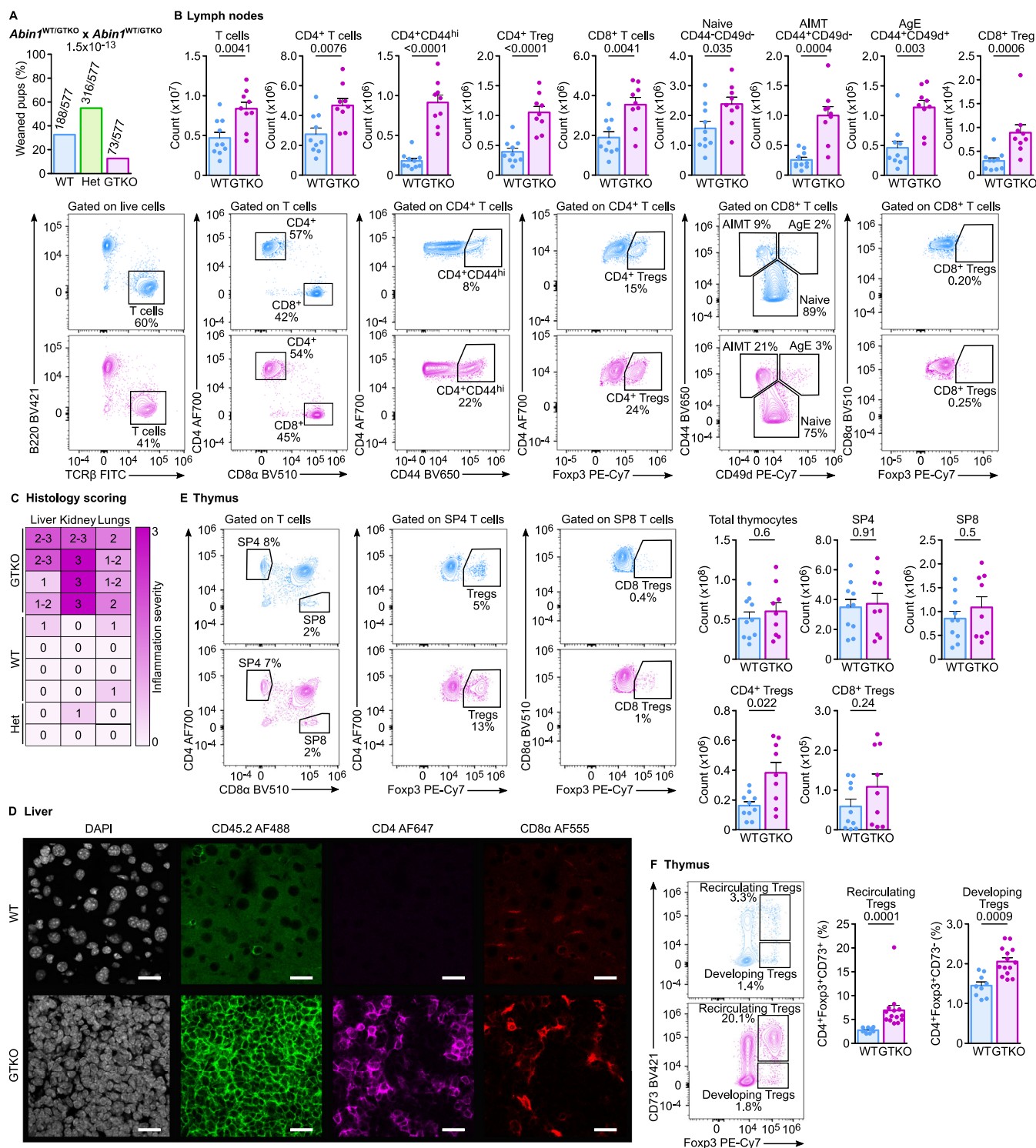

ratio (Fig. 2A), and showed altered T-cell and B-cell compartments similar to *Abin1*^GT/GT (Figs. 2B and EV2D,E; Appendix Fig. S4). Since these data indicated that *Abin1*^GTKO is a null or severely hypomorphic allele and that the *Abin1*^dE5 allele does not lead to functional ABIN1 deficiency in lymphocytes, we decided to continue with the *Abin1*^GTKO/GTKO mice.

## ABIN1 deficiency alters the lymphocyte compartment in extrinsic and intrinsic manners

*Abin1*^GTKO/GTKO had signs of autoimmune pathology demonstrated by tissue damage and infiltration of immune cells, mostly T cells, in the liver, kidney, and lungs (Figs. 2C,D and EV2F,G). Accordingly,

◄ **Figure 2. Characterization of the *Abin1*^WT/WT^ (WT) and *Abin1*^GTKO/GTKO^ (GTKO) mice.**

(A) Heterozygous *Abin1*^WT/GTKO^ (HET) mice were mated and the genotype of their offspring was determined upon weaning. The frequencies and numbers of pups with particular genotypes are indicated. *n* = 577 offspring mice in total from 18 breedings. (B) Lymph node cells were stained with indicated antibodies and analyzed by flow cytometry. Representative dot plots and aggregate counts of indicated subsets are shown. *n* = 10 (WT) or 9 (GTKO) mice per group. (C) Blinded histological scoring of indicated organs of 20–26 week-old mice based on H&E staining. 0—no pathology, 3—very strong leukocyte infiltration and tissue damage. *n* = 4 (WT and GTKO) or 2 (HET) mice per group. (D) Cryosections of livers of WT and GTKO mice were stained with indicated antibodies and DAPI (nuclei) and analyzed by confocal fluorescence microscopy. Representative sections out of four mice per group in total. (E, F) Fixed and permeabilized thymocytes from WT and GTKO mice were stained with indicated antibodies and analyzed by flow cytometry. (E) Representative dot plots and counts of indicated subsets are shown. *n* = 10 (WT) or 9 (GTKO) mice per group. (F) Gated SP4 thymocytes are shown. Representative dot plots and the frequencies of indicated subsets among SP4 cells are shown. *n* = 9 (WT) or 14 (GTKO) mice per group. Data information: In (B, E, F), the data are presented as mean + SEM and *P* values are indicated. (D) The scale bar represents 20 μm. Statistical significance was determined by a binomial test (A) or two-tailed Mann–Whitney test (B, E, F). AIMT antigen inexperienced memory-like T cells, AE antigen-experienced cells. Source data are available online for this figure.

*Abin1*^GTKO/GTKO^ showed an altered lymphocyte compartment, including expanded peripheral CD8^+^ and CD4^+^ T cells (Fig. 2B). A higher percentage of CD8^+^ T cells had phenotypes of CD44^+^ CD49d^+^ antigen-experienced (AE) or CD44^+^ CD49d^-^ antigen inexperienced memory-like T cells (AIMT) in *Abin1*^GTKO/GTKO^ than in the WT mice. Among CD4^+^ T cells, the *Abin1*^GTKO/GTKO^ mice had high frequencies of CD44^+^ activated cells and FOXP3^+^ regulatory T cells (Tregs).

The thymic development of T cells was comparable in the WT and *Abin1*^GTKO/GTKO^ mice (Figs. 2E and EV2H), indicating that the difference in the T-cell compartment occurs in mature T cells in the periphery. The only notable difference in the thymus was the abundance of CD4^+^ FOXP3^+^ Tregs. Based on their expression of CD73 (Fig. 2F) (Owen et al, 2019), we concluded that these additional Tregs recirculated to the thymus from the periphery.

To discriminate the intrinsic and extrinsic phenotypes of ABIN1 deficiency, we generated mixed bone marrow chimeras using a 1:1 mixture of Ly5.2 *Abin1*^GTKO/GTKO^ and congenic Ly5.1/Ly5.2 WT bone marrow donor cells transplanted into irradiated Ly5.1 WT hosts (Figs. 3A and EV3A; Appendix Fig. 2). We did not observe enhanced expansion and differentiation of *Abin1*^GTKO/GTKO^ T cells in these chimeras, suggesting that the steady-state changes in the *Abin1*^GTKO/GTKO^ mice were largely extrinsic. The count of peripheral T cells was even slightly lower in *Abin1*^GTKO/GTKO^ than in WT mice (Figs. 3A and EV3A). In contrast, the enhanced formation of Treg cells (Figs. 3A and EV3A) and the enhanced isotype switching of B cells and their differentiation into plasma cells (Figs. 3B and EV3B) were intrinsic phenotypes caused by the ABIN1 deficiency.

The *Abin1*^GTKO/GTKO^ Tregs showed slightly higher, but not significant, expression of GITR and OX40 than WT Tregs (Fig. EV3C) and had a comparable suppression capacity to WT Tregs ex vivo (Fig. 3C).

Overall, these data revealed extrinsic hyperactivation of T cells and intrinsically expanded Treg cells in *Abin1*^GTKO/GTKO^ mice.

## ABIN1 is a negative regulator of GITR co-stimulation signaling

To study the role of ABIN1 in GITR signaling, we activated T cells from the *Abin1*^GTKO/GTKO^ and WT mice with PMA/ionomycin overnight, followed by cultivation in media supplemented with IL-2 for 72 h to induce the expression of GITR (Fig. EV1A). Next, we stimulated them with GITRL. The GITR signaling triggered the NF-κB pathway (degradation of IκB) and p38 MAPK pathway (phosphorylation of p38) in CD4^+^ T cells (Fig. 3D). The ABIN1

deficiency led to slightly lower basal IκB levels in the non-activated T cells, but did not substantially alter IκB degradation in response to GITRL stimulation (Figs. 3D and EV3D). In contrast, p38 phosphorylation was elevated in CD4^+^ *Abin1*^GTKO/GTKO^ cells in comparison to WT cells, indicating that ABIN1 is a negative regulator of the GITR-p38 axis (Fig. 3D).

We did not observe excessive p38 phosphorylation and IκB degradation in polyclonal *Abin1*^GTKO/GTKO^ CD8^+^ T cells (Fig. EV3D). To remove the extrinsic effects of ABIN1 deficiency, such as the autoimmune environment, we crossed the *Abin1*^GTKO/GTKO^ mice to OT-I *Rag2*^KO/KO^ (henceforth OT-I) mice, which lack B cells, and the only T cells formed are H-2K^b^-SIINFEKL (OVA)-specific CD8^+^ OT-I cells. We observed slight hyper-responsiveness of these T cells to GITRL stimulation in terms of p38 phosphorylation, IκB degradation, and NF-κB nuclear translocation (Figs. 3E and EV3E,F). Overall, these results suggest the negative role of ABIN1 in the co-stimulation signaling.

## ABIN1 regulates TCR signaling via repressing p38 activation

The *Abin1*^GTKO/GTKO^ mice showed normal thymic development, but produced more mature SP8 T cells than WT mice, suggesting more efficient positive selection (Fig. EV4A). In the periphery, *Abin1*^GTKO/GTKO^ OT-I T cells had predominantly naïve CD44^-^ CD49d^-^ phenotype (Fig. EV4A). We observed higher frequencies of splenic antigen-experienced CD8^+^ T cells and TCR-negative splenocytes, perhaps innate lymphocytes, in the *Abin1*^GTKO/GTKO^ OT-I mice (Fig. EV4A). To examine the role of ABIN1 in the antigenic responses of primary T cells, we activated T cells isolated from *Abin1*^GTKO/GTKO^ and WT OT-I mice using anti-CD3/CD28 beads. The ABIN1-deficient T cells showed stronger upregulation of CD44, CD69, and CD25 than WT cells (Fig. 4A). In the next step, we activated *Abin1*^GTKO/GTKO^ and WT OT-I T cells with plate-bound anti-CD3ε antibody or K^b^-OVA monomer. In both cases, *Abin1*^GTKO/GTKO^ OT-I T cells showed more rapid proliferation, especially to the suboptimal concentration of the agonists (Fig. 4B).

In the next step, we analyzed the transcription profile of the *Abin1*^GTKO/GTKO^ and WT OT-I cells activated with anti-CD3/CD28 beads by RNA sequencing. ABIN1-deficient OT-I T cells triggered a different activation-induced reprogramming than WT OT-I T cells (Fig. EV4B). Differentially activated genes in *Abin1*^GTKO/GTKO^ OT-I T cells included effector molecules (*Gzmb* and *Ifng*), caspases (*Casp1* and *Casp4*), and protease inhibitors (*Serpina3f*, *Serpina3g*) (Fig. 4C,D; Dataset EV1). On the other hand, ABIN1-deficient

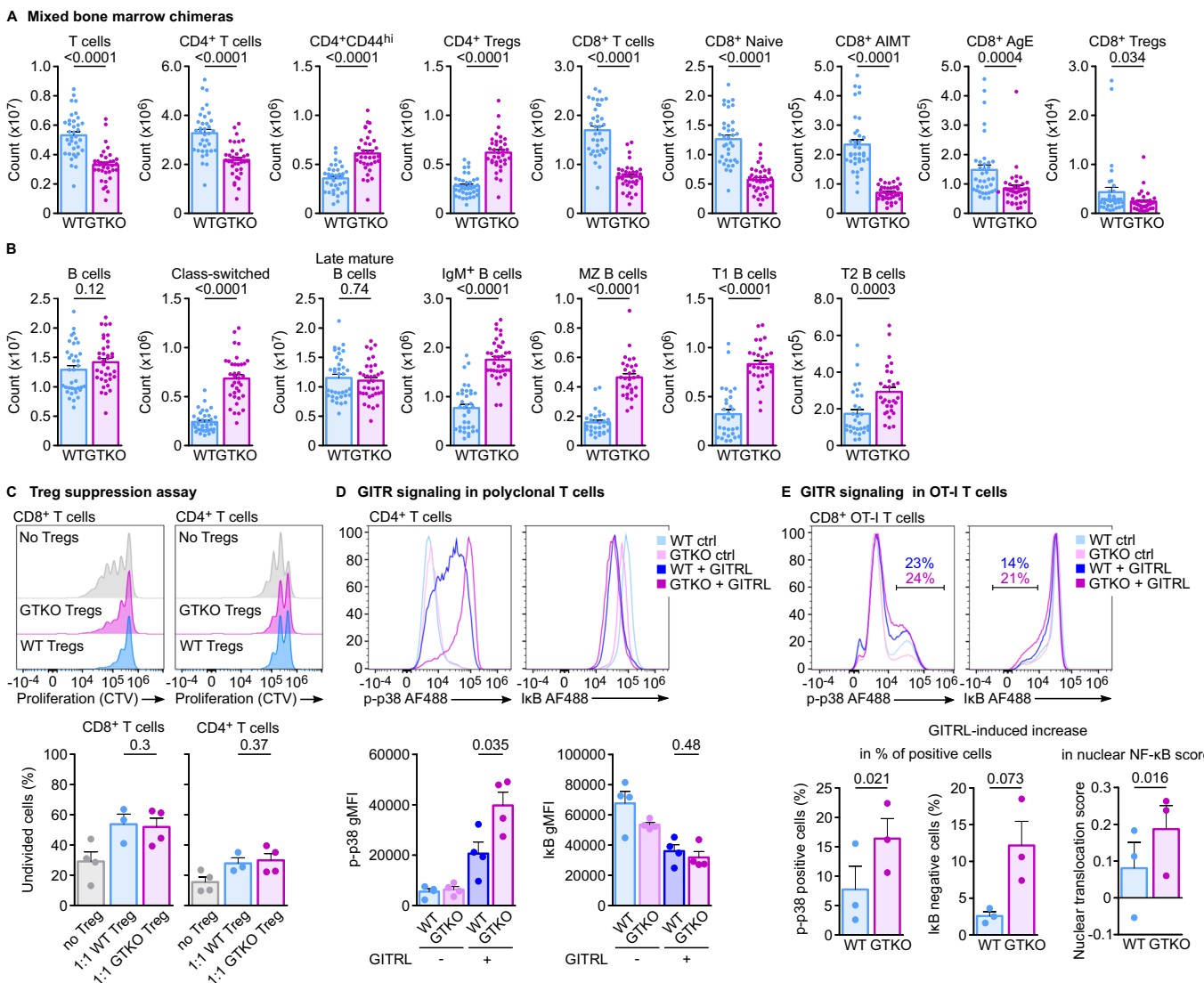

**Figure 3.  Intrinsic roles of ABIN1 in T cells.**

(**A, B**) Mixed bone marrow chimeras generated by transplanting Ly5.1/Ly5.2 *Abin1*^WT/WT (WT) and Ly5.2 *Abin1*^GTKO/GTKO (GTKO) bone marrow cells into irradiated Ly5.1 WT hosts, which were analyzed after 8 weeks post transplantation. Splenocytes were stained with the indicated antibodies and analyzed by flow cytometry. Counts of indicated subsets of T cells (**A**) and B-cell subsets (**B**) are shown. *n* = 36 mice per group. (**C**) CD4⁺ or CD8⁺ T cells were labeled with Cell Trace Violet dye (CTV) and FACS-sorted. CD4⁺ GFP⁺ (FOXP3⁺) Treg cells were FACS-sorted from DEREG⁺ WT or DEREG⁺ GTKO mice. CTV-loaded CD4⁺ or CD8⁺ T cells were mixed with WT Treg or GTKO Treg cells at 1:1 ratio. Cells were co-cultured for 72 h and their proliferation was measured by flow cytometry. As a control, CTV-loaded T cells were cultured alone. Representative histograms and the quantification of undivided cells are shown. *n* = 3 (WT) or 4 (GTKO) mice per group. (**D, E**) PMA/ionomycin pre-activated lymph node cells from WT or GTKO mice (**D**) or WT or GTKO OT-I *Rag2*^KO/KO mice (**E**) were activated with GITRL or left untreated (controls). Indicated signaling intermediates in CD4⁺ and CD8⁺ T cells were analyzed by flow cytometry. (**D**) Representative histograms and aggregate results of phospho-p38 and IκB levels in CD4⁺ T cells. *n* = 4 mice per group. (**E**) Representative histograms and aggregate results of phospho-p38 and IκB levels in OT-I T cells calculated as the percentage of phospho-p38-positive or IkB-negative cells in the activated sample minus the percentage of respective cells in the non-activated sample. *n* = 3 mice per group. When applicable, the results are shown as means + SEM and *P* values are indicated. Statistical significance was determined by two-tailed Mann–Whitney test (**A, B**). Data information: In (**A–E**) data are presented as mean + SEM and *P* values are indicated. Statistical significance was determined by two-tailed Mann–Whitney test (**A, B**), two-tailed Student's *t* test (**C, D**), or two-tailed paired Student's *t* test (**E**). AIMT antigen inexperienced memory-like T cells, AE antigen-experienced cells. Source data are available online for this figure.

T cells expressed lower levels of *Il17a*, *Il17f*, and *Il23r*, which are genes typical for unconventional Tc17 cells (Fig. 4C,D) (Huber et al, 2009). The caspases and protease inhibitors, but not the cytotoxic and Tc17 genes, were upregulated in the *Abin1*^GTKO/GTKO T cells even before the activation (Fig. EV4C). Accordingly, ABIN1-deficient T cells expressed higher levels of a transcription factor *Eomes*, but lower levels of *Rorc*, suggesting that ABIN1 suppresses the

conventional cytotoxic effectors differentiation program but promotes the Tc17 effectors. We confirmed the elevated expression of key effector molecules GZMB and IFNG in activated *Abin1*^GTKO/GTKO OT-I T cells by flow cytometry (Fig. 4E).

We used lists of validated NF-κB target genes responding to antigenic stimulation with different kinetics in B cells (Zhao et al, 2023). These genes were generally upregulated after the anti-CD3/

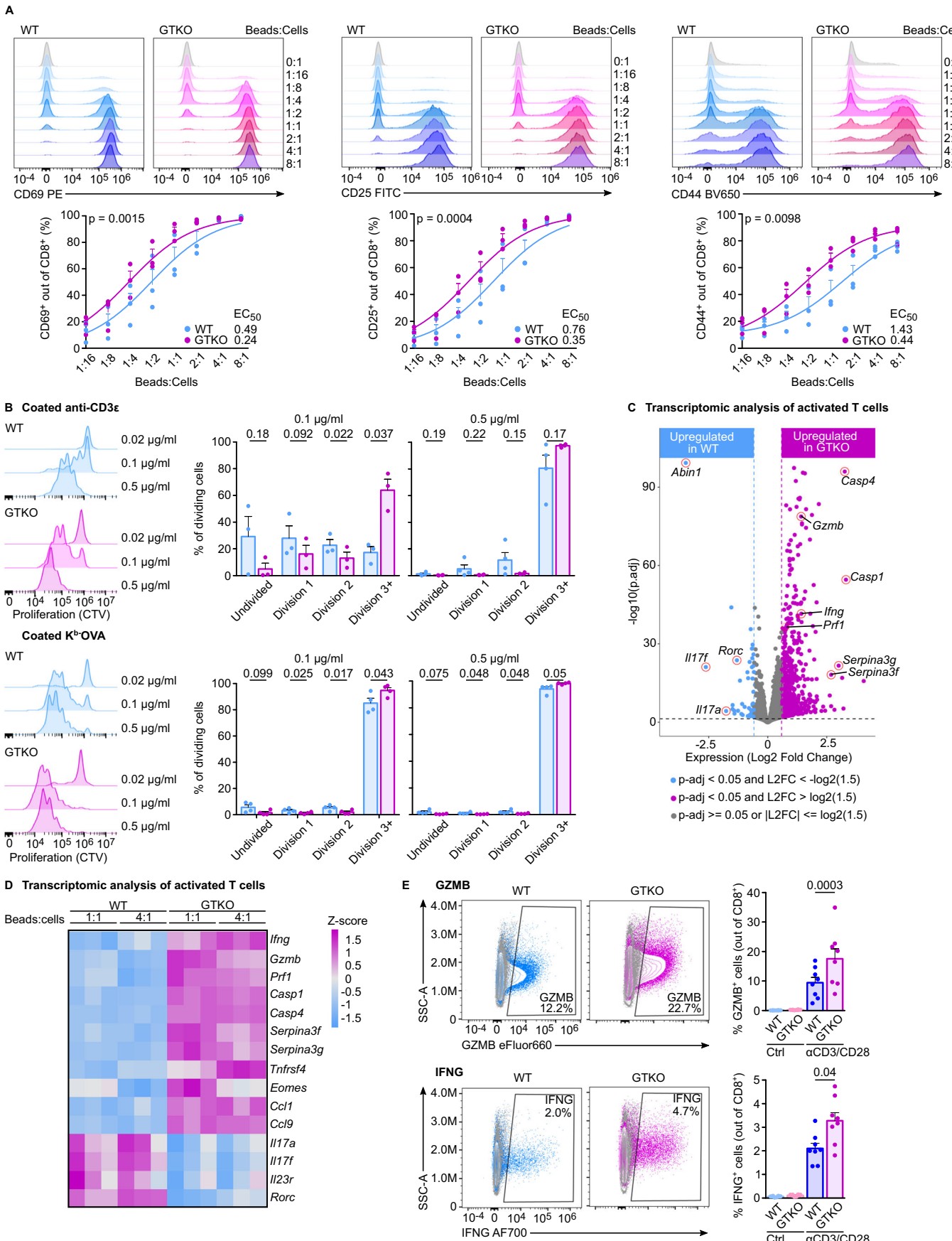

© The Author(s)

**Figure 4. ABIN1-deficient T cells are hyper-responsive.**

(A) FACS-sorted CD8$^+$ T cells from OT-I $Rag2^{KO/KO}$ $Abin1^{WT/WT}$ (WT) or $Abin1^{GTKO/GTKO}$ (GTKO) mice were activated with anti-CD3/CD28 beads at indicated ratios for 16 h and CD25, CD44, and CD69 was detected by flow cytometry. Representative histograms and aggregate results with log(agonist) vs. response nonlinear regression fits and EC$_{50}$ values are shown. $n = 3$ mice per group. (B) Splenocytes and lymph node cells from WT and GTKO mice were FACS-sorted and labeled with Cell Trace Violet Dye. Subsequently, the cells were cultivated in plates coated with indicated concentrations of anti-CD3 antibody or monomeric K$^b$-OVA for 72 h. Cell proliferation was analyzed by flow cytometry. Representative histograms and quantified frequencies in individual peaks are shown. $n = 4$ mice per group. (C, D) Lymph node cells from WT or GTKO mice were activated with anti-CD3/CD28 beads at 1:1 or 1:4 ratios for 16 h prior to RNA isolation and RNA sequencing. $n = 3$ mice per group. (C) A volcano plot. (D) A heatmap showing the differential expression of selected genes. (E) Lymph node cells from WT and GTKO mice were activated with anti-CD3/CD28 beads at 1:1 ratio for 16 h. Expression of GZMB and IFNG was analyzed by flow cytometry. Representative dot plots and aggregate results for indicated groups are shown. $n = 8$ (WT and GTKO) mice per group. Data information: In (A, B, E), data are presented as mean + SEM and $P$ values are indicated. Statistical significance was determined using the extra sum of squares F test (A), paired $t$ test (B), Wald test with Benjamini–Hochberg multiple testing correction (C), or by two-tailed Mann–Whitney test (E). Source data are available online for this figure.

CD28 activation in our cells as revealed by a gene set enrichment analysis (GSEA) (Fig. EV4D). Interestingly, the GSEA also revealed that NF-κB target genes as a group were upregulated in activated $Abin1^{GTKO/GTKO}$ OT-I T cells in comparison to WT OT-I T cells, with the exception of very transiently expressed genes (Fig. EV4D). This suggested that ABIN1 negatively regulates the NF-κB pathway with some delay, which keeps the expression of rapidly and transiently upregulated targets independent of ABIN1.

We subsequently investigated particular TCR signaling pathways regulated by ABIN1. We observed that freshly isolated and immediately fixed steady-state $Abin1^{GTKO/GTKO}$ OT-I T cells have higher levels of phospho-p38 than WT OT-I T cells (Fig. 5A). When we activated the $Abin1^{GTKO/GTKO}$ and WT OT-I T cells using T2-Kb cell line loaded with low (0.1 nM) or high (1 μM) concentration of OVA peptide, we observed slightly elevated phosphorylation of p38 in $Abin1^{GTKO/GTKO}$ OT-I T cells (Fig. 5B). In contrast, the phosphorylation of ERK1/2 and degradation of IκB was not enhanced $Abin1^{GTKO/GTKO}$ OT-I T cells (Fig. 5B). Moreover, ABIN1-deficient T cells showed slightly augmented nuclear translocation of NF-κB, but only very minor, if any, effect on NFAT nuclear translocation as shown by imaging flow cytometry (Fig. 5C). Overall, these results suggested that ABIN1 selectively regulates specific downstream signaling cascades leading to T-cell activation, particularly p38 and NF-κB pathways.

To address whether the differential activation of p38 in $Abin1^{GTKO/GTKO}$ and WT OT-I T cells contributes to the differences in their ex vivo responses to the antigen (Fig. 4), we activated T cells by immobilized anti-CD3ε antibody in the presence or absence of a p38 inhibitor. The proliferation advantage of $Abin1^{GTKO/GTKO}$ T cells was largely diminished by p38 inhibition (Fig. 5D). In the next step, we activated $Abin1^{GTKO/GTKO}$ and WT OT-I T cells with anti-CD3/CD28 beads in the presence or absence of the p38 inhibitor and performed the transcriptomic analysis by RNA sequencing (Fig. EV5A; Dataset EV2). The analysis of samples without the p38 inhibitor corresponded well to the results of the previous experiment (Fig. EV5B,C). We generated a set of genes, which were upregulated in $Abin1^{GTKO/GTKO}$ T cells in both experiments (Dataset EV3). Using a gene set enrichment analysis, we observed that many genes upregulated in activated $Abin1^{GTKO/GTKO}$ T cells are sensitive to the inhibition of the p38 pathway (Fig. 5E). The genes upregulated in ABIN1-deficient T cells and downregulated upon p38 inhibition included *Gzmb*, *Ifng*, and *Tnfrsf4* (OX40), but not *Casp1* and *Casp4* (Figs. 5F and EV5D). Accordingly, p38 inhibition decreased the protein levels of GZMB and IFNG in monoclonal and polyclonal CD8$^+$ T cells activated by anti-CD3/CD28 beads (Figs. 5G,H and EV5E). The activation-induced expression of CD25 was also

sensitive to p38 inhibition (Figs. 5G,H and EV5E). Overall, these data reveal ABIN1 as a negative regulator of p38 activation in CD8$^+$ T cells, which leads to upregulation of p38-dependent key effector genes in ABIN1-deficient T cells.

As our ex vivo data suggested that ABIN1 plays a role in the differentiation of CD8$^+$ T cells into Tc17 subtype, we tested whether it is involved in Th1 versus Th17 commitment of CD4$^+$ T cells in vivo. However, we did not observe differences in the expression of T-bet or RORγt in CD4$^+$ T cells from polyclonal $Abin1^{GTKO/GTKO}$ and WT mice (Fig. EV5F). $Abin1^{GTKO/GTKO}$ CD4$^+$ T cells showed a slightly, but not significantly, higher percentage of cells producing IFNG after PMA/ionomycin activation. The percentage of IL17A expressing CD4$^+$ T cells was comparable (Fig. EV5F), suggesting that ABIN1 does not substantially regulate the commitment of CD4$^+$ T cells, at least in this polyclonal model.

As the activation of T cells leads to metabolic reprogramming manifesting as enhanced glycolysis, we compared the energy metabolism of $Abin1^{GTKO/GTKO}$ and WT OT-I T cells activated by anti-CD3/CD28 antibodies. We observed slightly higher glycolysis in $Abin1^{GTKO/GTKO}$ than in WT T cells in the basal state, after activation, and at the maximum after the inhibition of oxidative phosphorylation by Rotenone and Antimycin A, but these differences were not significant (Fig. 5I). However, the inhibition of p38 at the time of activation lead to a significant decrease of maximal glycolysis in $Abin1^{GTKO/GTKO}$ T cells and to lesser extent also in WT T cells (Fig. 5I). These results suggested that p38 regulates the increase in the glycolytic capacity in T cells upon antigenic activation.

## ABIN1 is a negative regulator of cytotoxic T-cell responses in vivo

We adoptively transferred $Abin1^{GTKO/GTKO}$ or WT OT-I T cells into congenic Ly5.1 recipients, which were subsequently infected with *Listeria monocytogenes* expressing OVA (Lm-OVA) or its lower-affinity variant Q4H7 (Lm-Q4H7). We observed higher expansion, preferential differentiation of $Abin1^{GTKO/GTKO}$ OT-I T cells into KLRG1$^+$ IL7R$^-$ short-lived effector T cells (SLEC) at the expense of KLRG1$^-$ IL7R$^+$ memory precursors and higher expression of CD25 in comparison to the WT controls upon Lm-OVA infection (Figs. 6A and EV6A; Appendix Fig. S2). The hyper-responsiveness of $Abin1^{GTKO/GTKO}$ T cells was pronounced in the infection with *Listeria* expressing an altered suboptimal OT-I antigen Q4H7 (Lm-Q4H7) (Figs. 6B and EV6B). We obtained very similar results with $Abin1^{GT/GT}$ OT-I T cells (Fig. EV6C), i.e., T cells bearing the original

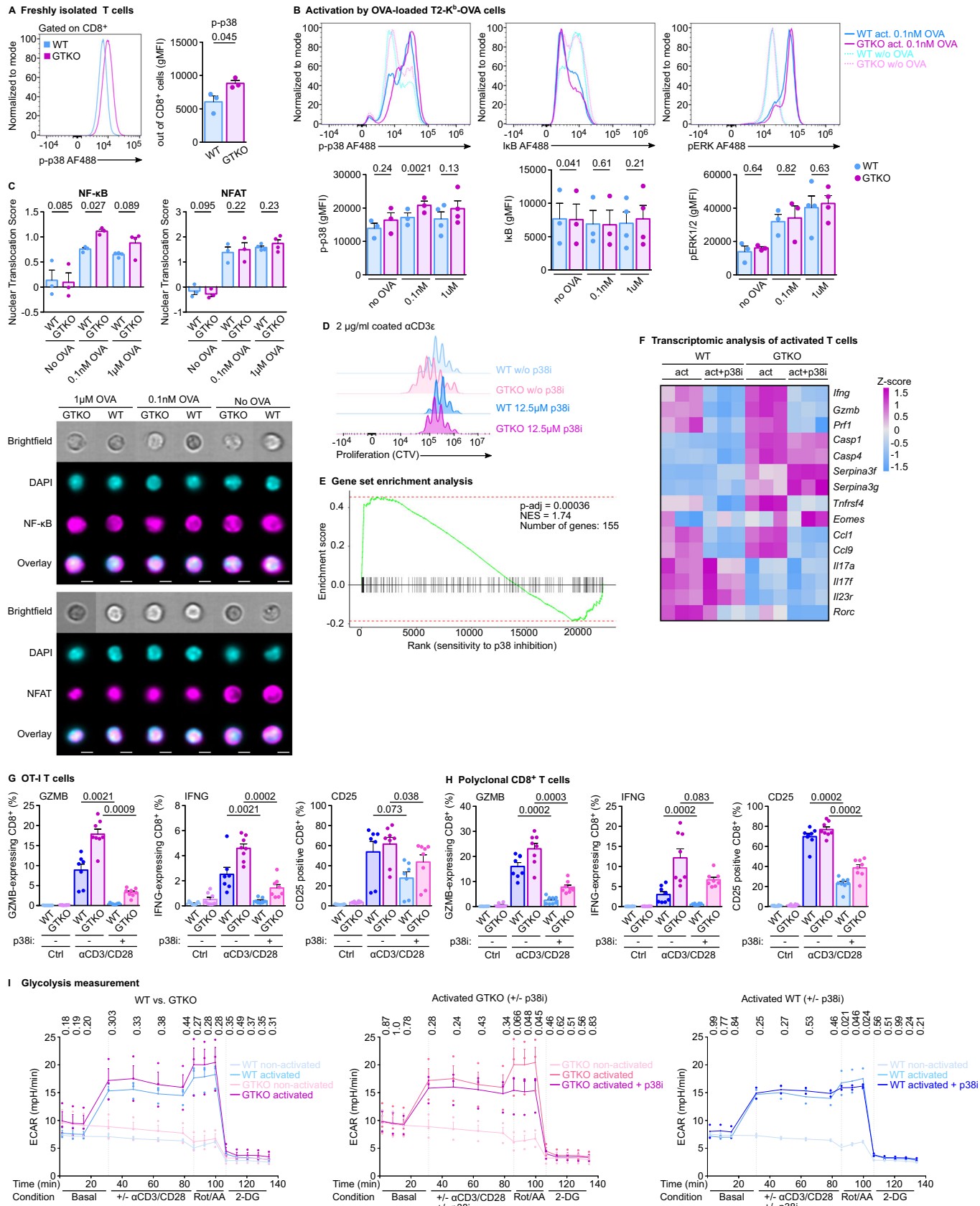

◀

**Figure 5. ABIN1 regulates p38 and NF-κB signaling pathways.**

(A) Freshly isolated and immediately fixed and permeabilized splenocytes from *Abin1*[WT/WT] OT-I *Rag2*[KO/KO] (WT) and *Abin1*[GTKO/GTKO] OT-I *Rag2*[KO/KO] (GTKO) were stained with anti-phospho-p38 antibody and analyzed by flow cytometry. Representative histograms are shown. *n* = 3 mice per group. (B, C) OT-I T cells were magnetically sorted (B) or FACS-sorted (C) from the WT and GTKO mice. T2-Kb cells were loaded with indicated concentrations of OVA peptide. Subsequently, CD8+ T cells were activated with T2-Kb cells at 2:1 ratio. (B) Activation of indicated signaling pathways was analyzed by flow cytometry. *n* = 4 mice per group. Representative histograms are shown. (C) Nuclear translocations of NF-κB p65 and NFAT were analyzed by imaging flow cytometry. The plots show median nuclear translocation scores from 3 (no OVA, 0.1 nM OVA) or 4 (1 μM OVA) mice per group. Representative images are shown. (D) Lymph node cells from WT and GTKO mice were labeled with Cell Trace Violet Dye. Subsequently, the cells were cultured in anti-CD3 coated plates for 72 h with or without p38 MAPK inhibitor SB203580 (12.5 μM) and analyzed by flow cytometry. Representative histograms are shown. *n* = 4 mice per group. (E, F) Lymph node cells from WT or GTKO mice were activated with anti-CD3/CD28 beads at 1:1 ratio with or without the p38 MAPK inhibitor (12.5 μM) for 16 h prior to RNA isolation and RNA sequencing. *n* = 3 biological replicates. (E) A gene set enrichment analysis using a set of genes which were upregulated in GTKO vs. WT T cells upon anti-CD3/CD28 in this and the previous set of experiments (Fig. 4C,D; Dataset EV3) and genes ranked according to their sensitivity to p38 inhibition upon activation (irrespective of the genotype). (F) A heatmap showing the differential expression of selected genes. (G, H) Lymph node cells from OT-I WT and GTKO (G) or polyclonal WT and GTKO (H) mice were activated with anti-CD3/CD28 beads at 1:1 ratio with or without the p38 MAPK inhibitor (12.5 μM) for 16 h. Expression of GZMB, IFNG, and CD25 was analyzed by flow cytometry. The frequencies of positive cells are shown. (G) *n* = 7 (WT), or 8 (GTKO) mice per group. (H) *n* = 8 mice per group. (I) The glycolysis of OT-I WT and GTKO mice was analyzed by measuring extracellular acidification rate (ECAR) at basal state, after anti-CD3/CD28 activation with or without p38 MAPK inhibitor (12.5 μM), after mitochondrial electron transport chain inhibition by Rotenone/Antimycin A (Rot/AA), and hexokinase inhibition by 2-deoxy-D-glucose (2-DG) as indicated. The *p* values calculated by two-tailed paired *t* test are shown for each data point and indicate the comparison between activated WT and GTKO cells (left, *n* = 3 mice per group), activated GTKO cells with and without p38 inhibition (middle, *n* = 3), or activated WT cells with and without p38 inhibition (right, *n* = 2). Data information: In (A–C, G–I), data are represented as mean + SEM. *P* value are indicated. (C) The scale bar represents 5 μm. Statistical significance was determined using unpaired *t* test (A), two-tailed Mann–Whitney test (B, C, G, H) or two-tailed Weighted Kolmogorov–Smirnov test (E), or two-tailed paired *t* test (I). Source data are available online for this figure.

GT version of the ABIN1 targeted allele (Fig. EV2A). The enhanced formation of SLEC cells in *Abin1*[GTKO/GTKO] OT-I T cells was also observed upon the infection with LCMV-OVA (Fig. EV6D).

We tested the response of *Abin1*[GTKO/GTKO] and WT OT-I T cells to tumors, using mice with implanted subcutaneous MC-38-OVA carcinomas. We adoptively transferred Ly5.2 *Abin1*[GTKO/GTKO] OT-I T cells or WT OT-I T cells mixed with competitor Ly5.1 WT OT-I T cells at 1:1 ratio into MC-38-OVA tumor bearing Ly5.1/Ly5.2 heterozygous mice. After seven days, we determined the relative abundance of WT and *Abin1*[GTKO/GTKO] OT-I T cells in the lymphoid organs and in the tumor as a ratio of Ly5.2+ to Ly5.1+ OT-I T cells (Figs. 6C and EV6E). Whereas the ratio of WT Ly5.2 OT-I to WT Ly5.1 OT-I cells was close to one, as expected, we observed a competitive advantage of *Abin1*[GTKO/GTKO] over the Ly5.1 WT OT-I T cells, which was most pronounced in the tumor (Fig. 6C). The strong ability of *Abin1*[GTKO/GTKO] OT-I T cells to infiltrate the tumor motivated us to assess the anti-tumor response of these cells. We noticed that both *Abin1*[GTKO/GTKO] and WT OT-I T are able to suppress the growth of the MC-38-OVA tumor when transferred at high numbers. However, we did not observe that *Abin1*[GTKO/GTKO] OT-I T cells have significantly better anti-tumor properties than their WT counterparts (Figs. 6D and EV6F).

The above-presented data indicated that ABIN1 is a negative regulator of T-cell responses in vivo and thus might be important for maintaining T-cell peripheral tolerance. We compared the ability of *Abin1*[GTKO/GTKO] and WT OT-I T cells to induce experimental autoimmune diabetes. The model is based on a transfer of relatively low numbers of OT-I T cells into a transgenic RIP.OVA mouse expressing OVA under the rat insulin promoter (Kurts et al, 1998) followed by their in vivo priming with bone marrow-derived dendritic cells loaded with OVA (DC-OVA) (Tsyklauri et al, 2023). First, we observed that *Abin1*[GTKO/GTKO] OT-I T cells showed stronger expansion and effector formation in response to DC-OVA (Fig. 6E). Accordingly, the ABIN1-deficient OT-I T cells were more potent in inducing experimental diabetes than their WT counterparts measured as glucose in the blood and urine, insulitis, and immune cell infiltration in pancreatic islets, and OT-I expansion in the spleen (Figs. 6F–I and EV6G,H), suggesting

that *Abin1*[GTKO/GTKO] T cells have a supraphysiological ability to undergo antigen-induced expansion, effector cell formation, tissue infiltration, and target cell killing, which enables them to escape the peripheral tolerance largely established by Tregs in this model (Tsyklauri et al, 2023).

Altogether, our data identify ABIN1 as an intrinsic negative regulator of CD8+ T-cell responses via inhibiting antigen-induced and co-stimulatory signaling cascades, especially the p38 MAPK pathway.

# Discussion

We have identified ABIN1 as a part of the SCs of two T-cell co-stimulation receptors, GITR and OX40. Most likely, ABIN1 is recruited to the complex via its binding to linear polyubiquitin chains generated by the LUBAC ubiquitin ligase (Shinkawa et al, 2022), which was also identified in our analysis of the SC of GITR. LUBAC is a signal transducer in TNFR1 and MYD88 signaling (Iwai, 2021). Recently, LUBAC subunits were identified in the SC of CD137 TNFR superfamily receptor (Glez-Vaz et al, 2023), but its involvement in GITR signaling was not shown before. We speculate that the recruitment of LUBAC, which generates linear poly-ubiquitin chains and activates NEMO to trigger downstream NF-κB activation, is a common mechanism of signaling of T-cell co-stimulation receptors from the TNFR superfamily (Tokunaga et al, 2009). The recent report analyzed the SC of CD137 to find components largely overlapping with our detection of the GITR-SC (LUBAC, cIAP2, TRAF1, TRAF2, A20), although the authors did not detect ABIN1 (Glez-Vaz et al, 2023), perhaps for technical reasons. Recently, ABIN1 was reported to interact with the CARD11-CBM complex upon TCR signaling and to inhibit the TCR-induced NF-κB activation in a human T-cell line (Yin et al, 2022), indicating that ABIN1 regulates antigenic as well as co-stimulation signaling in T cells.

We characterized the role of ABIN1 in primary T cells. Our *Abin1*[GTKO/GTKO] mouse model largely recapitulated previous models of ABIN1 deficiency, such as partial pre-weaning lethality, intrinsic

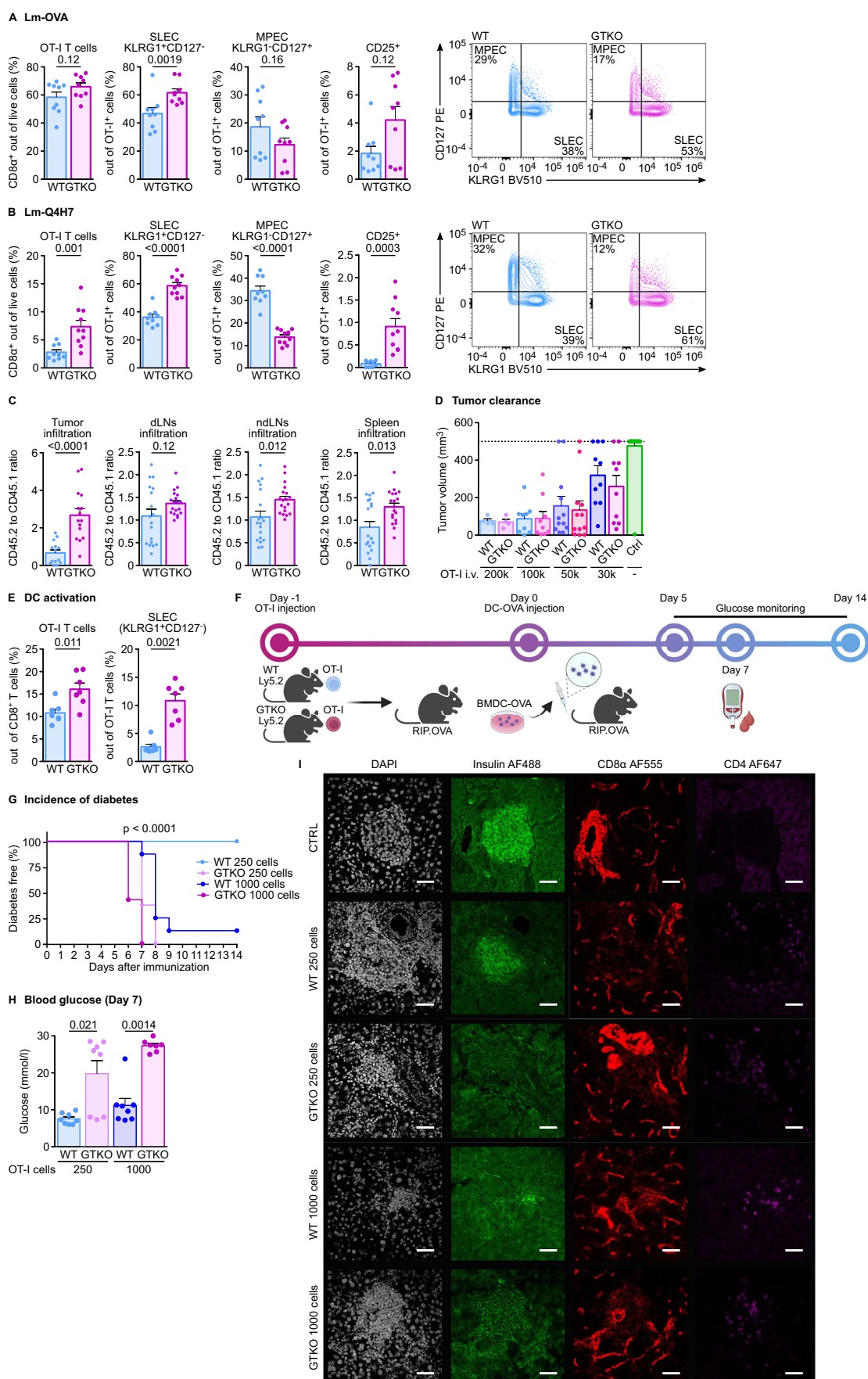

**Figure 6. ABIN1 regulates T-cell responses in vivo.**

(A, B) Overall, $1 \times 10^5$ T cells from $Abin1^{WT/WT}$ OT-I $Rag2^{KO/KO}$ (WT) or $Abin1^{GTKO/GTKO}$ OT-I $Rag2^{KO/KO}$ (GTKO) were adoptively transferred to Ly5.1 hosts that were infected with *Listeria monocytogenes* expressing (A) ovalbumin (Lm-OVA) or (B) its lower-affinity variant (Q4H7). Splenocytes were analyzed by flow cytometry on day 6 post infection. Representative dot plots and frequencies of indicated T-cell subsets are shown. (A) $n = 9$ mice per group. (B) $n = 9$ (WT) or 10 (GTKO) mice per group. (C) Lymph node cells from Ly5.1 WT OT-I $Rag2^{KO/KO}$ mice were mixed at a 1:1 ratio with cells from Ly5.2 WT or GTKO littermates, and mixed donor cells were adoptively transferred to congenic Ly5.1/Ly5.2 WT mice bearing MC-38 tumors. On day 7 post transfer, the tumors, draining lymph nodes, non-draining lymph nodes, and spleens were analyzed by flow cytometry and the ratio of Ly5.2 (WT or GTKO) to Ly5.1 (WT) donor cells was calculated. $n = 18$ mice per group. (D) Indicated numbers of lymph node cells from WT and GTKO OT-I mice were adoptively transferred into CD3ε$^{-/-}$ host-bearing MC-38 tumor. No cells were transferred in the control group. The tumor volume was monitored for consecutive 23 days, or up to the endpoint volume of 500 mm³. The maximal tumor volume for each mouse is shown. $n = 4$ mice for $2 \times 10^5$ transferred cells, 9 for $1 \times 10^5$ transferred cells, 12 (WT and GTKO) for $5 \times 10^4$ transferred cells, 10 for $3 \times 10^4$ transferred cells, or 21 for no transfer per group. (E) In total, $1 \times 10^4$ WT or GTKO OT-I T cells were adoptively transferred into congenic Ly5.1 mice followed by immunization with $1 \times 10^6$ OVA-loaded BMDCs on the next day. The frequency and phenotype of the OT-I T cells was analyzed on day 6 post immunization by flow cytometry. $n = 7$ mice per group. (F–H) Indicated numbers of WT or GTKO OT-I T cells were adoptively transferred to RIP.OVA host mice. On the following day, mice were immunized with OVA-loaded bone marrow-derived dendritic cells. The concentration of glucose in urine was monitored on a daily basis for 14 days and blood glucose was measured on day 7. The mouse was considered diabetic when the concentration of glucose reached 1000 mg/dl in urine. $n = 8$ mice per group. (F) The scheme of the experiment. (G) Kaplan Meyer curves showing the onset of diabetes. (H) Blood glucose concentration on day 7. (I) In the experimental setup shown in (F), mice have received 250 or 1000 WT or GTKO OT-I T cells as indicated and were sacrificed on day 6. Pancreatic sections of the host mice and control untreated mice were stained with indicated antibodies and DAPI (nuclei) and analyzed by confocal fluorescence microscopy. One pancreatic islet per condition is shown out of four mice per group in total. Please note the background of the anti-CD8α antibody, which is stronger than the specific signal coming from infiltrating CD8$^+$ T cells. Data information: In (A–E, H), data are represented as mean + SEM and *P* values are indicated. (I) The scale bar represents 45 µm. Statistical significance was determined using two-tailed Mann–Whitney test (A–C, E, H) or Log-rank (Mantel–Cox) test (G). Source data are available online for this figure.

hyperactivation of B cells, high frequencies of antigen-experienced T cells, and immune infiltrates in tissues (Nanda et al, 2019; Nanda et al, 2011). Whereas we observed increased numbers of CD4$^+$ and CD8$^+$ T cells in the lymph nodes, but not in the spleens of the $Abin1^{GTKO/GTKO}$. A previous study even observed decreased T-cell numbers in the spleen of ABIN1-deficient mice (Zhou et al, 2011). Accordingly, our analysis of mixed bone marrow chimeras showed relatively low numbers of $Abin1^{GTKO/GTKO}$ T cells indicating that the increased numbers of T cells in the lymph nodes of $Abin1^{GTKO/GTKO}$ was caused by an extrinsic factor. On the contrary, the enhanced formation of CD4$^+$ Treg cells in $Abin1^{GTKO/GTKO}$ mice, also previously observed in $Abin1$ knock-in mice with disrupted ubiquitin binding (Nanda et al, 2011), is a T-cell intrinsic phenotype.

To limit the role of the inflammatory environment, we crossed the $Abin1^{GTKO/GTKO}$ mice to monoclonal OT-I $Rag2^{KO/KO}$ mice, completely lacking the B-cell compartment and T-cell diversity. The $Abin1^{GTKO/GTKO}$ OT-I $Rag2^{KO/KO}$ mice showed more mature SP8 thymocytes than their $Abin1^{WT/WT}$ counterparts, probably because of enhanced positive selection. We assume that their altered thymic development has only negligible effects on the subsequent biology of peripheral T cells, although we could not formally confirm it. We compared WT and ABIN1-deficient OT-I T cells in ex vivo assays and in experiments based on the adoptive transfer into WT hosts to normalize the environment.

$Abin1^{GTKO/GTKO}$ OT-I T cells exhibited enhanced responses to antigenic, anti-CD3/CD28, and GITRL stimulation ex vivo. Moreover, they showed upregulation of key effector molecules GZMB and IFNG. We observed slightly upregulated NF-κB and p38 pathways and virtually undetectable effects on the ERK and NFAT pathways upon the ex vivo activation. The over-activation of the NF-κB pathway in ABIN1-deficient T cells was highly expected based on the previous reports (Yin et al, 2022; G'Sell et al, 2015). However, the differences in the signaling intermediates such as IκB or nuclear NF-κB translocation were relatively minor, at the borderline of statistical significance in our experiments. The transcription analysis showed that ABIN1 fails to regulate immediate and transient NF-κB target genes, but limits the expression of genes activated by NF-κB with slower kinetics after TCR activation. Altogether, these data suggest that ABIN1 does not regulate the initiation of the NF-κB signaling pathway, but is rather involved in the termination of the NF-κB activation induced by TCR and GITR signaling. The inhibition of p38 resulted in the downregulation of a significant proportion of genes, which were also upregulated in the $Abin1^{GTKO/GTKO}$ OT-I T cells upon activation, including GZMB and IFNG.

To our knowledge, the fact that the p38 pathway upregulates the glycolytic switch and expression of GZMB upon TCR activation was not previously shown, although there is a report that p38 controls GZMB expression in neutrophils in rats (Martin et al, 2018). The upregulation of IFNG by p38 was previously demonstrated for CD8$^+$ as well as CD4$^+$ T cells (Merritt et al, 2000; Rincon et al, 1998). It was also proposed that the p38 pathway contributes to the T-cell apoptosis (Merritt et al, 2000) and senescence (Lanna et al, 2014; Henson et al, 2014). Although we did not specifically focus on this question, the ~2 fold lower numbers of $Abin1^{GTKO/GTKO}$ than WT CD8$^+$ T cells in the periphery of the mixed bone marrow chimeras suggested, that the enhanced apoptosis might occur. A recent study showed that p38 inhibition in cultured activated murine T cells leads to upregulation of memory-signature genes (*Tcf7*) and downregulation of effector genes (*Klrg1*, *Gzmc*, *Prf1*, *Havcr2*) (Gurusamy et al, 2020), which is essentially in line with our in vitro and in vivo data with $Abin1^{GTKO/GTKO}$ and p38-inhibited T cells. The hyperactivation of p38 in ABIN1-deficient T cells leading to the preferential formation of effector T cells on the expense of memory T cells might explain why these cells rapidly infiltrate to the tumor, but are not superior in the tumor clearance for which stem-like T cells are essential (Connolly et al, 2021).

Although not explicitly mentioned in that article, their data show downregulation of *Ifng* and *Gzmb* upon p38 inhibition in these cells, which corresponds with our data (Gurusamy et al, 2020). However, they also showed that deletion of p38 *Mapk14* improves the anti-cancer properties in adoptive T-cell therapy in mice and enhances the proliferation and expression of effector molecules in human T cells (Gurusamy et al, 2020). In a striking contrast to their and our murine data, their study also describes

enhanced IFNG production upon p38 inhibition in human CD8$^+$ T cells (Gurusamy et al, 2020), indicating a species-dependent or context-dependent role of p38. Overall, these discrepancies uncover that the role of p38, and potentially also the role of ABIN1, in T-cell biology is complex and requires further studies to be completely understood.

P38 is activated via TAB2/3-TAK1 complex, which is recruited via K63 polyubiquitin chains in multiple signaling pathways (David et al, 2018). The K63 polyubiquitin chains are formed by TRAF6 ubiquitin ligase in the CBM complex and plausibly by TRAF2/cIAP2 in the GITR complex. Since ABIN1 recruits and/or activates A20 in various signaling complexes, we propose that ABIN1 might downregulate the p38 pathway via A20, which cleaves the K63 polyubiquitin chains (Verstrepen et al, 2010), leading to the release and deactivation of the TAB2/3-TAK1 complex. Additionally, the TCR signaling was shown to activate p38 via an alternative pathway, triggered by the direct phosphorylation of p38 by a proximal ZAP70 kinase (Jun et al, 2019). It is unclear, whether and how this pathway might be related to the ABIN1-mediated regulation of p38. Although the p38 pathway is upregulated in the $Abin1^{GTKO/GTKO}$ T cells and seems to be at least partially responsible for the observed phenotype, other downstream signaling pathways, such as the NF-κB pathway, probably also play a role in the hyperactivation of $Abin1^{GTKO/GTKO}$ T cells observed in vitro and in vivo.

We have seen that the deficiency of ABIN1 was coupled with the upregulation of key effector molecules GZMB and IFNG characteristic for type 1 immunity, but also with the downregulation of $Il17a$, $Il17f$, $Il23$, and $Rorc$, which are typical for type 3 immunity and expressed in a subset of Tc17 cells (Luckel et al, 2020). This suggests that ABIN1 not only generally regulates T-cell activation, but it also differentially regulates the outcomes of individual signaling pathways underlying T-cell differentiation program. We did not observe an effect of ABIN1 on the formation of Th17 CD4$^+$ T cells in the steady-state polyclonal mice. However, the potential role of ABIN1 on the fate of CD4$^+$ T cells should be investigated using more suitable models, such as monoclonal MHCII-restricted TCR transgenic mice in future studies.

Although we did not observe a superior anti-tumor activity of ABIN1-deficient T cells, it is possible that in another context with stronger antigenic and/or co-stimulation signaling, ABIN1 deficiency would increase anti-tumor potency. This is indicated by the experiments with autoimmune diabetes, in which $Abin1^{GTKO/GKTO}$ T cells were very powerful in the disease induction, probably as a consequence of their strong response to the priming by cognate inflammatory bone marrow-derived dendritic cells. One potential direction for future research is the role of ABIN1 in adoptive cancer therapies using T cells with chimeric antigen receptors consisting of a canonical TCR signaling unit (ZETA chain) and TNFR superfamily signaling domain (CD137) (Singh and Maus, 2023), both potentially regulated by ABIN1.

Overall, we show that ABIN1 is an important regulator of CD8$^+$ T-cell activation, which limits the proliferation and formation of effector T cells and maintains peripheral tolerance. ABIN1 inhibits p38 and NF-κB pathways, probably via delivering A20 deubiquitinase to NEMO (Yin et al, 2022; Mauro et al, 2006) and/or by physically blocking the polyubiquitin binding sites that recruit key signaling proteins. Our findings that ABIN1 acts as a negative regulator of CD8$^+$ T-cell immune response in vivo open potential research directions to study ABIN1 as a target for T-cell therapy.

# Methods

## Cell lines

Mouse DO11.10 cell line (RRID:CVCL_4163, provided by T. Brdicka, IMG) and human T2-Kb (Alexander et al, 1989) (provided by E. Palmer, University Hospital Basel) were cultivated in RPMI media (Sigma). Human HEK293FT (RRID:CVCL_6911, provided by T. Brdicka, IMG) and mouse MC-38 cells expressing OVA (Horkova et al, 2023) were cultivated in DMEM media (Sigma). The media were supplemented with 10% fetal bovine serum (FBS, Gibco #10270-106), 100 U/ml penicillin (BB Pharma #15/156/69-A/C), 100 μg/ml streptomycin (Sigma #59137), and 40 μg/ml gentamicin (Sandoz), and cells were cultured in 37 °C, 5% $CO_2$ incubator. Cell lines were regularly tested for mycoplasma by PCR. The authentication was performed based on typical cell morphology and features.

## Antibodies, dyes, and inhibitors

Following antibodies were used for flow cytometry: anti-CD1d PE-Cy7 (clone 1B1, Biolegend #123524), anti-CD4 AF700 (clone RM4-5, #100536), anti-CD8α BV421 (clone 53-6.7, #100753), anti-CD8α BV510 (clone 53-6.7, #100752), anti-CD8α FITC (clone 53-6.7, #100706), anti-CD8α PE-Cy7 (clone 53-6.7, #100722), anti-CD19 PE (clone 6D5, #115508), anti-CD23 APC (clone b3b4, #101620), anti-CD24 FITC (clone M1/69, #101806), anti-CD25 BV605 (clone PC61, #102036), anti-CD25 FITC (clone PC61, #102006), anti-CD38 AF488 (clone 90, #102714), anti-CD44 BV650 (clone IM7, #103049), anti-CD45 AF700 (clone 30-F11, #103127), anti-CD45.1 AF700 (clone A20, #110724), anti-CD45.1 FITC (clone A20, #110706), anti-CD45.1 PerCP-Cy5.5 (clone A20, #110728), anti-CD45.2 AF700 (clone 104, #109822), anti-CD45.2 APC (clone 104, #109814), anti-CD45.2 APC-Cy7 (clone 104, #109824), anti-CD45.2 FITC (clone 104, #109806), anti-CD45.2 PE (clone 104, #109808), anti-CD45R/B220 BV421 (clone RA3-6B2, #103240), anti-CD49d PE-Cy7 (clone R1-2, #103618), anti-CD69 PE (clone H1.2F3, #104508), anti-CD73 BV421 (clone TY/11.8, #127217), anti-CD127 PE (clone SB/199, #121112), anti-CD138 BV421 (clone 281-2, #142523), anti-CD138 BV510 (clone 281-2, #142521), anti-IgD AF700 (clone 11-26c.2a, #405730), anti-IgM BV421 (clone RMM-1, #406518), anti-IgM PE-Cy7 (clone RMM-1, #406513), anti-IFNγ AF700 (clone XMG1.2, #505824), anti-KLRG1 BV510 (clone 2F1/KLRG1, #138421), anti-TCRβ APC (clone H57-597, #109212), anti-TCRβ BV711 (clone H57-597, #109243), anti-TCRβ FITC (clone H57-597, #109206), anti-TCRVα2 FITC (B20.1, #127806), anti-TCRVβ5.1/5.2 (MR9-4, #139506), anti-FLAG APC (clone L5, #637307), anti-FOXP3 PE-Cy7 (clone FJK-16s, eBioscience #25-5773-82), anti-GZMB eFluor660 (clone NGZB, #50-8898-82), anti-NF-κB p65 (clone C-20, Santa Cruz #sc-372), anti-IκB AF488 (clone L35A5, Cell Signaling #5743S), anti-NFAT XP (clone D43B1, #5861S), anti-pERK XP (clone D13.14.4E, #4370), anti-p-p38 (clone D3F9, #4511).

Following dyes were used for flow cytometry: Cell Trace Violet (CTV, Invitrogen #34557), Hoechst 33258 (Invitrogen #H3569), LIVE/DEAD near-IR (ThermoFisher #L34976).

Following primary antibodies were used for immunofluorescence microscopy: anti-CD4 AF647 (clone RM4-5, Biolegend #100530), anti-CD8α (clone EPR21769, Abcam #ab217344), anti-CD45.2 AF488 (clone 104, Biolegend #109816), DAPI (ThermoFisher #D1306).

Following primary antibodies were used for immunoblotting, cell activation: anti-CD3ε (clone 145-2C11, Biolegend #100302), anti-ABIN1 (MRC PPU Reagents and Services #S345C), anti-A20 (clone A-12, Santa Cruz #sc-166692), anti-β-Actin (clone AC-15, Sigma #A1978), anti-p-p38 (clone D3F9, Cell signaling #4511), anti-DYKDDDDK Tag (clone D6W5B, Cell Signaling #14793).

Following secondary antibodies were used: goat anti-rabbit conjugated with AF488 (Invitrogen #A-11008), AF555 (Invitrogen #A-32732), or AF647 (Invitrogen #A-21245), donkey anti-sheep-HRP (Jackson #713-035-147), goat anti-mouse-HRP (Jackson #115-035-146), goat anti-rabbit-HRP (Jackson #111-035-144).

P38 MAPK inhibitor SB203580 (#ab120162) was purchased from Abcam.

## Mice

All the mice had C57Bl/6J background. CD3ε$^{KO/KO}$ (RRID:IMSR_JAX:004177) (Sommers et al, 2000), DEREG (RRID: MMRRC_032050-JAX) (Lahl et al, 2007), Ly5.1 (RRID: IMSR_JAX:002014) (Jang et al, 2018), OT-I Rag2$^{KO/KO}$ (RRID:MGI:3783776, MGI:2174910) (Palmer et al, 2016; Shinkai et al, 1992), RIP.OVA (RRID:IMSR_JAX:005433) (Kurts et al, 1998) strains were described previously.

We generated Abin1$^{GT/GT}$ mice via in vitro fertilization of C57BL/6J oocytes with sperm carrying the GT allele (Tnip1t-m1a(EUCOMM)Hmgu; RRID:IMSR_EUMMCR:1521) obtained from Infrafrontiers.

The Abin1$^{GT/GT}$ mice were crossed to Cre-deleter mice Gt(ROSA) 26Sor$^{tm1(ACTB-cre,-EGFP)Ics}$ (MGI: 5285392) (Act-CRE) to generate Abin1$^{GTKO/GTKO}$ mice missing exon 5 in Abin1 and carrying a LacZ sequence after the exon 4 of Abin1. The Abin1$^{GT/GT}$ mice were crossed to FLP-deleter mice Gt(ROSA)26Sor$^{tm2(CAG-flpo,-EYFP)Ics}$ (MGI: 5285396) (CAG-Flp) to generate Abin1$^{flox/flox}$ mice, followed by crossing to Act-CRE to cut out exon 5 to obtain Abin1$^{dE5/dE5}$ mice.

ARRIVE Essential10 guidelines (https://arriveguidelines.org/) were followed. Mice were kept at the Institute of Molecular Genetics of the Czech Academy of Science in a specific-pathogen-free facility. In the facility, 12 h light–dark cycle was maintained, food and water ad libitum. Animal protocols were in accordance with the laws of the Czech Republic and approved by the Czech Academy of Science (identification no. 72/2017, 81/2018, 2378/2022 SOV II).

For the experiment, males and females were used. Mice were used in age 6–12 week at the beginning of the experiments. For the developmental studies, mice at age 2–3 weeks old were used. When possible, age- and sex-matched animals were used, if possible littermates were used and equally divided into experimental groups. Mice were divided into groups by their ID without prior contact with the experimenter. The histology examination and one out of two diabetic experiments were blinded. Other experiments were not blinded as no subjective scoring was used. The sample size was determined as a minimal number of mice to provide conclusive results based on our previous experience.

## Magnetic sorting

CD4$^+$ or CD8$^+$ T cells were negatively sorted using biotinylated anti-B220 (clone RA3-6B2, Biolegend #103204), anti-CD11b (clone M1/70, Biolegend #101204), and anti-CD8α (clone 2.43, ATCC# TIB-210) or anti-CD4 (clone 53-6.7, Biolegend #100704) antibodies, respectively. Subsequently, the cells were labeled with anti-Biotin MicroBeads (Miltenyi Biotec #130-090-485) and enriched using an AutoMACS Pro (Milteniy Biotech).

## Flow cytometry and cell sorting

Cells were stained with indicated antibodies for cell sorting or flow cytometry. To distinguish live cells, samples were stained with LIVE/DEAD near-IR dye (ThermoFisher #L34976) or Hoechst 33258 (Invitrogen #H3569). For staining of FOXP3 transcription factor, cells were fixed using Foxp3/Transcription Factor Staining Buffer Set (Invitrogen #00-5523-00) according to the manufacturer's instructions. For staining of Granzyme B or IFNγ, BD Cytofix/Cytoperm Fixation/Permeabilization Kit was used (BD Bioscience #554714) according to the manufacturer's instructions. For the analysis of signaling intermediates, the cells were fixed using 2% formaldehyde followed by 90% methanol or 0.3% Triton X-100. Samples for intracellular staining were stained with indicated primary antibodies overnight at room temperature.

The samples were analyzed using an Aurora (Cytek), or FACSymphony (BD Bioscience). Cell sorting was done using a BD Influx or a FACSAria Ilu sorters (both BD Bioscience). FlowJo software (BD Bioscience) was used for the data analysis.

## Imaging flow cytometry analysis

Cells were fixed with 2% formaldehyde and permeabilized using 0.3% Triton-X in PBS. Samples were stained with indicated primary antibodies at room temperature overnight. The next day, samples were stained with goat anti-rabbit secondary antibody conjugated with AF488 for 45 min and measured on Amnis Imagestream X Mk II Imaging Flow Cytometer (Luminex). Data were analyzed using Ideas 6.2 software (Amnis). The nuclear translocation was calculated as Similarity feature (the nuclear probe and protein of interest channels) within a bright field channel-based mask (Fig. 3E) or using the built-in analysis wizard (Fig. 5C).

## Preparation of knockout cell lines

To generate knockout cell line CRISPR/Cas9 method, gene-targeted single-guided RNA (sgRNA) was designed using chopchop software (Labun et al, 2019). sgRNA was then inserted into pSpCas9(BB)-2A-GFP (PX458) vector (Ran et al, 2013) that was provided by Feng Zhang (Addgene plasmid #48138). List of sgRNA target sequences to prepare the ABIN1 knockout, PAM motif in bold:

Mouse ABIN1 KO_1: 5′ – GATCCAGCGGCTCAATAAGG**TGG**

Mouse ABIN1 KO_2: 5′ – GGTGAATTCTGCTCCTCAGT**AGG**

DO11.10 cells were transfected with PX458 vector containing the sgRNA using Lipofectamine 2000 (Invitrogen #52887) according to the manufacturer's instructions. GFP+ cells were sorted as single cells into 96-well plate using BD Influx sorter (BD Bioscience). Obtained clones were tested via immunoblotting for expression of ABIN1 and confirmed by sequencing.

## Production and testing of recombinant GITRL

The DNA coding sequences of tagged recombinant GITRL and OX40L were produced by GeneArt Gene Synthesis service (ThermoFisher Scientific). The sequences of GITRL and OX40L constructs contain: human CD33 signal peptide (AA 1-17, Uniprot: P20138), 6× His, 2× Strep, 1× Flag tag, and 47–173 AA of murine GITRL (Uniprot: Q7TS55) or 87–197 AA of murine OX40L (Uniprot: P43488). The constructs were inserted into pcDNA3.1 vector for subsequent production in HEK293FT cells. HEK293FT cells were transfected with the plasmid (30 µg) using polyethylenimine (75 µg) transfection. Supernatants were collected after 3 days post transfection and purified using GraviTrap TALON columns (GE Healthcare #GE29-0005-94). The columns were first equilibrated with purification buffer (50 mM sodium phosphate pH 7.4, 300 mM NaCl) before sample loading. Following, the columns were washed with 20 mM imidazole in purification buffer and eluted using 350 mM imidazole in the purification buffer. Samples were then loaded on Amicon Ultra-15 centrifugal filters (10 kDa molecular weight cutoff, Merck Millipore #UFC9010) for imidazole removal and washed with purification buffer. Samples were concentrated by centrifugation and mixed with glycerol 1:1 for long-term storage in −80 °C. To test the production, GITRL and OX40L were mixed with Laemmli sample buffer +/− 50 mM DTT and tested using WB and visualization with InstantBlue Coomassie protein stain (Expedeon #ISB1L). To confirm the ability of ligands to bind the receptor on T cells, non-activated and PMA/Iono pre-activated cells were treated with GITRL or OX40L in FACS buffer (PBS/2% FBS/0.1% azide). The binding of GITRL or OX40L was detected using anti-FLAG APC antibody and measured on Accuri C6 cytometer (BD Bioscience). The data were analyzed using FlowJo software (BD Bioscience).

## Immunoprecipitation of receptors proximal signaling complexes

T cells were isolated from C57Bl/6 J mice and incubated with PMA (5 ng/ml) and Ionomycin (0.2 µM) in complete RPMI (10% FBS/ 100 U/ml penicillin/100 µg/ml streptomycin/40 µg/ml gentamicin) for 3 days. Following, cell suspension was split for the negative control, "post-lysis" control, and the sample. Samples were incubated for 30 min at 37 °C in serum-free media and activated or not with GITRL or OX40L (500 ng/ml, the sample only). After the indicated time cells were lysed in 1% *n*-dodecyl-β-ᴅ-maltoside (DDM, Thermo Scientific #89903) in lysis buffer (30 mM Tris pH 7.4, 120 mM NaCl, 2 mM KCl, 2 mM EDTA, 10% glycerol, 10 mM chloroacetamide (Sigma #C0267), 10 mM cOmplete protease inhibitor cocktail (Roche #5056489001), and PhosSTOP tablets (Roche #4906837001)). Samples and controls were incubated at 4 °C for 30 min and cleared by centrifugation for 30 min, 2 °C, $21,130 \times g$. Samples were used for following immunoprecipitation or mixed with Laemli sample buffer, reduced by 50 mM DTT,

heated for 3 min at 92 °C, and analyzed by immunoblotting. After the activation step, to the "post-lysis" control, 1.6 µg of GITRL or OX40L was added to the cell lysate. The supernatant was collected and mixed with the anti-FLAG M2 beads (Sigma #A2220) and incubated overnight at 4 °C. Beads were then washed three times with 10× diluted lysis buffer containing 0.1% of DDM and eluted with Laemmli sample buffer, reduced by 50 mM DTT, heated for 3 min at 92 °C, and analyzed by immunoblotting.

For the tandem affinity purification of receptor proximal signaling complexes for MS, 3 µg of GITRL or OX40L were added to the control sample post lysis. For the first purification step, samples were mixed with anti-FLAG M2 beads and incubated at 4 °C overnight. Samples were then washed with lysis buffer containing 0.1% DDM and incubated with lysis buffer containing 1% DDM and 3×Flag peptide (100 µg/ml, Sigma #F4799) overnight for elution. Subsequently, the supernatant was collected and the elution step was repeated with 8 h incubation. The second purification step was done with 50 µl of Strep-Tactin beads (IBA Life-science #2-1201-010) overnight. After incubation, samples were washed 3× with lysis buffer containing 0.1% DDM and 1× with lysis buffer only. The elution step was done with MS Elution buffer (2% sodium deoxycholate in 50 mM Tris pH 8.5).

## Protein digestion and MS analysis

The eluted protein samples (200 µl) were reduced with 5 mM tris(2-carboxyethyl)phosphine at 60 °C for 60 min and alkylated with 10 mM methyl methanethiosulfonate at room temperature for 10 min. Proteins were cleaved overnight with 1 µg of trypsin (Promega) at 37 °C. To remove sodium deoxycholate, samples were acidified with 1% trifluoroacetic acid, mixed with an equal volume of ethyl acetate, centrifuged ($15,700 \times g$, 2 min), and an aqueous phase containing peptides was collected (Masuda et al, 2008). This step was repeated two more times. Peptides were desalted using in-house-made stage tips packed with C18 disks (Empore) (Rappsilber et al, 2007) and resuspended in 20 µl of 2% acetonitrile with 1% trifluoroacetic acid.

The digested protein samples (12 µl) were loaded onto the trap column (Acclaim PepMap300, C18, 5 µm, 300 Å Wide Pore, 300 µm × 5 mm) using 2% acetonitrile with 0.1% trifluoroacetic acid at a flow rate of 15 µl/min for 4 min. Subsequently, peptides were separated on a Nano Reversed-phase column (EASY-Spray column, 50 cm×75 µm internal diameter, packed with PepMap C18, 2-µm particles, 100 Å pore size) using a linear gradient from 4% to 35% acetonitrile containing 0.1% formic acid at a flow rate of 300 nl/min for 60 min.

Ionized peptides were analyzed on a Thermo Orbitrap Fusion (Q-OT-qIT, Thermo Scientific). Survey scans of peptide precursors from 350 to 1400 m/z were performed at 120 K resolution settings with a $4 \times 10^5$ ion count target. Four different types of tandem MS were performed according to precursor intensity. The first three types were detected in Ion trap in rapid mode, and the last one was detected in Orbitrap with 15000 resolution settings: (1) For precursors with intensity between $1 \times 10^3$ to $7 \times 10^3$ with CID fragmentation (35% collision energy) and 250 ms of ion injection time. (2) For ions with intensity in the range from $7 \times 10^3$ to $9 \times 10^4$ with CID fragmentation (35% collision energy) and 100 ms of ion injection time. (3) For ions with intensity in the range from $9 \times 10^4$ to $5 \times 10^6$ with HCD fragmentation (30% collision energy) and

100 ms of ion injection time. (4) For intensities $5 \times 10^6$ and more with HCD fragmentation (30% collision energy) and 35 ms of ion injection time. The dynamic exclusion duration was set to 60 s with a 10 ppm tolerance around the selected precursor and its isotopes. Monoisotopic precursor selection was turned on. The instrument was run in top speed mode with 3 s cycles.

All MS data were analyzed and quantified with the MaxQuant software (version 1.6.15.0) (Cox et al, 2014). The false discovery rate (FDR) was set to 1% for both proteins and peptides, and the minimum length was specified as seven amino acids. The Andromeda search engine was used for the MS/MS spectra search against the murine Swiss-Prot database (downloaded from Uniprot in December 2020). Trypsin specificity was set as C-terminal to Arg and Lys, also allowing the cleavage at proline bonds and a maximum of two missed cleavages. β-methylthiolation, N-terminal protein acetylation, carbamidomethylation, Met oxidation, and STY phosphorylation were included as variable modifications. Label-free quantification was performed using the Intensity Based Absolute Quantification (iBAQ) algorithm, which divides the sum of all precursor-peptide intensities by the number of theoretically observable peptides (Schwanhäusser et al, 2011). Data analysis was performed using Perseus 1.6.14.0 software (Tyanova et al, 2016).

## DO11.10 cell line stimulation

Cells were kept in serum-free media for 30 min prior activation. Following, cells were stimulated with GITRL (500 ng/ml) for indicated times. Subsequently, cells were lysed in 1% DDM in lysis buffer (30 mM Tris pH 7.4, 120 mM NaCl, 2 mM KCl, 2 mM EDTA, 10% glycerol, 10 mM chloroacetamide (Sigma #C0267), 10 mM cOmplete protease inhibitor cocktail (Roche #5056489001), and PhosSTOP tablets (Roche #4906837001)). Lysates were incubated at 4 °C for 30 min and cleared by centrifugation for 30 min, 2 °C, $21,130 \times g$. Samples were mixed with Laemmli sample buffer, reduced by 50 mM DTT, heated for 3 min at 92 °C, and analyzed by immunoblotting.

## T-cell stimulation

Primary T cells were isolated from the lymph nodes and/or spleen. The spleen was incubated in ACK buffer for 3 min in room temperature to remove red blood cells.

For the activation of T cells using T2-Kb, CD8$^+$ T cells were enriched using magnetic or FACS sorting. T2-Kb cells were stained in RPMI with DDAO dye (2.5 μM) for 15 min at 37 °C, 5% $CO_2$. Following, cells were resuspended in RPMI (10% FBS, 100 U/ml penicillin, 100 mg/ml streptomycin, 40 mg/ml gentamicin) and split into a 48-well plate and loaded with the indicated concentration of OVA peptide for 2 h at 37 °C, 5% $CO_2$. Following, T cells were incubated for 30 min in serum-free RPMI media. Cells were incubated for 2 h with OVA-loaded T2-Kb cells in 2:1 ratio. After activation, cells were fixed with formaldehyde (2%, Sigma #F8775) followed by permeabilization with 90% methanol on ice for flow cytometry or 0.3% Triton X-100 in FACS buffer (PBS, 5% goat serum, 2 mM EDTA, 0.1% sodium azide) for imaging flow cytometry.

To study the proliferation of T cells, cells were stained with Cell Trace Violet (1000× diluted) for 10 min at 37 °C, 5% $CO_2$. Cells were resuspended IMDM (10% FBS, 100 U/ml penicillin, 100 mg/ml

streptomycin, 40 mg/ml gentamicin) and split to a 48-well plate coated with indicated concentration of anti-CD3ε or Kb-OVA monomer. Cells were incubated for 72 h at 37 °C, 5% $CO_2$ and analyzed by flow cytometry.

For the analysis of expression of effector molecules, cells were (i) left untreated, (ii) treated with p38 inhibitor (12.5 μM), (iii) activated with anti-CD3/CD28 beads at beads to cells ratio 1:1, or (iv) activated with anti-CD3/CD28 beads (ThermoFisher #11453D) at 1:1 beads to cells ratio and treated with p38 inhibitor (12.5 μM). The incubation time was 16 h at 37 °C, 5% $CO_2$. After 12 h, Golgi stop was added (500×) for 4 h. Samples were fixed and permeabilized with BD Cytofix/Cytoperm Fixation/Permeabilization Kit (BD Bioscience #554714) according to the manufacturer's protocol. Samples were stained with indicated antibodies and analyzed by flow cytometry.

For the analysis of surface activation markers, CD8$^+$ cells were FACS-sorted from the lymph nodes. Cells were activated with anti-CD3ε/CD28 beads at indicated beads-to-cell ratios for 16 h. Samples were then stained with indicated antibodies and analyzed by flow cytometry.

For the analysis of GITR signaling, cells were activated with PMA (5 ng/ml) and ionomycin (0.5 μM) overnight. Following, cells were incubated in IMDM (10% FBS, 100 U/ml penicillin, 100 mg/ml streptomycin, 40 mg/ml gentamicin) supplemented with IL-2 for 72 h. Next, cells were activated with recombinant GITRL (500 ng/ml) for 15 min at 37 °C. Cells were fixed with 2% formaldehyde and permeabilized with 90% methanol on ice. Subsequently, the cells were processed for flow cytometry or imaging flow cytometry.

## Analysis of energy metabolism

The analysis was performed using Seahorse XF Pro Analyzer (Agilent) according to the manufacturer's instructions. Briefly, CD8$^+$ T cells from WT and Abin1$^{GTKO/GTKO}$ OT-I Rag2$^{KO/KO}$ mice were MACS-enriched. Cells were resuspended in Seahorse XF RPMI medium (pH 7.4, Agilent #103576) supplemented with 10 mM Seahorse XF 1.0 M glucose solution (Agilent #103577), 1 mM Seahorse XF 100 mM pyruvate solution (Agilent #103578), and 2 mM Seahorse XF 200 mM glutamine solution (Agilent #103579). In total, $2 \times 10^5$ cells per well were plated to poly-D-lysin (PDL)-coated 96-well microplate, 3–4 technical replicates per each condition. In some experiments, some replicates were excluded based on their unusual pattern of oxygen consumption (steep decrease from the beginning of the measurement). This was a pre-established criterion.

Cells were centrifuged 200 g, 1 min, light brake and plate were incubated in 37 °C, no $CO_2$ for minimum of 45 min. The Seahorse XF Glycolytic Rate Assay Kit (Agilent 103344) was used. For the activation of cells, anti-CD3ε (3 μg/ml), anti-CD28 (5 μg/ml) were used with or without addition of p38 inhibitor SB203580 (12.5 μM). Rotenon/Antimycin A and 2-deoxyglucose were part of the kit and injected at the indicated times. The data were analyzed using the Seahorse Wave Pro program (Agilent).

## In vitro suppression assay

CD4$^+$ and CD8$^+$ T cells were FACS-sorted from C57Bl/6J WT mice and labeled with the CTV dye. Tregs were FACS-sorted based on CD4$^+$GFP$^+$ expression from WT or Abin1$^{GTKO/GTKO}$ DEREG mice. A

48-well tissue culture plate was coated with anti-CD3ε antibody. Conventional cells were plated to a 48-well plate in complete IMDM (10% FBS, 100 U/ml penicillin, 100 mg/ml streptomycin, 40 mg/ml gentamicin) in 1:1 ratio with Tregs, one well with no Tregs was used as control. Cells were incubated at 37 °C, 5% $CO_2$ for 72 h, and measured by flow cytometry.

## Histology and immunofluorescence

Standard descriptive histopathology H&E staining was performed in the Czech Centre of Phenogenomics according to internal standard operating procedures. Briefly, tissue processing: automatic vacuum tissue processor (Leica ASP6025), embedding: Leica EG1150 H + C embedding station, sectioning: Leica RM2255 rotary microtome (2 μm sections), staining: Leica ST5020 + CV5030 stainer and coverslipper.

For the H&E staining of the pancreas, the samples were kept in acetone for 15 min and air-dry for 20 min. Stained with hematoxylin for 1–2 min and washed with water. Following, the samples were washed withed 0.1% hydrochloric acid for 10 s, rinsed with water and stained with eosin for 1 min. Again, washed with water, dehydrated with 96% ethanol for 10–15 min, followed with 1:1 isopropanol and 96% ethanol for 5 min, and xylene for 2 × 5 min. Samples were mounted in water-free solakryl medium. The pictures were acquired with ×40 magnification using Leica DM6000 microscope equipped with Leica DFC490 camera.

Immunofluorescent staining was done on 5 μm thick cryosections of the kidney, livers, and lungs, or on 10 μm thick cryosection of the pancreas from WT and *Abin1*<sup>GTKO/GTKO</sup> mice prepared using a cryostat (Leica CM1950). The sections on microscopy slides were fixed with 4% paraformaldehyde for 10 min and permeabilized with 0.1% Triton X-100 in PBS for 10 min. The samples were then blocked with 5% goat serum in PBS. Next, samples were stained with primary antibodies (CD4 AF647, CD8α, CD45.2 AF488, Insulin) diluted in 1% BSA in PBS at room temperature for 1 h, followed by staining with the secondary antibody (goat anti-guiny pig AF488 and/or goat anti-rabbit AF555) for 1 h, and nuclear staining with DAPI at room temperature for 15 min. Samples were mounted with Prolong Gold antifade mountant (Invitrogen #P36930) and images were acquired on confocal microscope Leica TCS SP8 (Leica Microsystems, objective HC PL APO 40×/1.30 OIL CS2).

## Bulk RNAseq

In total, $1 \times 10^6$ CD8<sup>+</sup> cells were FACS-sorted from lymph nodes and spleen of WT or *Abin1*<sup>GTKO/GTKO</sup> OT-I littermates per sample. Non-activated controls were directly used for RNA isolation. The remaining cells were activated with anti-CD3/CD28 (ThermoFisher #11453D) beads at indicated beads-to-cells ratios for 16 h prior to the RNA isolation. In some experiments, the samples were treated with p38 inhibitor SB203580 (12.5 μM) for the whole activation period. The RNA was isolated using RNA Clean & Concentrator-5 kit (Zymoresearch #R1014) according to the manufacturer's instruction. The cDNA libraries were prepared using KAPA mRNA Hyperprep Kit (Roche #KK8580) according to the manufacturer's instructions. The single-end sequencing was performed on Illumina NextSeq 500 using NextSeq 500/550 High Output Kit v2.5 (75 cycles) (Illumina #20024906) with final reads having a length of

76 bp. The base calling was performed using Illumina BaseSpace GenerateFASTQ workflow v1.37.0.

## Data analysis

The quality of reads was corrected for batch effects using package limma 3.52.2 (Ritchie et al, 2015). The genes upregulated in the *Abin1*<sup>GTKO/GTKO</sup> cells were selected as genes with fold-change over 1.5 in the activated cells from the *Abin1*<sup>GTKO/GTKO</sup> mice vs. the activated cells from the *Abin1*<sup>WT/WT</sup> mice (both without the p38 inhibitor) in each of the two experiments that were performed. The GSEA was then performed on these selected genes using R package fgsea 1.22.0 (Subramanian et al, 2005). The genes upregulated in the *Abin1*<sup>GTKO/GTKO</sup> cells were selected as genes with the average upregulation in the cells from the *Abin1*<sup>GTKO/GTKO</sup> mice compared to the cells from the *Abin1*<sup>WT/WT</sup> mice in each of the two experiments that were performed. The GSEA was then performed on these selected genes using R package fgsea 1.22.0 (Sergushichev, 2016). GSEA analyses focused on the regulation of NF-κB target genes used lists of NF-κB responsive genes with different kinetics as described in a study by Zhao et al (Zhao et al, 2023).

## Autoimmune diabetes

The model of autoimmune diabetes was described previously (Tsyklauri et al, 2023). Briefly, bone marrow-derived dendritic cells (BMDCs) derived from the bone marrow of congenic Ly5.1 C57BL/6J mice were cultured in IMDM media supplemented with 10% FBS (GIBCO), 100 U/ml penicillin (BB Pharma), 100 mg/ml streptomycin (Sigma-Aldrich), 40 mg/ml gentamicin (Sandoz), and with 2% of supernatant from J558 cells producing GM-CSF for 10 days at 37 °C, 5% $CO_2$ for 10 days (Kralova et al, 2018) and finally matured with LPS (25 μg) and loaded with 50 μg OVA peptide (SIINFEKL) for 3 h. Indicated numbers of T cells isolated from WT or *Abin1*<sup>GTKO/GTKO</sup> OT-I *Rag2*<sup>KO/KO</sup> mice were adoptively transferred to RIP.OVA mice intravenously. On the following day, $1 \times 10^6$ OVA-loaded BMDCs were injected intravenously. The glucose in urine was monitored with test strips (GLUKOPHAN, Erba Lachema) on daily basis for 2 weeks. The blood glucose was measured using Contour blood glucose meter (Bayer) on day 7 post immunization. The animal was considered as diabetic when the concentration of glucose in the urine was higher than 1000 mg/dl for 2 consecutive days.

## Tumor infiltration model

Overall, $5 \times 10^5$ MC-38 cells expressing OVA (Horkova et al, 2023) were injected subcutaneously to Ly5.1/Ly5.2 heterozygous mice. After 9 days, OT-I T cells were isolated from Ly5.1 mouse and mixed with Ly5.2 OT-I T cells isolated from the *Abin1*<sup>GTKO/GTKO</sup> mice or their WT littermates at 1:1 ratio. A total number of $2 \times 10^6$ OT-I T cells was injected intravenously. One week after OT-I T-cell injection, cells from tumor, dLN, ndLN, and spleen were isolated. For isolation of cells from tumor, tumors were cut into small pieces and incubated with Liberase (100 μg/ml, Roche #5401020001) and DNAse I (50 μg/ml, Roche #101104159001) in wash buffer (1% BSA and 1 mM EDTA in HBSS without $Ca^{2+}/Mg^{2+}$) at 37 °C, 350 RPM shaking for 45 min. Every 10 min, the samples were resuspended with a wide-bore pipette tip (1 ml). After the

enzymatic digestion, samples were filtered through 100-μm strainer and centrifuged $350 \times g$ at $4\,°C$ for 3 min. The pellet was resuspended in 5 ml of 40% Percoll (Cytiva #17089101) in DMEM. To the bottom of the tube, 5 ml of 80% Percoll was added to create a gradient. Samples were spun $320 \times g$, at room temperature acc/dec 0, for 23 min. The interphase that contained the lymphocytes was collected and centrifuged $400 \times g$, $4\,°C$ for 5 min. Following, the samples were analyzed by flow cytometry.

## The assessment of the anti-tumor response

In all, $5 \times 10^5$ MC-38 cells expressing OVA were injected subcutaneously to the CD3ε$^{KO/KO}$ host mice. Indicated numbers of OT-I T cells from WT or $Abin1^{GTKO/GTKO}$ were adoptively transferred intravenously. Tumor growth was monitored till day 23 after OT-I T cells injection using caliper. Tumor volume was calculated using the formula $V = (L \times S^2)/2$ (L is the longest diameter, S is the shortest diameter). When the tumor reached the endpoint volume of $500\ mm^3$, the mouse was sacrificed.

## Listeria and LCMV

Ly5.1 or Ly5.1/Ly5.2 heterozygotes were used as host mice. At day −1, sorted OT-I T cells from WT or $Abin1^{GTKO/GTKO}$ mice were adoptively transferred, $5 \times 10^4$ or $10 \times 10^4$ per mouse, depending on the experimental setup. Following day 0, host mice were infected intravenously with 5000 CFU of Lm-OVA or Lm-Q4H7 (Zehn et al, 2009) or $2–3 \times 10^5$ PFU of LCMV-OVA (provided by D. Pinschewer, University Hospital Basel) intraperitoneally, respectively. On day 6 (Lm) or 5 (LCMV), splenocytes were analyzed by flow cytometry.

## Bone marrow chimeras

Bone marrow cells were isolated from femurs of Ly5.1/Ly5.2 WT or Ly5.2 $Abin1^{GTKO/GTKO}$ mice. Cells were mixed in 1:1 ratio and total number of $10^6$ cells was injected into lethally irradiated (6 Gy, X-RAD 225 XL) congenic Ly5.1 mice intravenously. The splenocytes and lymph node cells of these mice were analyzed after 8 weeks by flow cytometry.

## Statistical analysis

The statistical significance was calculated using tests indicated in Figure Legends using GraphPad Prism 5.0 or R. We preferentially used nonparametric tests. In exceptional cases, the sample size was too small to perform nonparametric tests or to perform testing for normality. In these cases, we assumed normal distribution and calculated the statistical significance using paired or unpaired $t$ test. In the latter case, we assumed equal variance.

## Data availability

The mass spectrometry proteomics data have been deposited to the ProteomeXchange Consortium via the PRIDE (Perez-Riverol et al, 2022) under accession number PXD046422. The raw sequences as well as generated raw count matrices are available at GEO NCBI under accession number GSE245397 (https://www.ncbi.nlm.nih.gov/geo/query/acc.cgi?acc=GSM7841985). The code used to generate counts, downstream

analyses and relevant figures for this publication are available at https://github.com/Lab-of-Adaptive-Immunity/ABIN_KO_project. Source data (FCS files) for Figs. 2B, E, F, 3C–E, and 6A,B are deposited at https://www.ebi.ac.uk/biostudies/studies/S-BSST1400.

The source data of this paper are collected in the following database record: biostudies:S-SCDT-10_1038-S44319-024-00179-6.

## Peer review information

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

## Acknowledgements

This project was supported by the Czech Science Foundation (22-18046 S to OS), project National Institute of Virology and Bacteriology (Programme EXCELES, LX22NPO5103 to OS)—funded by the European Union—Next Generation EU, EMBO Installation Grant (4420 to PD), Charles University (PRIMUS/20/MED/003 to PD, Charles University Grant Agency: 984120 to SJ, 274323 to VU), European Union's Horizon 2020 research and innovation programme under grant agreement No. 802878 (ERC Starting Grant FunDiT to OS), Czech Ministry of Education, Youth and Sports and the European Regional Development Fund (OP RDI CZ.1.05/2.1.00/19.0395 and LM2023036 for the Czech Centre for Phenogenomics, OP RDI BIOCEV CZ.1.05/1.1.00/02.0109), and core funding provided by the Institute of Molecular Genetics of the Czech Academy of Sciences (RVO 68378050). The authors acknowledge Ladislav Cupak for technical assistance, Dagmar Zudova (Czech Centre for Phenogenomics) for the histological analysis, Karel Harant (Proteomic facility, Charles University in Prague) for proteomic analysis, Zdenek Cimburek and Matyas Sima (flow cytometry facility, IMG) for cell sorting, and the Light Microscopy Core Facility, Institute of Molecular Genetics of the Czech Academy of Sciences, supported by Czech Ministry of Education, Youth, and Sports (LM2023050 and RVO—68378050-KAV-NPUI), for their help. We thank our colleagues who shared material and reagents with us. ES is a student of the Faculty of Science, Charles University in Prague. The visual synopsis was created with BioRender.com.

## Author contributions

**Sarka Janusova**: Formal analysis; Investigation; Visualization; Methodology; Writing—original draft; Writing—review and editing. **Darina Paprckova**: Investigation; Methodology; Writing—review and editing. **Juraj Michalik**: Data curation; Software; Formal analysis; Writing—review and editing. **Valeria Uleri**: Investigation; Writing—review and editing. **Ales Drobek**: Investigation; Writing—review and editing. **Eva Salyova**: Investigation; Writing—review and editing. **Louise Chorfi**: Investigation; Writing—review and editing. **Ales Neuwirth**: Investigation; Methodology; Writing—review and editing. **Arina Andreyeva**: Investigation; Visualization; Writing—review and editing. **Jan Prochazka**: Resources; Methodology; Writing—review and editing. **Radislav Sedlacek**: Resources; Supervision; Methodology; Writing—review and editing. **Peter Draber**: Conceptualization; Resources; Data curation; Formal analysis; Supervision; Funding acquisition; Investigation; Visualization; Methodology; Project administration; Writing—review and editing. **Ondrej Stepanek**: Conceptualization; Resources; Data curation; Formal analysis; Supervision; Funding acquisition; Visualization; Methodology; Writing—original draft; Project administration; Writing—review and editing.

Source data underlying figure panels in this paper may have individual authorship assigned. Where available, figure panel/source data authorship is listed in the following database record: biostudies:S-SCDT-10_1038-S44319-024-00179-6.

## Disclosure and competing interests statement

The authors declare no competing interests.

# Expanded View Figures

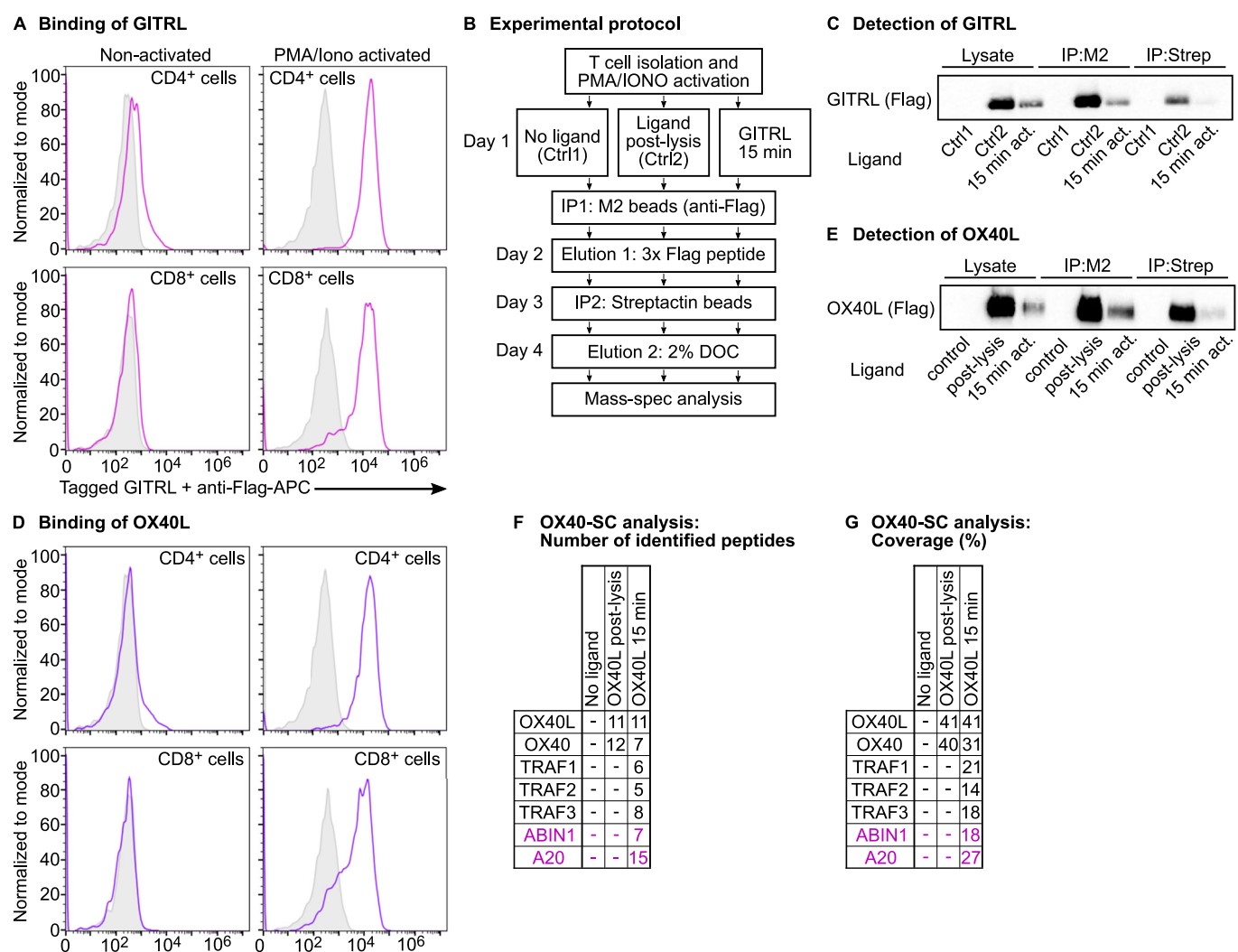

**Figure EV1. Analysis of the proximal GITR and OX40 signaling complex (SC).**

(A) Primary murine T cells were pre-activated with PMA/ionomycin for 72 h or not and stained with the anti-CD4 and anti-CD8 antibodies, recombinant GITRL followed by anti-FLAG antibody. A representative experiment out of 3 biological replicates in total. (B) Illustration of the protocol for the identification of the composition of GITR and OX40 signaling complexes. (C) Detection of the recombinant ligand in the lysate and after the first and second affinity purification step by immunoblotting. A representative experiment out of 3 biological replicates in total. (D) Primary murine T cells were pre-activated with PMA/ionomycin for 72 h or not and stained with the anti-CD4 and anti-CD8 antibodies, recombinant OX40 followed by anti-FLAG antibody. A representative experiment out of 3 biological replicates in total. (E–G) Primary murine T cells were pre-activated with PMA/ionomycin for 72 h and stimulated with the recombinant OX40L for 15 min. The cells were lysed and the OX40-SC was isolated via tandem affinity purification and samples were analyzed by mass spectrometry. (E) Detection of the recombinant ligand in the lysate and after the first and second affinity purification step by immunoblotting. Results of the protein identification are shown as the number of peptides (F) and the coverage (G). A single experiment was performed.

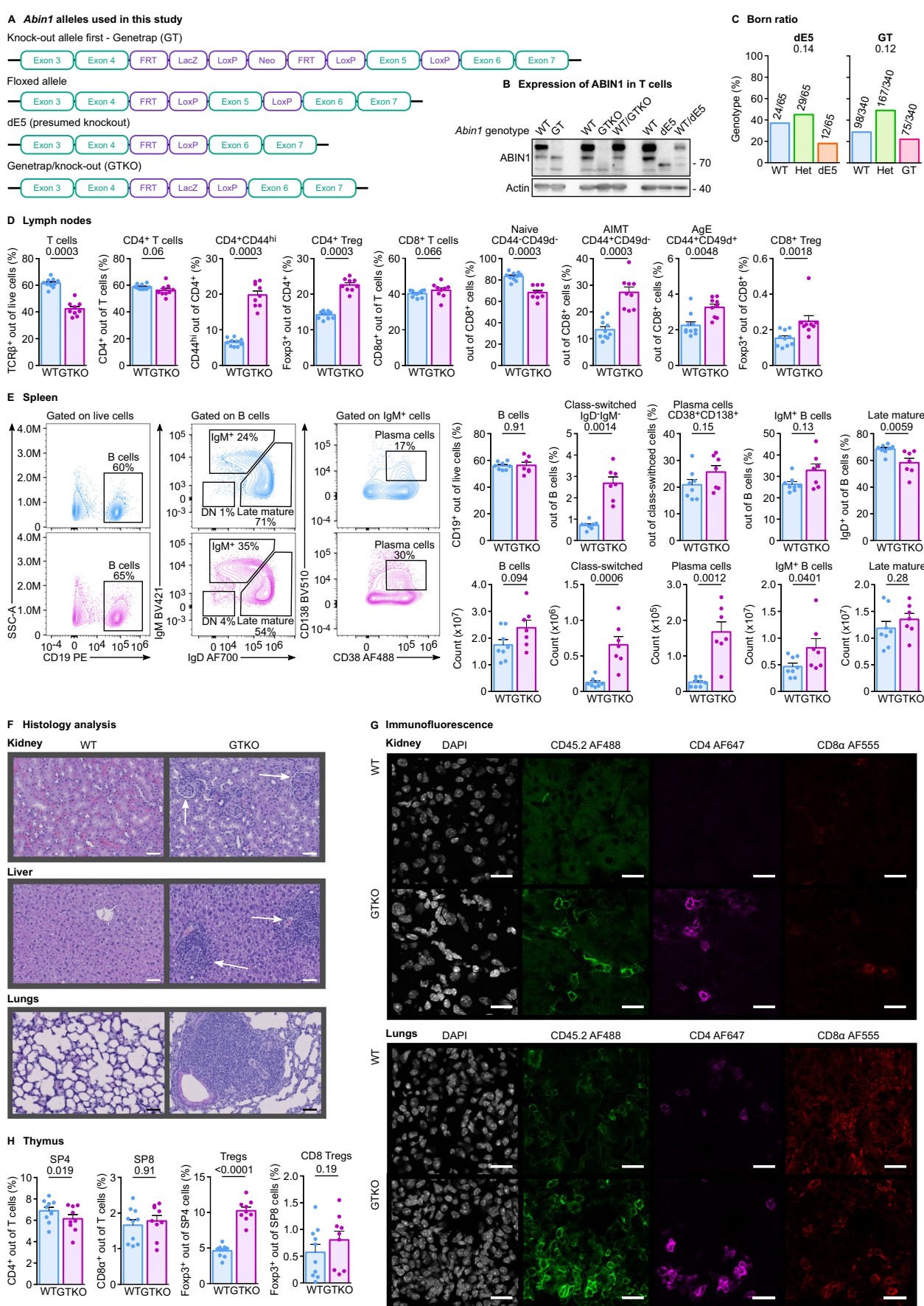

◀ **Figure EV2. Characterization of Abin1^{GT/GT} (GT), Abin1^{dE5/dE5} (dE5), and Abin1^{GTKO/GTKO} (GTKO) mice.**

(A) An overview of *Abin1* alleles used in this study. (B) Immunoblot analysis of ABIN1 in GT, GTKO, dE5 mice and corresponding littermate controls. A representative experiment is shown. The experiment was performed in 5 (GT), 3 (GTKO), or 2 (dE5) biological replicates. (C) Heterozygous *Abin1*^{WT/dE5} or *Abin1*^{WT/GT}, respectively, were bred and the genotype of the offspring was determined upon weaning. The frequencies and numbers of pups with particular genotypes are indicated. $n = 340$ (GT) or 65 (dE5) offspring mice per group in total from 16 (GT) or 2 (dE5) breedings. (D) Lymph node cells were stained with indicated antibodies and analyzed by flow cytometry. Aggregate results of the abundance of indicated subsets are shown, $n = 10$ (WT) or 9 (GTKO) mice per group. (E) Splenocytes were stained with indicated antibodies and analyzed by flow cytometry. Representative dot plots and aggregate results of the frequency of indicated subsets are shown. $n = 10$ (WT) or 9 (GTKO) mice per group. (F) Histological analysis using H&E staining of indicated organs of 20–26 week-old mice. Arrows pointing to inflammatory foci. Representative staining out of 4 mice per group in total. (G) Cryosections of lungs and kidneys of WT and GTKO mice were stained with indicated antibodies and DAPI (nuclei) and analyzed by confocal fluorescence microscopy. Representative sections out of 4 mice per group in total. (H) Fixed and permeabilized thymocytes from WT and GTKO mice were stained with indicated antibodies and analyzed by flow cytometry. $n = 10$ (WT) or 9 (GTKO) mice per group. Data information: In (D, E, H), data are represented as mean + SEM and *p* values are indicated. In (F, G), the scale bars indicate 20 μm and 50 μm, respectively. Statistical significance was determined by a binomial test (A) or two-tailed Mann–Whitney test (D, E, H).

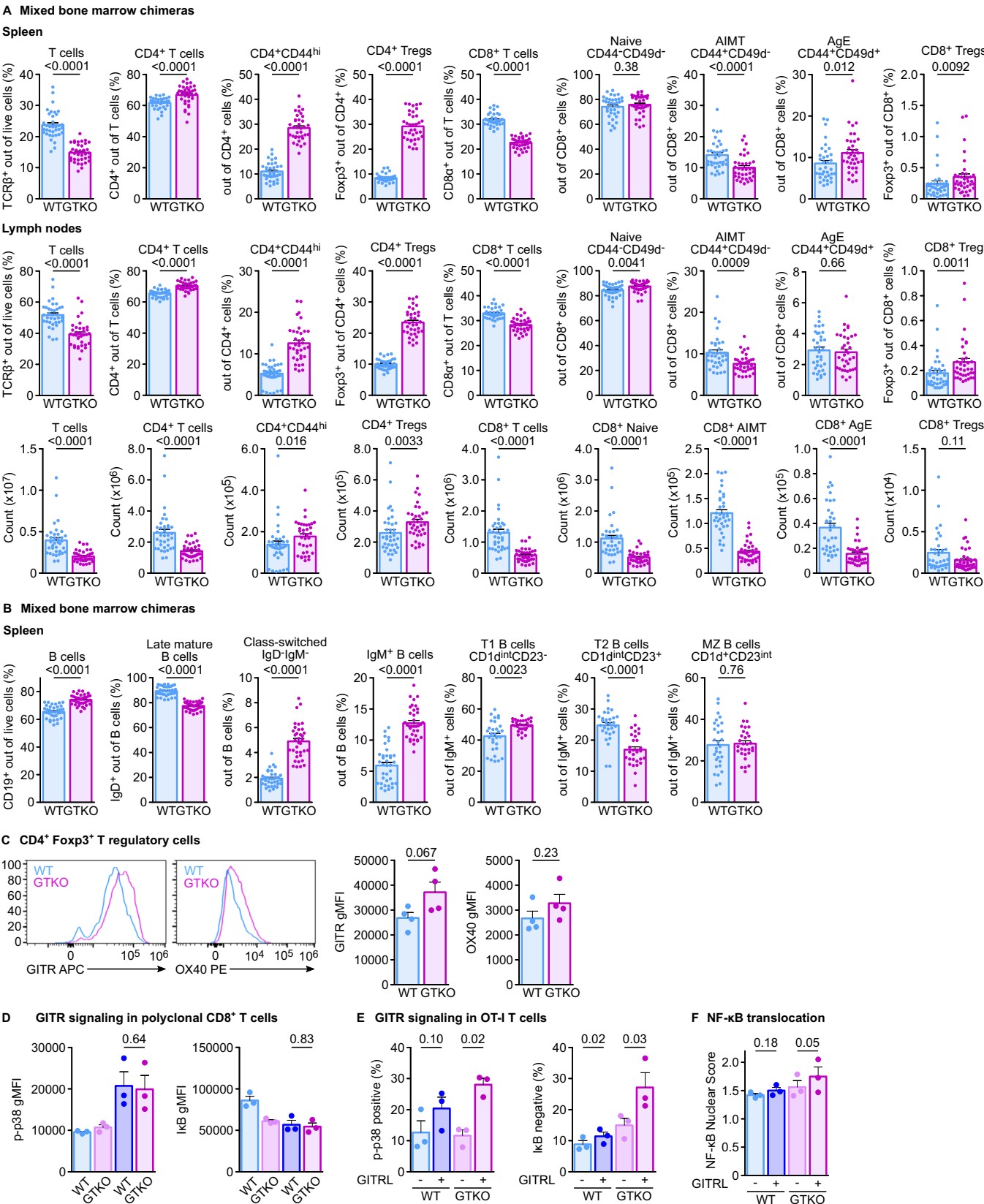

A  Mixed bone marrow chimeras

B  Mixed bone marrow chimeras

C  CD4⁺ Foxp3⁺ T regulatory cells

D  GITR signaling in polyclonal CD8⁺ T cells

E  GITR signaling in OT-I T cells

F  NF-κB translocation

**Figure EV3.  Intrinsic roles of ABIN1 in T cells.**

(**A, B**) The experiment shown in Fig. 3A,B. Aggregate results of the frequency and absolute counts of indicated subsets of T cells (**A**) or B cells (**B**). $n = 36$ mice per group. (**C**) Lymph node cells from WT or *Abin1*$^{GTKO/GTKO}$ mice were analyzed by flow cytometry. The expression of GITR and OX40 on CD4$^+$ FOXP3$^+$ T cells is shown. Representative histograms and aggregate data are shown. $n = 4$ mice per group. (**D**) The experiment shown in Fig. 3D. Lymph node cells from WT or *Abin1*$^{GTKO/GTKO}$ mice were pre-activated with PMA/ionomycin and stimulated with GITRL or left untreated (controls). Indicated activation pathways were analyzed by flow cytometry. Aggregate results of phospho-p38 and IκB levels in CD8$^+$ T cells. $n = 4$ mice per group. (**E, F**) The experiment shown in Fig. 3E. Lymph node cells from WT or *Abin1*$^{GTKO/GTKO}$ OT-I *Rag2*$^{KO/KO}$ mice were pre-activated with PMA/ionomycin and stimulated with GITRL or left untreated and analyzed by flow cytometry (**E**) or flow imaging (**F**). $n = 3$ mice per group. (**E**) The Percentage of phospho-p38 and IκB positive cells is shown. (**F**) The nuclear translocation score for NF-κB is shown. Data information: In (**A–E**), the data are presented as mean + SEM and *P* values are indicated. Statistical significance was determined by two-tailed Mann–Whitney test (**A, B**) or two-tailed Student's *t* test (**C, D**) or one-tailed paired *t* test (**E, F**).

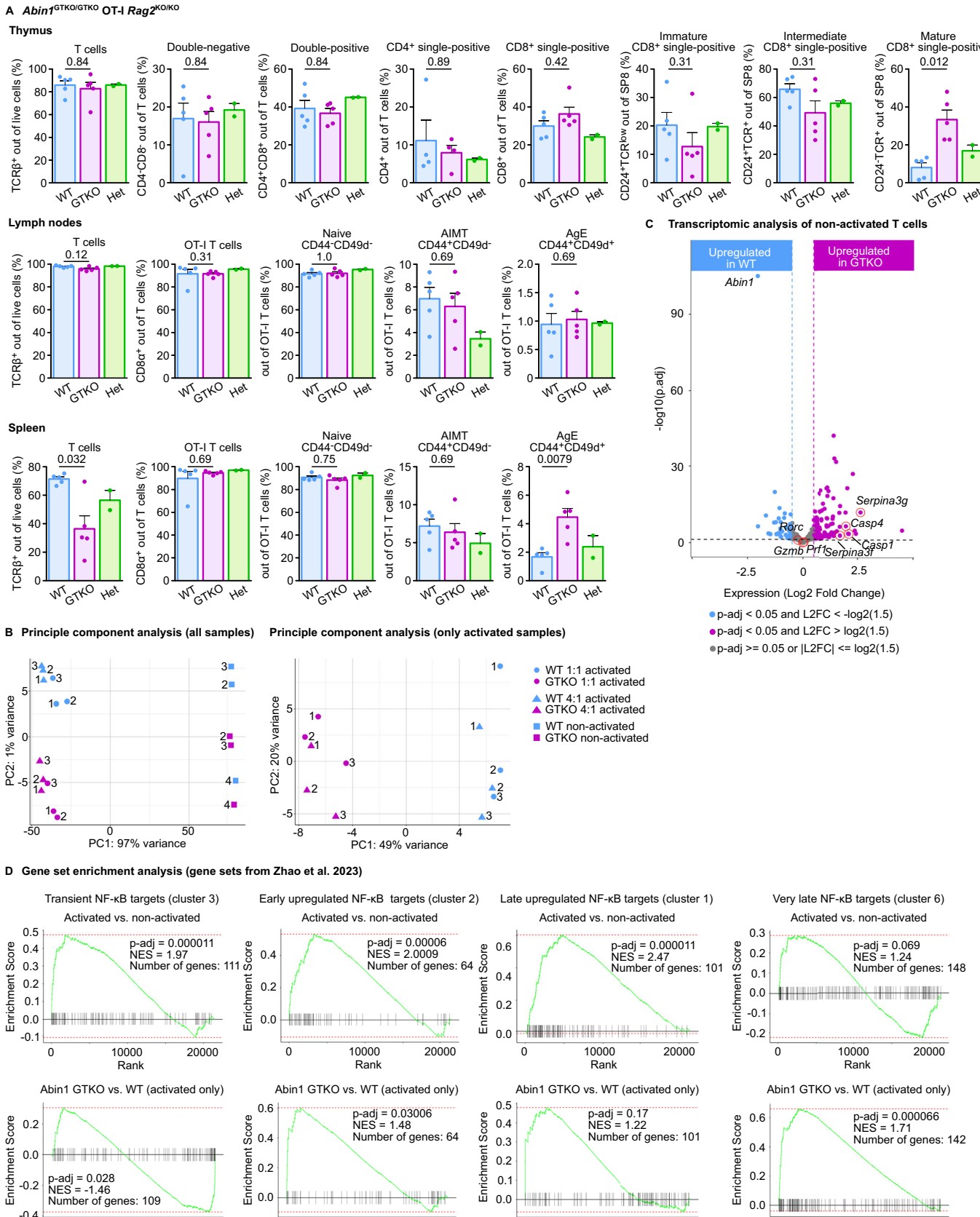

**Figure EV4. Characterization of *Abin1*<sup>GTKO/GTKO</sup> OT-I mice.**

(A) Cells from lymph nodes, spleen, and thymus from *Abin1*<sup>WT/WT</sup> OT-I *Rag2*<sup>KO/KO</sup> (WT), *Abin1*<sup>GTKO/GTKO</sup> OT-I *Rag2*<sup>KO/KO</sup> (GTKO), and *Abin1*<sup>WT/GTKO</sup> OT-I *Rag2*<sup>KO/KO</sup> (HET) were stained with indicated antibodies and analyzed by flow cytometry. Aggregate results of the frequency of indicated subsets are shown. $n = 5$ (WT and GTKO) or 2 (HET) mice per group. (B) Principal component analysis of the RNAseq experiment shown in Fig. 4C,D. (C) A volcano plot showing the differences in gene expression between non-activated WT and GTKO T cells from the experiment shown in Fig. 4C,D. (D) Four clusters of NF-κB responsive genes with distinct expression kinetics after the antigenic signaling were taken from a study by Zhao et al (Zhao et al, 2023). Using these gene lists, we performed a gene set enrichment analysis for the contrast between activated vs. non-activated OT-I T cells (upper lane) and activated GTKO vs. activated WT OT-I T cells (bottom lane) using the RNAseq data shown in Fig. 4C,D. Data information: In (A), the data are presented as mean + SEM. The statistical significance was calculated using two-tailed Mann–Whitney test (A), Wald test with Benjamini–Hochberg multiple testing correction (C), or Weighted Kolmogorov–Smirnov test (D).

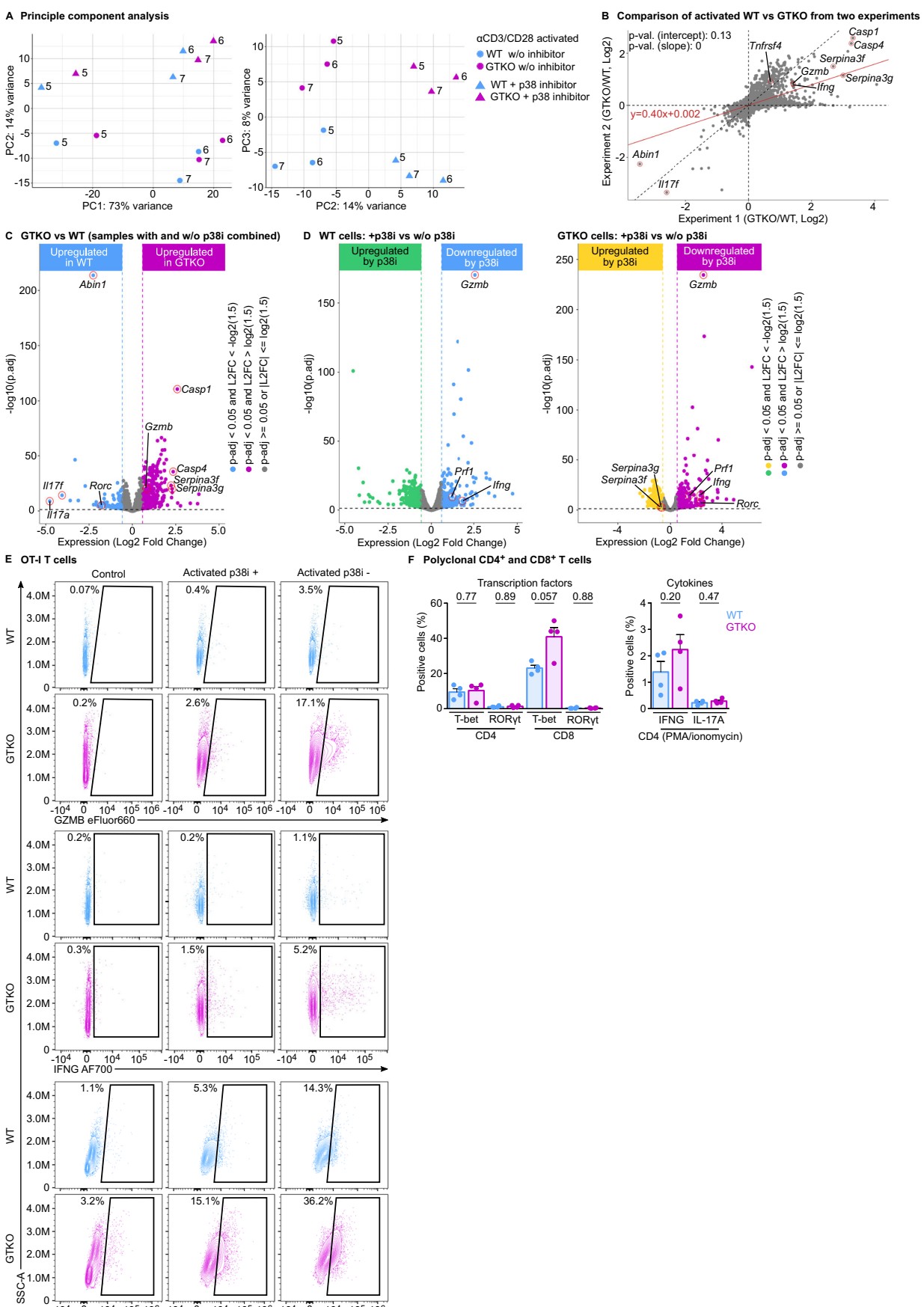

◄  **Figure EV5.  Analysis of signaling pathways in ABIN1-deficient T cells.**

(**A–D**) The same RNAseq experiment as presented in Fig. 5E,F. $n = 3$ biological replicates. (**A**) Principal component analysis of the samples. (**B**) Comparison of these experiments and the previous set of experiments (presented in Fig. 4C,D) by plotting the fold changes of activated *Abin1*<sup>GTKO/GTKO</sup> OT-I *Rag2*<sup>KO/KO</sup> (GTKO) vs. *Abin1*<sup>WT/WT</sup> OT-I *Rag2*<sup>KO/KO</sup> (WT) T cells in both sets of experiments. (**C**) A volcano plot showing expression changes of activated T cells treated p38 MAPK inhibitor (12.5 µM) for combined WT and GTKO samples. (**D**) A volcano plot showing expression changes of activated T cells treated or non-treated with p38 MAPK inhibitor (12.5 µM) for WT and GTKO samples separately. (**E**) Lymph node cells from WT and GTKO OT-I mice from the experiment shown in Fig. 5G. Representative dot plots showing the expression of indicated markers are shown. $n = 7$ (WT), or 8 (GTKO) mice per group. (**F**) Lymph node cells from WT or GTKO polyclonal mice were analyzed by flow cytometry for the expression of indicated transcription factors (left) or activated with PMA/ionomycin for 4 h and analyzed by flow cytometry for the production of indicated cytokines (right). $n = 4$ mice per group. Data information: In (**F**), data are presented as mean + SEM. The statistical analysis was calculated using two-tailed Wald test with Benjamini–Hochberg multiple testing correction (**C, D**) or two-tailed Student's *t* test (**F**).

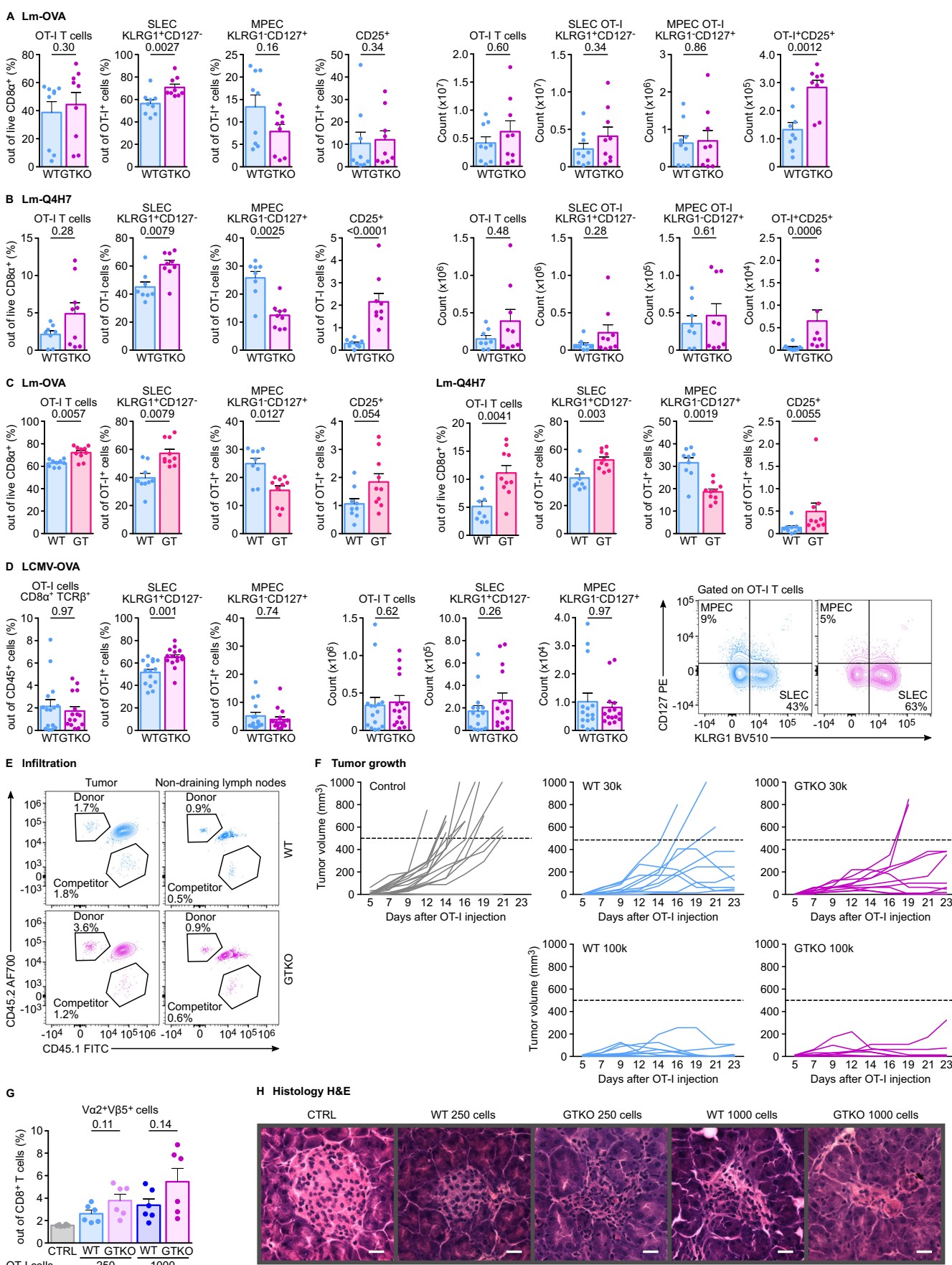

**Figure EV6. ABIN1 regulates T-cell responses in vivo.**

(A, B) The experiments as shown in Fig. 6A,B with identical protocol, but using a transfer of $5 \times 10^4$ T cells. (A) $n = 9$ mice per group. (B) $n = 8$ (WT), or 9 (GTKO) mice per group. Quantified frequencies and counts of indicated subsets of donor cells are shown. (C) $1 \times 10^5$ OT-I cells from WT or *Abin1*$^{GT/GT}$ OT-I *Rag2*$^{KO/KO}$ (GT) mice were adoptively transferred to Ly5.2 hosts that were infected with Lm-OVA. Splenocytes were analyzed by flow cytometry on day 6 post infection. The frequency of indicated subsets is shown. $n = 9$ (WT) or 10 (GT) mice per group. (D) $1 \times 10^4$ OT-I cells from WT or *Abin1*$^{GTKO/GTKO}$ OT-I *Rag2*$^{KO/KO}$ mice were adoptively transferred to the *CD3ε*$^{KO/KO}$ hosts that were infected with LCMV expressing OVA peptide. Splenocytes were analyzed by flow cytometry on day 5 post infection. Representative dot plots and quantified frequencies and absolute counts of indicated subsets of donor cells are shown. $n = 15$ mice per group. (E) The experiment shown in Fig. 6C. A representative experiment for tumor and non-draining lymph nodes. (F) The experiment shown in Fig. 6D. The tumor growth in individual mice is shown. The dashed line represents the endpoint of the experiment (tumor volume 500 mm³). The genotype and number of transferred OT-I *Rag2*$^{KO/KO}$ T cells are indicated. (G) In the experimental autoimmune assay (Fig. 6F), splenocytes of the host mice were analyzed by flow cytometry on day 6. The frequency of Vα2$^+$ Vβ5$^+$ cells among all CD8$^+$ T cells was quantified as a proxy for donor OT-I T cells. $n = 6$ mice per group. (H) Pancreatic cryosections (same experiments as shown in Fig. 6I) were stained with hematoxylin and eosin and imaged by microscopy. Data information: In (A–D, G), data are presented as mean + SEM. In (H), the scale bars represent 20 μm. The statistical significance was calculated using two-tailed Mann–Whitney test (A–D) or two-tailed Student's *t* test (G).

