## [Peer Review File · EMBO Reports]

ABIN1 is a negative regulator of effector functions in cytotoxic T cells

Sarka Janusova, Darina Paprckova, Juraj Michalik, Valeria Uleri, Ales Drobek, Eva Salyova, Louise Chorfi, Ales Neuwirth, Jan Prochazka, Radislav Sedlacek, Peter Draber, Arina Andreyeva, and Ondrej Stepanek

Corresponding author(s): Ondrej Stepanek (ondrej.stepanek@img.cas.cz)

Review Timeline:

Transfer Date:	2nd Dec 23
Editorial Decision:	13th Dec 23
Revision Received:	21st Apr 24
Editorial Decision:	15th May 24
Revision Received:	31st May 24
Accepted:	6th Jun 24

Editor: Achim Breiling

Transaction Report: This manuscript was transferred to EMBO reports following peer review at The EMBO Journal.

Revision plan for EMBO Reports

We will address the Referees' comments by revising the figures and text of the manuscript, by reanalyzing previous experiments, and by performing additional experiments, as specified in the point-by-point response.

We plan to do the following set of new experiments:

1. We will analyze the role of ABIN1 in the experimental diabetes model to analyze the expansion and phenotype of OT-I T cells in the spleen in this model by flow cytometry and the histological immunofluorescence analysis of CD8⁺ T-cell infiltration and insulin production in the pancreatic islets. We have already performed the analysis of WT and *Abin1* GTKO cells activated in OVA-loaded DCs, which is a part of the protocol in the experimental diabetes model. We observed increased expansion and short-lived KLGR1+ effector cell formation in the *Abin1* GTKO cells (shown below).

Short-lived effector T cells

Figure 1. Percentage of SLECs among WT and *Abin1* GTKO OT-I T cells activated by OVA-loaded DCs.

2. We will do additional analysis of polyclonal CD4 and CD8 T cells from the polyclonal WT and *Abin1* GTKO mice. We will analyze the cytokine production/expression of key transcription factors to address the role of ABIN1 in the Th1 vs. Th17 commitment in CD4 T cells. We will also address the basal phosphorylation of p38 in these cells, and assess the expression of CD44, GITR, and OX40 on Treg cells.

3. We will perform immunoblot analysis of freshly isolated T cells from WT and *Abin1* KO OT-I mice with the emphasis on phospho-p38 and other signaling intermediates.

4. We will address the role of ABIN1 in the co-stimulation pathways using GITRL stimulation of OTI CD8⁺ T cells. We believe that using monoclonal T cells from OT-I Rag2KO mice will largely normalize the differences between WT and GTKO and between individual mice. We will analyze mostly phospho-p38 and IKB by flow cytometry and nuclear NFKB translocation by Image Stream.

5. We will include an analysis of carbon metabolism in activated WT and GTKO OT-I T cells by Sea Horse analyzer. Our preliminary data show that ABIN1 down-regulates the glycolytic capacity of CD8 T cells and that this is at least partially mediated via p38 inhibition. We have already two experiments done and waiting for the third repetition. If our results will be reproduced, we include the results, which will contribute to the understanding of the role of ABIN1 and p38 in T-cell activation.

Point-by-point response

Referee #1:

The manuscript by Janusova and colleagues reports a negative regulator role of ABIN1 in T-cell antigen receptor (TCR) signaling in cytotoxic T cells. Yin, Krappman and colleagues have previously reported that A20 and ABIN1 cooperate to inhibit TCR signaling (Yin et al., Cell and Mol Life Sciences 2022). Deficiency of ABIN1 in primary human CD4⁺ T cells results in a hyperactive TCR with enhanced IL-2 and TNF α production. This previous work further demonstrates that ABIN1 deficiency results in enhanced NF κ B-activation in Jurkat T cells. The current study by Janusova and colleagues extends our current understanding of TCR signaling and ABIN1 functions in CD8⁺ T cells. However, the manuscript needs additional experimental validation to support their conclusions. Their data further suggest that are more complex functions of ABIN1 to both positively and negatively modulate TCR activation that require further exploration and dissection.

The authors conclude that ABIN1 negatively regulates TCR-activated p38 and NF κ B signaling pathways that, in turn, controls effector programs including IFN γ and GZMB. Supporting this hypothesis, ABIN1(KO) OT-1 CD8⁺ T cells demonstrate enhanced IFN γ and GZMB (Fig 4C-D). Conversely, the authors also demonstrate that ABIN1 positively regulates IL17, IL23R and RORC as ABIN1(KO) OT-1 CD8⁺ T cells exhibit decreased TCR-induced IL17 gene signatures. This is interesting, but demonstrates a more complicated model for ABIN1 in controlling potentially distinct effector arms of cytotoxic T cells than the authors' present model. This needs further exploration and discussion to better understand this potential dichotomous ABIN1 functions.

We are thankful to this Ref. for the evaluation of the manuscript and for their useful comments. We agree that the role of ABIN1 is complex and we will discuss it in the revised manuscript. Part of this complexity is probably caused by the ABIN1-mediated regulation of p38, which has ambiguous roles itself as discussed in the manuscript, but other signaling pathways probably also contribute to this.

As the scope of this study was primarily the role of ABIN1 in CD8⁺ T cells, we do not have proper TCR transgenic models to study the role of ABIN1 in the CD4⁺ T-cell lineage yet. We plan to add a more thorough analysis of polyclonal CD4⁺ T cells from the ABIN1 GTKO mice focusing on the Th lineage specification/cytokine production (see above).

Major concerns:

1. This complexity may confound the in vivo experimental data presented. Three in vivo systems are explored- (1). bacterial infection with a Listeria (OVA) model, (2) anti-tumor immunity with a syngeneic

MC38(OVA) model and (3) RIP.OVA experimental autoimmune diabetes (AID) model. However, data presented for each of the model lacks a complete functional and mechanistic understanding. For Listeria(OVA), while greater expansion of activated CD8+ T cells, there is no data reported for bacteria load. For the MC38(OVA) model, while there is greater tumor infiltration of ABIN1(KO) OT-1 CD8+ T cells, there is not greater anti-tumor immunity. The most promising in vivo phenotype is reported for the RIP1.OVA AID model with higher levels of blood glucose by day 7 and greater mortality of mice with ABIN1(KO) OT-1 CD8+ T cells. However, in this model, there is no immunologic characterization. It would be better to focus on one system with complete immunologic and phenotypic characterization to support the authors' conclusions.

We did not observe a large reproducible effect on Listeria clearance. However, the Listeria clearance is very rapid in WT mice and the contribution of transferred naïve cells is not often apparent (in contrast to the transfer of memory T cells). As mentioned above, we will focus on the diabetic model in more detail. We will analyze the expansion and phenotype of OT-I T cells in this model and we will analyze the pancreatic islets (CD8+ T cell infiltration and insulin production) similar to Tsyklauri et al. (PMID: 36705564, Figure 1E).

2. As ABIN1 also positively regulates IL-17, do ABIN1(KO) CD8+ T cells demonstrate an in vivo model (e.g., experimental autoimmune encephalitis or colitis)? These in vivo data would support the in vitro experimental observations of ABIN1(KO) CD8+ T cells.

Although ABIN1 might play a role in multiple disease conditions, it would be extremely difficult to study this and it would be definitely beyond the scope of our study. Using the conventional EAE/colitis models in polyclonal ABIN1 GTKO mouse would not lead to any conclusive results. ABIN1 has pleiotropic roles in many cell type, which would extrinsically influence the outcome of these models. For instance, we used chronic LCMV clone 13 for infecting WT and Abin1 GTKO mice, which turned out to be lethal to all Abin1 GTKO mice. Because uncovering the mechanism behind this lethality would be a whole project on its own and because of ethical reasons, we did not follow this further. Moreover, the major line of our study focuses on CD8+ T cells, whereas CD4+ T cells are mostly involved in the classical EAE and colitis models. We speculate that ABIN1 might inhibit the formation of Tc17, but this is rather a small unconventional population without a widely established autoimmune model. Overall, we believe that this would go beyond the scope of our manuscript.

3. The thymic development effect in GTKO mice appears impressive (EV4A- thymus). There appears to be a ~300% increase in CD8+ SP thymocytes which is substantial. Hence, the peripheral CD8+ T cells may well be different due to thymic development effects. This appears to be supported by the PC analysis of WT vs GTKO non-activated CD8+ T cells (EV4B). These baseline differences are not accounted for in subsequent TCR-activation experiments.

This Ref. is correct that there are clearly more mature SP8 thymocytes in Abin1 GTKO than WT OT-I Rag2KO mice. We believe that this is most likely caused by enhanced positive selection mediated by the lack of negative regulation of the TCR signal by ABIN1 (OT-I T cells are MHC-I-restricted and committed to differentiation in SP8 thymocytes). We think that the risk that this leads to substantially altered peripheral phenotype is relatively low and acceptable. We would definitely love to address this potential issue, but as the Abin1 flox allele did not work (Figure EV2B, Appendix Figure 3), we were not able to generate the inducible KO to get over this. We will discuss this in the revised version of the manuscript.

The PCA analysis (EV4B) does not actually show such a clear difference between non-activated WT and GTKO cells. The PCA clearly separates these two groups only after activation. However, we agree that some differences are there, such as a higher percentage (although still quite low) of CD49d⁺ CD44⁺ cells in the spleen (Figure EV4A) and the steady-state hyperphosphorylation of p38 (Figure 5A). We will address this by a more elaborate analysis of the gene expression in activated and non-activated WT and GTKO cells in the Extended View Figure 4A.

4. Figure 4B- Missing are statistical analysis of cell division comparisons between WT vs GTKO CD8⁺ OT-1 T cells.

We will include the statistical analysis.

5. Figure 5B- statistical analysis needs to be applied to this data set before concluding that the activated GTKO CD8⁺ T cells have elevated p38.

We will add the statistical analysis documenting that the difference in the activation by the low concentration of antigen is statistically significant between WT and ABIN1 GTKO.

6. Figure 5C- I am not quite following the authors' conclusion that this figure demonstrates augmented nuclear translocation of NFκB, but not NFAT. The statistical analysis shown in the figure that both NFκB and NFAT translocation are different for WT vs GTKO, though the data look quite similar.

We agree with this Ref. that the current representation of the data might be confusing. To fix this, we will separate the graph showing individual cells (will go to Extended View Figure) and the medians per experiment (main figure), which shows the reproducibility. We will show that only differences in NFκB, not NFκB, translocation were reproduced with statistical significance for the low concentration of the antigen. It should be noted that the difference in the WT and GTKO is relatively minor. The interpretation of these data is quite difficult, but it is what it is. We will revise the manuscript to tone down the message that NFκB is regulated by ABIN1 in T cells (which is anyway NOT the main line in the manuscript).

Minor concerns:

1. The description of dE5 construct is unnecessary for the crux of this study and adds confusion. As it turns out the dE5 construct did not result in a true knockout, but there is little understanding of how the residual low molecular weight ABIN1 protein arises. Further, there is no additional characterization of the dE5 knockout. I suggest taking all of this data out of the text and EV figure 2. The manuscript should just start with the true GTKO.

We defer this issue to the handling editor. We agree that we spend quite some time on this and it harms the flow of the text. However, we have two reasons, why to describe other genotypes in the Extended View Figures and their phenotypes in the Appendix Figures.

(I) We got the GT allele, which can be used for the generation of presumed whole-body and conditional KOs, from the Infrafrontiers consortium. We believe that other scientists potentially interested in this strain

would appreciate our data that it does not work the way it should. We can save them time and money. However, we do think that additional characterization of the dE5 KO mouse would improve our story in any way.

(II) If we start this part of the manuscript straight with the GTKO mice, the readers might get confused why we use mice with this allele, as this is rather unusual (the first choice would be to use the dE5 allele).

Overall, we would prefer keeping this as it is, only with a small revision of the text. What is the opinion of the handling editor?

2. The in vitro demonstration of enhanced p38 activation with GITRL in CD4+ T cells (figure 3D) is of unclear biological significance. There is no additional in vivo exploration of this in vitro observation.

We agree with this Ref. that there is not much follow-up on this, unfortunately. As ABIN1 apparently regulates both the co-stimulation and TCR signaling pathway, we believe that both pathways contribute to the in vivo phenotype of CD8+ T cells. However, it is not possible to separate the contribution of these particular pathways in vivo easily. This results with phospho-p38 is an important piece of puzzle supporting that ABIN1 is a negative regulator of the p38 pathway in T cells.

As mentioned above, we will perform additional characterization of polyclonal ABIN1-deficient CD4+ and CD8+ T cells, including their steady-state p38 phosphorylation. We will also perform additional experiments focusing on the role of ABIN1 in GITR signaling in CD8+ T cells using monoclonal OTI T cells.

3. Page 3, line 7- I believe that Fig. EV1A should be Figure 1A.

The reference to Fig. EV1A is correct – it shows that the ligand binds to the pre-activated T cells. However, we agree that we should refer to Fig. 1A as well in this sentence. We will include it in the revised version of the manuscript.

Referee #2:

In the present study, Janusova and colleagues identify the adaptor protein A20-binding inhibitor of NF- κ B 1 (ABIN1) as component of the signaling complexes of GITR and OX40, members of the T cell co-stimulatory superfamily. To achieve this, the authors generated a Flag-tagged recombinant GITR (and OX40) ligand to isolate its signaling complex and identify components thereof by mass spec. Different strategies were used to establish gene-targeted mouse strains lacking ABIN1 to address its function in T cells. One of these strains, Abin1 GTKO/GTKO, was further characterized and served to show that mice lacking ABIN1 possess elevated numbers of peripheral CD4 and CD8 T cells. Among the CD4 T cells, the frequency of activated and regulatory T cells was higher in the ABIN1-KO mice, whereas in the CD8 T cell fraction, CD44+ cells were more abundant. Interestingly, the authors found that ABIN1-deficiency resulted in elevated GITRL-mediated p38 MAPK signaling and a decreased basal NF- κ B activity in CD4 cells, indicative of a negative regulatory function of ABIN1 in T helper cells. Surprisingly, this was much weaker or not observed in CD8 T cells. It is unclear why and a pity that the authors solely further investigated the role of ABIN1 in CD8 but not in CD4 T cells. Consequently, *Listeria* (expressing the model antigen OVA) infection was similarly cleared by both WT and ABIN1-deficient OT-I CTLs, despite ABIN1-GTKO OT-I showed higher expansion and preferred differentiation into SLECs as compared to WT OT-I cells. A difference, however, was seen with a low affinity OVA peptide variant. Similarly, in a colon carcinoma model, ABIN1-GTKO OT-I cells more strongly infiltrated the tumor than WT OT-I cells but without possessing a better anti-tumor immune response. Notably, in a final auto-immune diabetes mouse model, ABIN1-deficient OT-I cells were much more prone in inducing an autoimmune pathology. This implies that ABIN1 owns critical regulatory functions in CD4 T cells, and Th17 and/or Treg cells in particular, rather than in CTLs. This however, has not been assessed. The potential importance of Th17 cells is supported by the performed RNA seq experiments, which indicate lower expression levels of IL-17/23 genes in ABIN1-deficient OT-I cells.

Overall, this is a very well executed, state-of-the-art, technically sound, rigorous and impressive study that deserves publication. The only critic I have is that CD4 T cells have not been included for in depth functional analysis. For this crossing ABIN1-GTKO e.g. with OI-II mice would have been required.

We are very thankful to the positive evaluation by this Ref. We agree that the analysis of the role of ABIN1 in CD4+ T cells would be very interesting and it can be a nice follow up to this study. We will perform an additional characterization of polyclonal CD4+ T cells from ABIN1 deficient mice during the revisions as mentioned above.

Referee #3:

Janusova et al identified ABIN1 (A20 binding inhibitor of NF- κ B1) as a component of the GITR and OX40 co-stimulatory complexes in T cells. To investigate the physiological role of ABIN1 in T cells, the authors generated Abin1 KO mice (GTKO) and performed immune phenotypic analyses. Using bone marrow chimeras, they elucidate lymphocyte-extrinsic and -intrinsic functions of ABIN1. While T cell populations were increased in ABIN1 GTKO, only CD44⁺ activated T cells and FOXP3⁺ Treg cells retained higher frequencies in a competitive setting with WT bone marrow cells, arguing for a cell intrinsic effect. In addition, enhanced isotype switching and plasma cell differentiation were largely cell intrinsic. The authors provide evidence for augmented p38 and NF- κ B signaling in ABIN1 GTKO in response to GITRL stimulation or CD3/CD28 co-stimulation. Titration experiments reveal that T cell activation markers (CD69, CD25, CD44) and proliferation are induced more strongly in ABIN1 KO T cells especially with weaker antigenic stimulations. Transcriptional profiling indicated stronger activation of pro-inflammatory genes such as IFNG and GSMB in ABIN1 KO T cells. The authors performed three in vivo models using adoptive transfer ABIN1 KO or WT OT-1 T cells. ABIN1 KO OT1 cells showed mildly increased cytotoxic responses in listeria and LCM infection models, enhanced OT-1 tumor infiltration in a MC38 syngeneic tumor model and stronger induction in a murine autoimmune diabetes model. Overall, the manuscript demonstrates for the first time a T cell intrinsic function of the NF- κ B negative regulator ABIN1.

We are thankful for the evaluation of our manuscript and for the useful comments by this Ref.

Major points:

The manuscript contains a lot of data and it is in parts very difficult to follow. Some data are highly relevant, some may be not quite as important.

We are pleased that this Ref. appreciates the amount of our data. We would like to take this opportunity to mention here that this Ref. and the Ref#1 ask us for quite large amount of additional data, some of seem to be beyond the main scope of our study (e.g., Tregs, characterization of CD4 T cells), which would make the manuscript even bulkier.

For instance, three ABIN1 KO strategies have been performed, but apparently only ABIN1 GTKO yielded a full KO without any truncated form. Why are data from the other KOs still included, especially because it is not clear to the reader what truncations of ABIN1 are retained in GT or dE5 KO?

This Ref has a similar comment as Ref#1. Our response is the same:

We defer this issue to the handling editor. We agree that we spend quite some time on this and it harms the flow of the text. However, we have two reasons, why to describe other genotypes in the Extended View Figures.

(I) We got the GT allele, which can be used for the generation of presumed whole-body and conditional KOs, from the Infrafrontiers consortium. We believe that other scientists potentially interested in this strain would appreciate our data that it does not work the way it should. We can save them time and money. However, we do think that additional characterization of the dE5 KO mouse would improve our story in any way.

(II) If we start straight with the GTKO mice, the readers might get confused why we use mice with this allele, as this is rather unusual (the first choice would be to use the dE5 allele).

Overall, we would prefer keeping it as it is, only with a small revision of the text. What is the opinion of the handling editor?

Further, unconditional ABIN1 KO mice have been published before and some immune phenotyping has been done. What is already known about the effects on lymphocytes population compared to previous ABIN1 KO mice (also in PMID: 22011580)? Have other strategies maybe failed to obtain a full KO? Better discussion and explanation is needed in light of previous findings. Is it really necessary to show ABIN1 GT and dE5 KOs? If so, please indicate what truncated variants may still be expressed.

We agree that a thorough discussion of the lymphocyte compartment in different ABIN1-deficient mouse strains is missing. We will discuss our data and the data published by Philip Cohen (PMID: 21606507) and Zhou et al. (PMID: 22011580) concerning the analysis of lymphocyte populations in the revised manuscript in more detail. The mentioned study (Zhou et al., PMID: 22011580) generated a GT allele, with a GT insertion in the first intron of *Abin1*. Their documentation of this allele/mouse is brief. However, it seems plausible that their allele might still retain the higher MW isoform of ABIN1. The study does not show the whole immunoblot, just a tiny section corresponding to ABIN1 MW (without showing any MW standard). In this section, a double band is apparent and only the lower band is missing in their ABIN1 GT (see below, Figure 2 here). They labeled the band with higher apparent MW and higher intensity as non-specific signal without any evidence shown. We also see a double band of ABIN1 (see below, Figure 3 here). However, the analysis of our GT, GTKO, and dE5 mice clearly shows that both bands represent ABIN1. Accordingly, it is well established that ABIN1 has two isoforms in mice: “The two splice variants of the ABIN cDNA contained an open reading frame of 1,941 and 1,782 nucleotides respectively, initiating at two different methionines [ABIN (1-647) and ABIN (54-647)] (GenBank/EMBL/DDBJ accession numbers AJ242777 and AJ242778). These cDNAs encode proteins of 72 and 68 kD” (Heyninck et al. 1999, PMID: 10385526).

We could not find in the article by Zhou et al., which antibody they used for the detection of ABIN1 in these immunoblots.

Figure 2. Immunoblot of ABIN1 in the WT and GT mice from the study by Zhou et al. (their Figure 2C). Showing the upper band remaining.

Figure 3. Our results showing that both low- and high-MW isoforms of ABIN1 are missing in the GTKO mice (our Fig. EV2B).

Although we find it important to show that the dE5 allele is not a null allele, we do not find it so crucial to investigate the identity of the artificial truncated protein is expressed from this allele. Although it might provide additional insight into the structural function of ABIN1, it would require challenging and time-taking experiments going beyond the scope of this manuscript.

Studies on T cell signaling could be more rigorous to draw clear conclusions. I κ Ba protein amounts in single cell assays is difficult to interpret, because of the NF- κ B feedback regulation. Further, the single cell image stream assays in Fig. 6C is not convincing for showing relevant enhancement of NF- κ B activation in ABIN1 GTKO T cells. Freshly isolated CD4 T cells should be used to perform biochemical analyses by Western blotting (pI κ Ba, I κ Ba, p-p65, translocation of NF- κ B and/or EMSA) to demonstrate effects on TCR/CD28 and PMA/Iono stimulation. Also, activation of MAPKs can be monitored.

Perhaps this Ref. meant "Figure 5C" here. Anyway, we are not sure if we got the point of this comment. We do not see any advantage of immunoblotting over flow cytometry/image stream in the cases, where good validated antibodies are available (such as I κ B, NFKB). The immunoblotting is definitely less quantitative than FC/Image stream. Any feedback regulations of the signaling intermediates would manifest on the bulk level as well (bulk is nothing more than an average of single cell levels – we can have the average from the FC data as well). However, based on this comment and other comments, we decided to quantify the data in Fig. 5B as geometric mean fluorescence intensity, not as the percentage of positive cells as was in the previous version of the manuscript. Also, as stated above, we will change the presentation of the Figure 5C to show the median nuclear translocation from individual experiments in the main Figure and the individual cells in the supplemental material to make the graph less confusing. We hope that it will be more convincing. It is true that the difference between WT and GTKO, albeit statistically significant, is relatively small. The observed differences between WT and GTKO in the *in vivo* and *ex vivo* activation are larger than in the analysis of the signaling intermediates, perhaps because the signaling intermediates are transient and the cells are not perfectly synchronized. Overall, we believe that our data on the phosphorylation of p38 (Figure 3D, Figure 5A-B), effects of p38 inhibition (Figure 5D) and gene expression between untreated and p38 inhibited WT and GTKO T cells (Figure 5E-H) altogether convincingly show p38 is negatively regulated by ABIN1.

The data on NFKB are relatively weak, but as the role of ABIN1 in this pathways is heavily studied, we performed these experiments and show the data, which do not show striking, but still significant differences. We will clarify our standpoint in the revised version of the discussion.

Motivated by this comment. We reanalyzed our RNAseq data to see how NFKB/RelA genes are expressed. We took a set of 113 NFKB/RelA responsive genes in B lymphocytes identified in a recent study by Zhao et al. 2023 (PMID: 37524800). Using a GSEA analysis, we observed that these cells are upregulated in aCD3/aCD28 beads in our OTI T cells as expected (Figure 4 here).

Figure 4. GSEA showing the NFKB/RelA responsive genes are upregulated in activated T cells a set of 113 NFKB/RelA target genes.

Surprisingly, we observed that these genes are not overexpressed in the activated GTKO vs WT T cells. Actually, it seems that these genes are rather upregulated in the WT cells. See GSEA results from both experiments below (Figure 5 and 6 here).

Figure 5. The GSEA analysis of activated GTKO vs WT T cells using a set of 113 NFKB/RelA target genes. Experiment 1.

Figure 6. The GSEA analysis of activated GTKO vs WT T cells using a set of 113 NFKB/RelA target genes. Experiment 1.

These data indicate that the ABIN1 deficiency does not lead to the upregulation of NFKB target genes *ex vivo*. We even observed that TNF (not a part of the set of 113 genes mentioned above, perhaps because not expressed by B cells), is downregulated in ABIN1 GTKO T cells in these assays (Figure 7). We are thankful to this reviewer for motivating us to investigate this. We will include this analysis in the manuscript and adjust the conclusion of our manuscript in a way that ABIN1 might play a minor role in the negative regulation of TCR-mediated NFKB signaling, but we did not find a strong evidence for it and that the transcriptional analysis even shows rather the opposite. However, we would like to stress that the major focus of our study is the overall role of ABIN1 in T-cell activation *in vivo* and *ex vivo* and on the p38 pathway, which was identified as a major pathway negatively regulated by ABIN1 in T cells.

Figure 7. The expression of TNF is triggered by activation, but less in ABIN1 GTKO than in WT T cells. Analysis of RNAseq data from both experiments.

We are not sure, why this Ref suggests to use freshly isolated CD4 T cells here, since the major subject of this study is CD8⁺ T cells. However, we will strengthen the manuscript by the analysis of steady-state p38 phosphorylation and other signaling intermediates in freshly isolated and fixed polyclonal CD8⁺ and CD4⁺ T cells using flow cytometry to complement the results in Fig. 5A, and by performing immunoblotting of the steady state OT-I T cells to confirm the hyperphosphorylation of p38 (which seems to be the key pathway regulated by ABIN1) by another method.

Along the same line, the authors start by identifying ABIN1 in the context of the GITR and OX40 co-receptor complexes. With the exception of Fig. 6D, all subsequent analyses on signaling pathways were performed using CD3, CD3/CD28 or OVA stimulation. Given the opening of the manuscript, bit stronger evidence would be helpful, if GITR and OX40 signaling are controlled by ABIN1. Is it possible to monitor NF-kB or MAPK translocation in response to GITR or OX40 stimulation using image stream? Is PMA/Iono-induced expression of GITR and OX40 changed in ABIN1 GTKO CD4 T cells?

We will do additional signaling experiments using GITR ligands. We will use PMA/iono pre-activated WT and GTKO OT-I T cells to perform these experiment as stated above. The major advantages of this approach over the analysis of the mixed polyclonal population is that we normalize the baseline by eliminating several extrinsic factors which are present in the polyclonal mice. We will also include the NFKB translocation by the Image Steam.

GITR levels were comparable in CD8⁺ and CD4⁺ T cells prior to the GITRL activation. We will include these data in the revised manuscript. OX40 staining was also comparable, but we have only 2 experiments.

Specific points

Fig. 2B: In PMID 22011580, the authors show a significant reduction in CD4 and CD8 T cell numbers in the spleen. What is different to ABIN1 GTKO mice used in this study, which shows increased levels of CD4 and CD8 T cells?

Percentage-wise we also see fewer T cells in the ABIN1 GTKO. However, the overall numbers are slightly increased. This corresponds to ABIN1[D485N] knock-in mice (Nada et al. 2011, PMID: 21606507). In both these studies, quite normal thymic development in polyclonal mice is shown.

It is unclear, why the mentioned study by Zhou et al. shows so dramatic reduction of splenic T cells, although the overall splenocyte counts are significantly elevated. This study does not show thymic development or even lymphocytes from lymph nodes. They also do not show any data on lymphocytes from the bone marrow chimeras. Although not explicitly stated, this study probably used mice backcrossed to the C57BL/6J background for at least 5 generations, which is the same background we used. Potentially, this difference can be caused by different housing conditions (microbial colonization) or by the difference in the modification of the Abin1 allele (as discussed above).

As already mentioned, we will discuss this in the revised version of the manuscript.

Fig. 3A and text: The authors write that no enhanced expansion and differentiation of T cells was noted in the T cells of ABIN1 GTKO chimeras, but an increase in CD44^{hi} effector points to effects on differentiation. The authors should show effects on naive, effector-memory, central-memory like T cell populations.

This Ref. is correct that there are increased frequencies of CD44⁺ CD4⁺ T cells and marginally increased antigen-experienced CD8⁺ T cells of the GTKO origin in the mixed bone-marrow chimeras. We believe that CD44⁺ CD4⁺ cells mostly represent Tregs, especially in the Abin1 GTKO mice, although we did not co-stain CD44 with FOXP3 in these experiments. We will perform the co-staining of FOXP3 and CD44 (and GITR and OX40) in T cells from the polyclonal WT and Abin1 GTKO mice. In the current manuscript, we are already showing the frequencies and absolute numbers of naive, AIMT (central-memory like) and antigen experienced cells (AgE) CD8⁺ T cells in the spleen and lymph nodes in Figure 3A and Figure EV3A, which are the main populations we monitor also in the steady-state mice. We will make the labeling in the figure clearer to indicate the markers of these subsets, as we did in other figures.

Fig. 3C: The authors show that ABIN1 GTKO have more Tregs and that Treg cells are functional. The authors identified ABIN1 as a component of the OX40 co-stimulatory complex, which serves critical functions on CD4+ Treg cells. Is OX40 expression changed on ABIN1 KO Treg cells? Have the authors closely looked by titrating Treg and conv CD4 T ratios, if ABIN1 GTKO Treg cells are even stronger in suppressing proliferation compared to WT Treg cells?

As mentioned above, we will co-stain FOXP3, OX40, GITR (and CD44) on T cells from polyclonal mice to see if the levels of these co-stimulation receptors are influenced by ABIN1 deficiency.

We did not do a detailed titration of Tregs in the suppression assay, as this was not the major scope of our study. Anyway, in our ex vivo suppression assay using anti-CD3 antibody, there are probably no cells expressing the GITR and OX40 ligands. We agree that this might be a nice experiments and will comment on this in the revised, but we believe that this goes beyond the current major scope of the manuscript, which is the regulation of CD8⁺ T cells by ABIN1 and the p38 axis.

Fig. 3D: I κ B α is prone to degradation and strong re-synthesis, which makes interpretation of FACS experiments in the re-stimulation difficult to interpret, especially since I κ B α expressed at lower levels in ABIN1 GTKO. Alternative assays may be used, e.g. p-p65 or induction of 'classical' NF- κ B target genes like p-p65 staining or induction of 'classical' NF- κ B target genes like NFBIA, TNFAIP3 and TNFA by qPCR to support that ABIN1 does or does not affect NF- κ B signaling.

We are thankful for this comment. It motivated us to analyze a group NFKB/RelA target genes (and TNF) in the α CD3/ α CD28-activated WT and GTKO CD8⁺ OTI T cells. As shown and discussed above, these genes were rather down-regulated in ABIN1-deficient T cells. Since the NFKB target genes are not upregulated in ABIN-deficient T cells upon TCR signaling, we find it unlikely that they will be strongly induced by the GITR co-stimulation signaling in these cells. As mentioned above, we will revise the manuscript concerning the potential role of ABIN1 in the NFKB signaling in T cells.

Fig. 5C: The authors note 'observed that ABIN1-deficient T cells showed slightly augmented nuclear translocation of NF- κ B, but only very minor, if any, effect on NFAT nuclear translocation'. Despite being highly significant, it is difficult to detect any different between WT and ABIN1 KO and why is the change in NFAT less important.

As mentioned above, we will change the representation of this experiment to make it clearer. We will also revise the text accordingly.

Fig. 5D: The proliferation effects should be quantified. The authors should also use an IKK/NF- κ B pathway inhibitor (e.g. MLN120B) to determine the effects of the NF- κ B axis.

We will add the quantification of this experiment using expansion indexes. We agree that using an IKK/NF- κ B pathway inhibitor might provide interesting data. However, we decided to focus on the p38 pathway in this manuscript, since it seemed to be more promising. The re-analysis of our RNAseq data with the emphasis on the NFKB target genes did not suggest that the hyperactivation of ABIN1-deficient T cells is mediated by the NFKB pathway (Figures 4-7 here).

Fig. 6: Overall, it is not so clear if there are indeed strong effects on cytotoxic T cells in the in vivo model as suggested by the title. A, B: Was there any effects on LM clearance by ABIN1 GTKO OT-I cells? C: What markers are expressed on tumor-infiltrating ABIN1 KO OT-1? Is there evidence for T cell exhaustion? E, F: The effect of ABIN1 KO OT-1 on inducing autoimmune diabetes looks impressive, but F measures glucose in urine and G in blood (day 7 only). It seems that the result in blood could look quite different, if measurements have been done one or two days later. How do the authors explain that there is almost no difference in blood glucose between 250 and 1000 OT-1 cells, while the difference in urine is only strongly evident with 250 OT-1 cells. With the few OT-1 cells it is of course difficult to track if the ABIN1 KO cells are indeed more activated in vivo, but can this be done ex vivo by incubating OT-1 cells with OVA-expressing pancreas cells? Such ex vivo data could strongly support the model that ABIN1 deficiency enhances cytotoxic T cell responses.

We did not observe a large reproducible effect on Listeria clearance. However, the Listeria clearance is very rapid in WT mice and the contribution of transferred naïve cells is not often apparent (in contrast to the transfer of memory T cells). We have measured selected surface markers on the tumor infiltrating cells, including PD-1, but we did not observe any clear differences. We will include these data in the revised version of the manuscript. As we could not see a clear role of ABIN1 in the tumor clearance, we did not follow this further, e.g. to see whether these infiltrating cells get exhausted faster in later time points. We still believe that ABIN1 deficient T cells might be able to efficiently kill tumor cells in certain conditions (e.g., when preactivated), but we decided not to perform additional experiments in this project.

This diabetic model is very well established in our hands (PMID: 36705564, PMID: 36275704, PMID: 27188212). The glucose levels in the blood almost perfectly correlates with the glucose levels in the urine (Tsyklauri et al. PMID:6705564, Figure1 – Figure Supplement 1A; also Figure 8 here). For this reason, we measure the blood only once (typically on day 7) to reduce the stress of the animals. The reviewer is correct that the blood glucose would recapitulate the urine levels on day 8 and 9. We do not find any potential issue in this. In this assay, we have titrated the number of transferred OT-I cells. We know that 1000 OT-I T cells is a threshold number to induce diabetes in the WT settings. Thus, it is not surprising that the mice receiving 1000 WT OT-I T cells do develop diabetes, although it takes them longer than for those receiving 1000 GTKO OT-I T cells. It is also not surprising that 250 WT OT-I T cells are not sufficient. Thus, the most important result is that 250 GTKO OT-I T cells are able to induce diabetes in this setup. It is even more striking that they are able to do it slightly faster than 1000 WT OT-I T cells. We agree that additional data

in this model are desired. We do not believe that ex vivo co-culture of OT-I T cells and the OVA-expressing pancreatic beta cells would give us much valuable data. It would be also quite a difficult experiment for us, as we never isolated pancreatic islets, which is believed to be technically challenging. However, as mentioned above we will analyze OT-I T cell numbers and activation state, the insulin production by pancreatic islets and the CD8 T cells infiltration in the pancreatic islets. We already have data on the in vivo priming of (larger numbers of) OT-I T cells by DC-OVA, which was used in this model (example is in the Figure 1 here).

Figure 8. Blood glucose (mmol/l) in RIP.OVA mice stratified according to urine glucose levels (mg/dl). n= 427 mice in 23 experiments done in our laboratory in different projects.

Minor points:

A20 is introduced as a deubiquitinase (page 2). Indeed, the function of A20 DUB activity remains obscure and recent work indicates that A20 binding to ubiquitin chains may be more relevant to counterbalance immune and inflammatory signaling.

We agree with the Ref. We will mention it in the manuscript and refer to a study by De et al. (PMID: 24878851).

CMB complex page 10 should be CBM complex.

Thank you for spotting this. We will correct it.

Fig. 6F and G have been mixed up legend/figure. Diabetes-free Kaplan Meyer curve in Fig. 6F is not a survival curve as mentioned in legend.

Thank you for spotting this. We will correct it.

Dear Dr. Stepanek,

Thank you for transferring your manuscript to EMBO reports. I now went through the manuscript, the referee reports from The EMBO Journal (attached again below) and your preliminary point-by-point-response (revision plan). The referees have several concerns and suggestions to improve the manuscript, or to strengthen the data and the conclusions drawn.

Given the constructive referee comments, I would like to invite you to revise your manuscript with the understanding that all concerns of the referees must be addressed in the revised manuscript or in a finalized detailed point-by-point response, as indicated in your revision plan. Acceptance of your manuscript will depend on a positive outcome of another round of review at EMBO reports, using the same referees.

1) a .docx formatted version of the final manuscript text (including legends for main figures, EV figures and tables), but without the figures included. Please make sure that changes are highlighted to be clearly visible. Figure legends should be compiled at the end of the manuscript text.

2) individual production quality figure files as .eps, .tif, .jpg (one file per figure), of main figures and EV figures. Please upload these as separate, individual files upon re-submission. Please make sure that all figure panels are called out separately and sequentially in the manuscript text

For more details please refer to our guide to authors:

See also our guide for figure preparation:

Moreover, please consult our guidelines for figure legend preparation:

4) a complete author checklist, which you can download from our author guidelines

(<https://www.embopress.org/page/journal/14693178/authorguide>). Please insert page numbers in the checklist to indicate where the requested information can be found in the manuscript. The completed author checklist will also be part of the RPF.

5) that primary datasets produced in this study (e.g. RNA-seq, ChIP-seq and array data) are deposited in an appropriate public database. This is now mandatory (like the COI statement). If no primary datasets have been deposited in any database, please state this in this section (e.g. 'No primary datasets have been generated and deposited').

The accession numbers and database should be listed in a formal "Data Availability" section (placed after Materials & Methods) that follows the model below. Please note that the Data Availability Section is restricted to new primary data that are part of this study.

Data availability

8) Regarding data quantification and statistics, please make sure that the number "n" for how many independent experiments were performed, their nature (biological versus technical replicates), the bars and error bars (e.g. SEM, SD) and the test used to calculate p-values is indicated in the respective figure legends (also for potential EV figures and all those in the final Appendix). Please also check that all the p-values are explained in the legend, and that these fit to those shown in the figure. Please provide statistical testing where applicable. Please avoid the phrase 'independent experiment', but clearly state if these were biological or technical replicates. Please also indicate (e.g. with n.s.) if testing was performed, but the differences are not significant. In case n=2, please show the data as separate datapoints without error bars and statistics.

See also:

<http://www.embopress.org/page/journal/14693178/authorguide#statisticalanalysis>

If $n < 5$, please show single datapoints for diagrams.

9) Please add scale bars of similar style and thickness to microscopic images, using clearly visible black or white bars (depending on the background). Please place these in the lower right corner of the images themselves. Please do not write on or near the bars in the image but define the size in the respective figure legend.

10) Please note our reference format:

11) We updated our journal's competing interests policy in January 2022 and request authors to consider both actual and perceived competing interests. Please review the policy <https://www.embopress.org/competing-interests> and add a statement declaring your competing interests. Please name that section 'Disclosure and Competing Interests Statement' and add it after the author contributions section.

12) Please order the manuscript sections like this using these names:

Title page - Abstract - Keywords - Introduction - Results - Discussion - Materials and Methods - Data availability section (DAS) - Acknowledgements - Disclosure and Competing Interests Statement - References - Figure legends - Expanded View Figure legends

13) Please make sure that all the funding information is also entered into the online submission system and is complete and similar to the one in the manuscript text file (in the Acknowledgements).

14) We now use CRediT to specify the contributions of each author in the journal submission system. CRediT replaces the author contribution section. Please use the free text box to provide more detailed descriptions. Thus, please remove the author contributions section from the final manuscript text file. See also guide to authors:

<https://www.embopress.org/page/journal/14693178/authorguide#authorshipguidelines>

15) We would encourage you to use 'Structured Methods', our new Materials and Methods format. According to this format, the Materials and Methods section should include a Reagents and Tools Table (listing key reagents, experimental models, software and relevant equipment and including their sources and relevant identifiers) followed by a Methods and Protocols section in which we encourage the authors to describe their methods using a step-by-step protocol format with bullet points, to facilitate the adoption of the methodologies across labs. More information on how to adhere to this format as well as downloadable templates (.doc or .xls) for the Reagents and Tools Table can be found in our author guidelines (section 'Structured Methods'):

I look forward to seeing a revised version of your manuscript when it is ready. Please let me know if you have questions or comments regarding the revision.

Kind regards,

Achim

Referee #1:

The manuscript by Janusova and colleagues reports a negative regulator role of ABIN1 in T-cell antigen receptor (TCR) signaling in cytotoxic T cells. Yin, Krappman and colleagues have previously reported that A20 and ABIN1 cooperate to inhibit TCR signaling (Yin et al., Cell and Mol Life Sciences 2022). Deficiency of ABIN1 in primary human CD4+ T cells results in a hyperactive TCR with enhanced IL-2 and TNF α production. This previous work further demonstrates that ABIN1 deficiency results in enhanced NF κ B-activation in Jurkat T cells. The current study by Janusova and colleagues extends our current understanding of TCR signaling and ABIN1 functions in CD8+ T cells. However, the manuscript needs additional experimental validation to support their conclusions. Their data further suggest that are more complex functions of ABIN1 to both positively and negatively modulate TCR activation that require further exploration and dissection.

The authors conclude that ABIN1 negatively regulates TCR-activated p38 and NF κ B signaling pathways that, in turn, controls effector programs including IFN γ and GZMB. Supporting this hypothesis, ABIN1(KO) OT-1 CD8+ T cells demonstrate enhanced IFN γ and GZMB (Fig 4C-D). Conversely, the authors also demonstrate that ABIN1 positively regulates IL17, IL23R and RORC as ABIN1(KO) OT-1 CD8+ T cells exhibit decreased TCR-induced IL17 gene signatures. This is interesting, but demonstrates a more complicated model for ABIN1 in controlling potentially distinct effector arms of cytotoxic T cells than the authors' present model. This needs further exploration and discussion to better understand this potential dichotomous ABIN1 functions.

Major concerns:

1. This complexity may confound the in vivo experimental data presented. Three in vivo systems are explored- (1). bacterial infection with a Listeria (OVA) model, (2) anti-tumor immunity with a syngeneic MC38(OVA) model and (3) RIP.OVA experimental autoimmune diabetes (AID) model. However, data presented for each of the model lacks a complete functional and mechanistic understanding. For Listeria(OVA), while greater expansion of activated CD8+ T cells, there is no data reported for bacteria load. For the MC38(OVA) model, while there is greater tumor infiltration of ABIN1(KO) OT-1 CD8+ T cells, there is not greater anti-tumor immunity. The most promising in vivo phenotype is reported for the RIP1.OVA AID model with higher levels of blood glucose by day 7 and greater mortality of mice with ABIN1(KO) OT-1 CD8+ T cells. However, in this model, there is no immunologic characterization. It would be better to focus on one system with complete immunologic and phenotypic characterization to support the authors' conclusions.

2. As ABIN1 also positively regulates IL-17, do ABIN1(KO) CD8+ T cells demonstrate an in vivo model (e.g., experimental autoimmune encephalitis or colitis)? These in vivo data would support the in vitro experimental observations of ABIN1(KO) CD8+ T cells.

3. The thymic development effect in GTKO mice appears impressive (EV4A- thymus). There appears to be a ~300% increase in CD8+ SP thymocytes which is substantial. Hence, the peripheral CD8+ T cells may well be different due to thymic development effects. This appears to be supported by the PC analysis of WT vs GTKO non-activated CD8+ T cells (EV4B). These baseline differences are not accounted for in subsequent TCR-activation experiments.
4. Figure 4B- Missing are statistical analysis of cell division comparisons between WT vs GTKO CD8+ OT-1 T cells.
5. Figure 5B- statistical analysis needs to be applied to this data set before concluding that the activated GTKO CD8+ T cells have elevated p38.
6. Figure 5C- I am not quite following the authors' conclusion that this figure demonstrates augmented nuclear translocation of NFkB, but not NFAT. The statistical analysis shown in the figure that both NFkB and NFAT translocation are different for WT vs GTKO, though the data look quite similar.

Minor concerns:

1. The description of dE5 construct is unnecessary for the crux of this study and adds confusion. As it turns out the dE5 construct did not result in a true knockout, but there is little understanding of how the residual low molecular weight ABIN1 protein arises. Further, there is no additional characterization of the dE5 knockout. I suggest taking all of this data out of the text and EV figure 2. The manuscript should just start with the true GTKO.
2. The in vitro demonstration of enhanced p38 activation with GITRL in CD4+ T cells (figure 3D) is of unclear biological significance. There is no additional in vivo exploration of this in vitro observation.
3. Page 3, line 7- I believe that Fig. EV1A should be Figure 1A.

Referee #2:

In the present study, Janusova and colleagues identify the adaptor protein A20-binding inhibitor of NF- κ B 1 (ABIN1) as component of the signaling complexes of GITR and OX40, members of the T cell co-stimulatory superfamily. To achieve this, the authors generated a Flag-tagged recombinant GITR (and OX40) ligand to isolate its signaling complex and identify components thereof by mass spec. Different strategies were used to establish gene-targeted mouse strains lacking ABIN1 to address its function in T cells. One of these strains, Abin1 GTKO/GTKO, was further characterized and served to show that mice lacking ABIN1 possess elevated numbers of peripheral CD4 and CD8 T cells. Among the CD4 T cells, the frequency of activated and regulatory T cells was higher in the ABIN1-KO mice, whereas in the CD8 T cell fraction, CD44+ cells were more abundant. Interestingly, the authors found that ABIN1-deficiency resulted in elevated GITRL-mediated p38 MAPK signaling and a decreased basal NF- κ B activity in CD4 cells, indicative of a negative regulatory function of ABIN1 in T helper cells. Surprisingly, this was much weaker or not observed in CD8 T cells. It is unclear why and a pity that the authors solely further investigated the role of ABIN1 in CD8 but not in CD4 T cells. Consequently, Listeria (expressing the model antigen OVA) infection was similarly cleared by both WT and ABIN1-deficient OT-I CTLs, despite ABIN1-GTKO OT-I showed higher expansion and preferred differentiation into SLECs as compared to WT OT-I cells. A difference, however, was seen with a low affinity OVA peptide variant. Similarly, in a colon carcinoma model, ABIN1-GTKO OT-I cells more strongly infiltrated the tumor than WT OT-I cells but without possessing a better anti-tumor immune response. Notably, in a final auto-immune diabetes mouse model, ABIN1-deficient OT-I cells were much more prone in inducing an autoimmune pathology. This implies that ABIN1 owns critical regulatory functions in CD4 T cells, and Th17 and/or Treg cells in particular, rather than in CTLs. This however, has not been assessed. The potential importance of Th17 cells is supported by the performed RNA seq experiments, which indicate lower expression levels of IL-17/23 genes in ABIN1-deficient OT-I cells.

Overall, this is a very well executed, state-of-the-art, technically sound, rigorous and impressive study that deserves publication. The only critic I have is that CD4 T cells have not been included for in depth functional analysis. For this crossing ABIN1-GTKO e.g. with OI-II mice would have been required.

Referee #3:

Janusova et al identified ABIN1 (A20 binding inhibitor of NF- κ B1) as a component of the GITR and OX40 co-stimulatory complexes in T cells. To investigate the physiological role of ABIN1 in T cells, the authors generated Abin1 KO mice (GTKO) and performed immune phenotypic analyses. Using bone marrow chimeras, they elucidate lymphocyte-extrinsic and -intrinsic functions of ABIN1. While T cell populations were increased in ABIN1 GTKO, only CD44+ activated T cells and FOXP3+ Treg cells retained higher frequencies in a competitive setting with WT bone marrow cells, arguing for a cell intrinsic effect. In addition, enhanced isotype switching and plasma cell differentiation were largely cell intrinsic. The authors provide evidence for augmented p38 and NF- κ B signaling in ABIN1 GTKO in response to GITRL stimulation or CD3/CD28 co-stimulation. Titration experiments reveal that T cell activation markers (CD69, CD25, CD44) and proliferation are induced more strongly in ABIN1 KO T cells especially with weaker antigenic stimulations. Transcriptional profiling indicated stronger activation of pro-inflammatory gene such as IFNG and GSMB in ABIN1 KO T cells. The authors performed three in vivo models using adoptive transfer ABIN1 KO or WT OT-1 T cells. ABIN1 KO OT1 cells showed mildly increased cytotoxic responses in listeria and LCM infection models, enhanced OT-1 tumor infiltration in a MC38 syngeneic tumor model and stronger induction in a murine autoimmune diabetes

model. Overall, the manuscript demonstrates for the first time a T cell intrinsic function of the NF- κ B negative regulator ABIN1.

Major points:

The manuscript contains a lot of data and it is in parts very difficult to follow. Some data are highly relevant, some may be not quite as important. For instance, three ABIN1 KO strategies have been performed, but apparently only ABIN1 GTKO yielded a full KO without any truncated form. Why are data from the other KOs still included, especially because it is not clear to the reader what truncations of ABIN1 are retained in GT or dE5 KO? Further, unconditional ABIN1 KO mice have been published before and some immune phenotyping has been done. What is already known about the effects on lymphocytes population compared to previous ABIN1 KO mice (also in PMID: 22011580)? Have other strategies maybe failed to obtain a full KO? Better discussion and explanation is needed in light of previous findings. Is it really necessary to show ABIN1 GT and dE5 KOs? If so, please indicate what truncated variants may still be expressed.

Studies on T cell signaling could be more rigorous to draw clear conclusions. I κ Ba protein amounts in single cell assays is difficult to interpret, because of the NF- κ B feedback regulation. Further, the single cell image stream assays in Fig. 6C is not convincing for showing relevant enhancement of NF- κ B activation in ABIN1 GTKO T cells. Freshly isolated CD4 T cells should be used to perform biochemical analyses by Western blotting (pI κ Ba, I κ Ba, p-p65, translocation of NF- κ B and/or EMSA) to demonstrate effects on TCR/CD28 and PMA/Iono stimulation. Also, activation of MAPKs can be monitored.

Along the same line, the authors start by identifying ABIN1 in the context of the GITR and OX40 co-receptor complexes. With the exception of Fig. 6D, all subsequent analyses on signaling pathways were performed using CD3, CD3/CD28 or OVA stimulation. Given the opening of the manuscript, bit stronger evidence would be helpful, if GITR and OX40 signaling are controlled by ABIN1. Is it possible to monitor NF- κ B or MAPK translocation in response to GITR or OX40 stimulation using image stream? Is PMA/Iono-induced expression of GITR and OX40 changed in ABIN1 GTKO CD4 T cells?

Specific points

Fig. 2B: In PMID 22011580, the authors show a significant reduction in CD4 and CD8 T cell numbers in the spleen. What is different to ABIN1 GTKO mice used in this study, which shows increased levels of CD4 and CD8 T cells?

Fig. 3A and text: The authors write that no enhanced expansion and differentiation of T cells was noted in the T cells of ABIN1 GTKO chimeras, but an increase in CD44^{hi} effector points to effects on differentiation. The authors should show effects on naive, effector-memory, central-memory like T cell populations.

Fig. 3C: The authors show that ABIN1 GTKO have more Tregs and that Treg cells are functional. The authors identified ABIN1 as a component of the OX40 co-stimulatory complex, which serves critical functions on CD4⁺ Treg cells. Is OX40 expression changed on ABIN1 KO Treg cells? Have the authors closely looked by titrating Treg and conv CD4 T ratios, if ABIN1 GTKO Treg cells are even stronger in suppressing proliferation compared to WT Treg cells?

Fig. 3D: I κ Ba is prone to degradation and strong re-synthesis, which makes interpretation of FACS experiments in the re-stimulation difficult to interpret, especially since I κ Ba expressed at lower levels in ABIN1 GTKO. Alternative assays may be used, e.g. p-p65 or induction of 'classical' NF- κ B target genes like p-p65 staining or induction of 'classical' NF- κ B target genes like NFBIA, TNFAIP3 and TNFA by qPCR to support that ABIN1 does or does not affect NF- κ B signaling.

Fig. 5C: The authors note 'observed that ABIN1-deficient T cells showed slightly augmented nuclear translocation of NF- κ B, but only very minor, if any, effect on NFAT nuclear translocation'. Despite being highly significant, it is difficult to detect any different between WT and ABIN1 KO and why is the change in NFAT less important.

Fig. 5D: The proliferation effects should be quantified. The authors should also use an IKK/NF- κ B pathway inhibitor (e.g. MLN120B) to determine the effects of the NF- κ B axis.

Fig. 6: Overall, it is not so clear if there are indeed strong effects on cytotoxic T cells in the in vivo model as suggested by the title. A, B: Was there any effects on LM clearance by ABIN1 GTKO OT-I cells? C: What markers are expressed on tumor-infiltrating ABIN1 KO OT-1? Is there evidence for T cell exhaustion? E, F: The effect of ABIN1 KO OT-1 on inducing autoimmune diabetes looks impressive, but F measures glucose in urine and G in blood (day 7 only). It seems that the result in blood could look quite different, if measurements have been done one or two days later. How do the authors explain that there is almost no difference in blood glucose between 250 and 1000 OT-1 cells, while the difference in urine is only strongly evident with 250 OT-1 cells. With the few OT-1 cells it is of course difficult to track if the ABIN1 KO cells are indeed more activated in vivo, but can this be done ex vivo by incubating OT-1 cells with OVA-expressing pancreas cells? Such ex vivo data could strongly the model that ABIN1 deficiency enhances cytotoxic T cell responses.

Minor points:

A20 is introduced as a deubiquitinase (page 2). Indeed, the function of A20 DUB activity remains obscure and recent work

indicates that A20 binding to ubiquitin chains may be more relevant to counterbalance immune and inflammatory signaling.

CMB complex page 10 should be CBM complex.

Fig. 6F and G have been mixed up legend/figure. Diabetes-free Kaplan Meyer curve in Fig. 6F is not a survival curve as mentioned in legend.

Point by point response to Reviewers' comments

We address the Reviewers' comments by performing additional experiments and changing the text of the manuscript in line with our initial agreement with the handling editor, Dr. Achim Breiling. We believe that this revision substantially improved the quality of the manuscript. Our detailed response to the comments is below.

Referee #1:

The manuscript by Janusova and colleagues reports a negative regulator role of ABIN1 in T-cell antigen receptor (TCR) signaling in cytotoxic T cells. Yin, Krappman and colleagues have previously reported that A20 and ABIN1 cooperate to inhibit TCR signaling (Yin et al., Cell and Mol Life Sciences 2022). Deficiency of ABIN1 in primary human CD4+ T cells results in a hyperactive TCR with enhanced IL-2 and TNF α production. This previous work further demonstrates that ABIN1 deficiency results in enhanced NF κ B-activation in Jurkat T cells. The current study by Janusova and colleagues extends our current understanding of TCR signaling and ABIN1 functions in CD8+ T cells. However, the manuscript needs additional experimental validation to support their conclusions. Their data further suggest that are more complex functions of ABIN1 to both positively and negatively modulate TCR activation that require further exploration and dissection.

The authors conclude that ABIN1 negatively regulates TCR-activated p38 and NF κ B signaling pathways that, in turn, controls effector programs including IFN γ and GZMB. Supporting this hypothesis, ABIN1(KO) OT-1 CD8+ T cells demonstrate enhanced IFN γ and GZMB (Fig 4C-D). Conversely, the authors also demonstrate that ABIN1 positively regulates IL17, IL23R and RORC as ABIN1(KO) OT-1 CD8+ T cells exhibit decreased TCR-induced IL17 gene signatures. This is interesting, but demonstrates a more complicated model for ABIN1 in controlling potentially distinct effector arms of cytotoxic T cells than the authors' present model. This needs further exploration and discussion to better understand this potential dichotomous ABIN1 functions.

We thank this Reviewer for evaluating the manuscript and for their useful comments. We agree that the role of ABIN1 is complex and we discuss it in the revised manuscript. We mainly focused on the role of ABIN1 function in conventional cytotoxic CD8+ T cells, where our data identify ABIN1 as a negative regulator of the formation of cytotoxic effector T cells ex vivo and in vivo.

As the scope of this study was primarily the role of ABIN1 in CD8+ T cells, we have not generated proper TCR transgenic models to study the role of ABIN1 in the CD4+ T-cell lineage in controlled experiments, yet. However, we added an analysis of polyclonal CD4+ T cells from the ABIN1 GTKO mice showing that ABIN1 does not seem to promote IL-17 production and Th17 differentiation in the CD4+ T-cell compartment in this model (Fig. EV5F of the revised manuscript). We believe that the role of ABIN1 in the antigenic responses and fate choices of CD4+ T cells is an intriguing question and an important topic for a follow-up project.

Major concerns:

1. This complexity may confound the in vivo experimental data presented. Three in vivo systems are explored- (1). bacterial infection with a *Listeria* (OVA) model, (2) anti-tumor immunity with a syngeneic

MC38(OVA) model and (3) RIP.OVA experimental autoimmune diabetes (AID) model. However, data presented for each of the model lacks a complete functional and mechanistic understanding. For Listeria(OVA), while greater expansion of activated CD8+ T cells, there is no data reported for bacteria load. For the MC38(OVA) model, while there is greater tumor infiltration of ABIN1(KO) OT-1 CD8+ T cells, there is not greater anti-tumor immunity. The most promising in vivo phenotype is reported for the RIP1.OVA AID model with higher levels of blood glucose by day 7 and greater mortality of mice with ABIN1(KO) OT-1 CD8+ T cells. However, in this model, there is no immunologic characterization. It would be better to focus on one system with complete immunologic and phenotypic characterization to support the authors' conclusions.

We did not observe a large reproducible effect on *Listeria* clearance. However, the *Listeria* clearance is very rapid in WT mice and it is well known that the contribution of even high numbers of transferred naïve antigen-specific T cells to *Listeria* clearance is usually not apparent (in contrast to the transfer of memory T cells).

Concerning the tumor model, we speculate in the revised discussion that the over-activation of p38 leads to the preferential formation of effector T cells at the expense of stem-like cells, which are required for the tumor clearance. This is in line with previous reports showing the role of p38 in T-cell senescence and actually inhibiting the anti-tumor activities of T cells (PMID: 25151490, PMID: 32516591), which would explain strong initial infiltration of ABIN1-deficient T cells without increased anti-tumor activity in the long term in our experiments. We discuss this in the revised version of the manuscript.

During the revisions, we followed the recommendation of this Reviewer and focused more on one model – we chose the diabetic model, which showed significant functional differences. We observed greater expansion and effector cell formation of ABIN1-deficient OT-I T cells in this model (response of OTI to DC-OVA immunization in the congenic model – Fig. 6E, and slightly also in the RIP.OVA diabetic model – Fig. EV6G). Moreover, we added H&E and IF stained histology sections of the pancreas showing enhanced lymphocyte infiltration, insulinitis, and pancreatic islets destruction when 250 GTKO cells were transferred (Fig. 6I, EV6H). We believe that this additional data make the manuscript more convincing.

2. As ABIN1 also positively regulates IL-17, do ABIN1(KO) CD8+ T cells demonstrate an in vivo model (e.g., experimental autoimmune encephalitis or colitis)? These in vivo data would support the in vitro experimental observations of ABIN1(KO) CD8+ T cells.

Although ABIN1 might play a role in multiple disease conditions, it would be extremely difficult to study ABIN1 in these models and it would be definitely beyond the scope of our study. The EAE induction in *Abin1*^{GTKO/GTKO} mice would be very challenging to interpret because these mice already have spontaneous signs of the autoimmune pathology. The same would apply to DSS-induced colitis. The transfer of polyclonal T cells into Rag KO mice (colitis model) could be also influenced by prior changes in the CD4⁺ T-cell compartment in these mice. Overall, ABIN1 has pleiotropic roles in many cell types, which would extrinsically influence the outcome of these models. For instance, chronic LCMV infection is lethal for *Abin1*^{GTKO/GTKO} mice. Because uncovering the mechanisms behind this lethality would be a whole project on its own and because of ethical reasons, we did not follow this further.

We would like to stress that the major line of our study focuses on cytotoxic CD8⁺ T cells, whereas CD4⁺ T cells are mostly involved in the classical EAE and colitis models. We believe that the diabetes model based on the transfer of OT-I T cells into the RIP.OVA mice has two unique features: i) It is mediated by CD8⁺ T cells, which are the major focus of our study and ii) it largely neutralizes the steady-state effects of ABIN1-deficiency, because it is based on a transfer of monoclonal T cells. Although we speculate that ABIN1 might inhibit the formation of Tc17, this is not the major finding of our study, as Tc17 cells represent rather a small unconventional population. Also there is no widely established autoimmune model, where this subset would play a key role.

3. The thymic development effect in GTKO mice appears impressive (EV4A- thymus). There appears to be a ~300% increase in CD8+ SP thymocytes which is substantial. Hence, the peripheral CD8+ T cells may well be different due to thymic development effects. This appears to be supported by the PC analysis of WT vs GTKO non-activated CD8+ T cells (EV4B). These baseline differences are not accounted for in subsequent TCR-activation experiments.

This Reviewer is correct that there are more mature SP8 thymocytes in *Abin1*^{GTKO/GTKO} than in WT OT-I *Rag2*^{KO/KO} mice. We believe that this is most likely caused by enhanced positive selection mediated by the lack of negative regulation of the TCR signal by ABIN1 (MHCII-restricted OT-I T cells are committed to differentiate into SP8 thymocytes). We think the risk of this leading to substantially altered peripheral phenotype is relatively low and acceptable in this model. However, we revised the manuscript to highlight this potential limitation. Of course, we would definitely love to address this potential issue, but as the *Abin1*^{dE5} allele did not work (Fig. EV2B, Appendix Fig. 3), we were not able to generate the inducible KO to get over this.

We interpret the PCA analysis (Fig. EV4B) as not actually showing such a clear difference between non-activated WT and GTKO cells. The PCA clearly separates these two groups only after activation. However, we agree that some differences are there prior to the activation, such as a higher percentage (although still quite low) of CD49d⁺ CD44⁺ cells in the spleen (Fig. EV4A) and the steady-state hyperphosphorylation of p38 (Fig. 5A). We added a volcano plot showing that some genes are upregulated in the GTKO cells even prior activation (Fig. EV4C). However, the number of these genes is much lower than in the activated samples (compare with Fig. 4C). Concerning the genes of our major interest, *Il17a*, *Il17b*, *Rorc*, *Ifng*, *Prf1*, and *Gzmb* were not differentially expressed between non-activated WT and GTKO cells. However, *Casp1*, *Casp4*, *Serpina3g*, and *Serpin3f* were differentially expressed prior to the activation.

In the TCR-activation experiments, we always use non-stimulated controls to account for the baseline differences. We see baseline differences in the p38 activation (which is hyperphosphorylated even in freshly isolated cells – Fig. 5A), but not in other signaling intermediates.

4. Figure 4B- Missing are statistical analysis of cell division comparisons between WT vs GTKO CD8+ OT-1 T cells.

We included the statistical analysis.

5. *Figure 5B- statistical analysis needs to be applied to this data set before concluding that the activated GTKO CD8+ T cells have elevated p38.*

We included the statistical analysis.

6. *Figure 5C- I am not quite following the authors' conclusion that this figure demonstrates augmented nuclear translocation of NFkB, but not NFAT. The statistical analysis shown in the figure that both NFkB and NFAT translocation are different for WT vs GTKO, though the data look quite similar.*

We agree with this Reviewer that the previous representation of the data was confusing. To fix this, we only show the medians for individual experiments in the revised version, which documents the reproducibility of the differences in the NFkB nuclear translocation upon suboptimal antigen concentration, but not in NFAT translocation. Concerning the NFkB pathway, we added the analysis of the expression of NFkB target genes based on their kinetics, which suggests that ABIN1 does not regulate the induction of the rapid and transient NFkB targets, but rather the expression of NFkB genes with slower kinetics (Fig. EV4D). This explains why the observed differences in the activation of the NFkB pathway are relatively minor.

Minor concerns:

1. *The description of dE5 construct is unnecessary for the crux of this study and adds confusion. As it turns out the dE5 construct did not result in a true knockout, but there is little understanding of how the residual low molecular weight ABIN1 protein arises. Further, there is no additional characterization of the dE5 knockout. I suggest taking all of this data out of the text and EV figure 2. The manuscript should just start with the true GTKO.*

We discussed this issue with the handling editor, Dr. Achim Breiling. We agreed that this part should remain in the manuscript.

We agree that it is a bit confusing, but our reasons for keeping this part are:

(I) We got the GT allele, which can be used for the generation of presumed whole-body and conditional KOs, from the Infrafrontiers consortium. We believe other scientists potentially interested in this particular strain would appreciate our data that it does not work as expected. On the other hand, additional characterization of the *Abin1*^{dE5/dE5} mice would not improve our story in any way.

(II) If we started this part of the manuscript straight with the GTKO mice, the readers would get confused why we used mice with this allele, as this is rather unusual (the first choice would be to use the dE5 allele).

We adjusted this part of the manuscript to be clearer.

2. *The in vitro demonstration of enhanced p38 activation with GITRL in CD4+ T cells (figure 3D) is of unclear biological significance. There is no additional in vivo exploration of this in vitro observation.*

We have shown that ABIN1 regulates multiple signaling pathways in T cells (GITR, TCR, and probably OX40 and other TNF superfamily receptors). For this reason, it is difficult to dissect the contribution of individual pathways to the observed phenotypes in vivo. Honestly, we do not even think that it is very important to dissect the contribution of the individual pathways at this stage, when we report our initial observations of the in vivo phenotype. Most likely the dysregulation of both antigenic and co-stimulation pathways contribute to this.

We addressed the response of CD4⁺ and CD8⁺ T cells to GITRL to confirm that ABIN1 is a negative regulator of this pathway. We then addressed the role of ABIN1 in cytotoxic CD8⁺ T cells, which is the major research topic in our lab. We also observed unanticipated role of ABIN1 in TCR signaling (which was a bit later also published in the paper PMID: 35099607), which we followed in the second part of the manuscript.

We agree with this Reviewer that there is not much follow-up on CD4⁺ T cells. We did not cross the *Abin1*^{GTKO/GTKO} mice to a transgenic monoclonal MHCII-restricted TCR mouse to specifically address the role of ABIN1 in CD4⁺ T cells in controlled experiments. This would be an excellent research direction for future research. We added some analyses of steady state polyclonal CD4⁺ T cells from the *Abin1*^{GTKO/GTKO} mice (Fig. EV3C, Fig. Fig. EV5F) in the revised manuscript.

3. Page 3, line 7- I believe that Fig. EV1A should be Figure 1A.

We added the reference to Fig. 1A to this sentence.

Referee #2:

*In the present study, Janusova and colleagues identify the adaptor protein A20-binding inhibitor of NF- κ B 1 (ABIN1) as component of the signaling complexes of GITR and OX40, members of the T cell co-stimulatory superfamily. To achieve this, the authors generated a Flag-tagged recombinant GITR (and OX40) ligand to isolate its signaling complex and identify components thereof by mass spec. Different strategies were used to establish gene-targeted mouse strains lacking ABIN1 to address its function in T cells. One of these strains, *Abin1*^{GTKO/GTKO}, was further characterized and served to show that mice lacking ABIN1 possess elevated numbers of peripheral CD4 and CD8 T cells. Among the CD4 T cells, the frequency of activated and regulatory T cells was higher in the ABIN1-KO mice, whereas in the CD8 T cell fraction, CD44⁺ cells were more abundant. Interestingly, the authors found that ABIN1-deficiency resulted in elevated GITRL-mediated p38 MAPK signaling and a decreased basal NF- κ B activity in CD4 cells, indicative of a negative regulatory function of ABIN1 in T helper cells. Surprisingly, this was much weaker or not observed in CD8 T cells. It is unclear why and a pity that the authors solely further investigated the role of ABIN1 in CD8 but not in CD4 T cells. Consequently, *Listeria* (expressing the model antigen OVA) infection was similarly cleared by both WT and ABIN1-deficient OT-I CTLs, despite ABIN1-GTKO OT-I showed higher expansion and preferred differentiation into SLECs as compared to WT OT-I cells. A difference, however, was seen with a low affinity OVA peptide variant. Similarly, in a colon carcinoma model, ABIN1-GTKO OT-I cells more strongly infiltrated the tumor than WT OT-I cells but without possessing a better anti-tumor immune response. Notably, in a final auto-immune diabetes mouse model, ABIN1-deficient OT-I cells were much more prone in inducing an autoimmune pathology. This implies that ABIN1 owns critical regulatory functions in CD4 T cells, and Th17 and/or Treg cells in particular, rather than in CTLs. This however, has not been assessed. The potential importance of Th17 cells is supported by the performed RNAseq experiments, which indicate lower expression levels of IL-17/23 genes in ABIN1-deficient OT-I cells.*

Overall, this is a very well executed, state-of-the-art, technically sound, rigorous and impressive study that deserves publication. The only critic I have is that CD4 T cells have not been included for in depth functional analysis. For this crossing ABIN1-GTKO e.g. with OI-II mice would have been required.

We are very thankful for the positive evaluation by this Reviewer. We agree that the analysis of the role of ABIN1 in CD4⁺ T cells would be very interesting and it can be an important follow-up to this study. We did not cross the *Abin1*^{GTKO/GTKO} mice to a transgenic monoclonal MHCII-restricted TCR mouse to specifically address the role of ABIN1 in CD4⁺ T cells in controlled experiments. However, we added analyses of the steady-state polyclonal CD4⁺ T cells from the *Abin1*^{GTKO/GTKO} mice (Fig. EV3C, Fig. Fig. EV5F) in the revised manuscript.

Referee #3:

Janusova et al identified ABIN1 (A20 binding inhibitor of NF-kB1) as a component of the GITR and OX40 co-stimulatory complexes in T cells. To investigate the physiological role of ABIN1 in T cells, the authors generated Abin1 KO mice (GTKO) and performed immune phenotypic analyses. Using bone marrow chimeras, they elucidate lymphocyte-extrinsic and -intrinsic functions of ABIN1. While T cell populations were increased in ABIN1 GTKO, only CD44+ activated T cells and FOXP3+ Treg cells retained higher frequencies in a competitive setting with WT bone marrow cells, arguing for a cell intrinsic effect. In addition, enhanced isotype switching and plasma cell differentiation were largely cell intrinsic. The authors provide evidence for augmented p38 and NF-kB signaling in ABIN1 GTKO in response to GITRL stimulation or CD3/CD28 co-stimulation. Titration experiments reveal that T cell activation markers (CD69, CD25, CD44) and proliferation are induced more strongly in ABIN1 KO T cells especially with weaker antigenic stimulations. Transcriptional profiling indicated stronger activation of pro-inflammatory gene such as IFNG and GSMB in ABIN1 KO T cells. The authors performed three in vivo models using adoptive transfer ABIN1 KO or WT OT-1 T cells. ABIN1 KO OT1 cells showed mildly increased cytotoxic responses in listeria and LCM infection models, enhanced OT-1 tumor infiltration in a MC38 syngeneic tumor model and stronger induction in a murine autoimmune diabetes model. Overall, the manuscript demonstrates for the first time a T cell intrinsic function of the NF-kB negative regulator ABIN1.

We are thankful for the evaluation of our manuscript and for the useful comments by this Reviewer.

Major points:

1. The manuscript contains a lot of data and it is in parts very difficult to follow. Some data are highly relevant, some may be not quite as important.

We are pleased that this Reviewer appreciates the amount of our data. Indeed this project took many years and was not always straightforward. After obtaining the initial experimental results, we hypothesized that ABIN1-deficient T cells would show superior anti-tumor activity, which could be further followed in translational research. We were disappointed that this was not the case. We then focused on other aspects of T-cell biology, such as their role in the autoimmune model. Overall, we believe that all the data in the current manuscript have some value, even if they show negative results. We show the most interesting results in the main figures, whereas the less crucial results are in the Expanded View Figures and Appendix Figures. As we did not want to cut the corners, we include also negative or ambiguous results. One reason, why we did not proceed for the thorough analysis of CD4⁺ T cells/Tregs in this project, is not to make the manuscript even bulkier.

2. For instance, three ABIN1 KO strategies have been performed, but apparently only ABIN1 GTKO yielded a full KO without any truncated form. Why are data from the other KOs still included, especially because it is not clear to the reader what truncations of ABIN1 are retained in GT or dE5 KO?

We discussed this issue with the handling editor, Dr. Achim Breiling. We agreed that this part should remain in the manuscript.

We agree that it is a bit confusing, but our reasons for keeping this part are:

(I) We got the GT allele, which can be used to generate presumed whole-body and conditional KOs, from the Infrafrontiers consortium. We believe other scientists potentially interested in this particular strain would appreciate our data that it does not work as expected. On the other hand, additional characterization of the *Abin1*^{dE5/dE5} mice would not improve our story in any way.

(II) If we start this part of the manuscript straight with the GTKO mice, the readers might get confused why we use mice with this allele, as this is rather unusual (the first choice would be to use the dE5 allele).

We adjusted this part of the manuscript to be clearer.

3. Further, unconditional ABIN1 KO mice have been published before and some immune phenotyping has been done. What is already known about the effects on lymphocytes population compared to previous ABIN1 KO mice (also in PMID: 22011580)? Have other strategies maybe failed to obtain a full KO? Better discussion and explanation is needed in light of previous findings. Is it really necessary to show ABIN1 GT and dE5 KOs? If so, please indicate what truncated variants may still be expressed.

We agree that the results from the previous studies using ABIN1-deficient mice should be more highlighted in the manuscript. On the other hand, we would like to emphasize here that the analysis of the T-cell compartment by the previous studies was very limited as these studies focused mostly on B cells and/or myeloid cells. We adjusted the discussion of our results and the results of articles published by Philip Cohen (PMID: 21606507) and Zhou et al. (PMID: 22011580) in the revised version of the manuscript. We do not think that there is a huge difference between our results and the results published by Zhou et al. We observed differences in T-cell numbers mostly in the lymph nodes (Fig. 2A, Fig. EV2D), whereas Zhou et al. only showed the results from the spleen. We observed comparable numbers of T cells in the spleen (Appendix Figure 4), while Zhou et al. observed a decrease. In the end, we also observed a decrease in T-cell numbers in the spleens in the mixed bone marrow chimeras. The difference between the studies is apparently caused by T-cell extrinsic factors that might include the housing conditions such a microbiota colonization.

Zhou et al. generated a GT allele, with a GT insertion in the first intron of *Abin1*. Their documentation of this allele/mouse is very brief. However, it seems plausible that their allele might retain the higher molecular weight (MW) isoform of ABIN1. The study shows a tiny section of the membrane corresponding to presumed full-length ABIN1. In this section, a double band is apparent and only the lower band is missing in their ABIN1 GT mice (Figure 1 below). They labeled the band with higher apparent MW and higher intensity as non-specific signal without any evidence shown. We could not find in their article, which antibody they used for the detection of ABIN1 in these immunoblots.

Figure 1. Immunoblot of ABIN1 in the WT and GT mice from the study by Zhou et al. (their Figure 2C). Showing the upper band remaining.

We also detected a double band of ABIN1 (Fig. EV2B in our manuscript). However, the analysis of our GT, GTKO, and dE5 mice clearly shows that both bands represent ABIN1. Accordingly, it is well established that ABIN1 has two isoforms in mice: “The two splice variants of the ABIN cDNA contained an open reading frame of 1,941 and 1,782 nucleotides respectively, initiating at two different methionines [ABIN (1-647) and ABIN (54-647)] (GenBank/EMBL/DDBJ accession numbers AJ242777 and AJ242778). These cDNAs encode proteins of 72 and 68 kD” (Heyninck et al. 1999, PMID: 10385526).

Because of the limited documentation of the mouse reported by Zhou et al. we cannot draw any conclusions concerning the completeness of their KO. For these reasons, we do not comment on the genotypic differences between their mouse and our mouse in the manuscript.

We are convinced it is important to show that the dE5 allele is not a null allele, however, we do not find it crucial to investigate the identity of the artificial truncated protein expressed from this allele. Although it might provide additional insight into the structural function of ABIN1, it would require challenging and time-consuming experiments that are beyond the scope of this manuscript.

4. Studies on T cell signaling could be more rigorous to draw clear conclusions. IκBα protein amounts in single cell assays is difficult to interpret, because of the NF-κB feedback regulation. Further, the single cell image stream assays in Fig. 6C is not convincing for showing relevant enhancement of NF-κB activation in ABIN1 GTKO T cells. Freshly isolated CD4 T cells should be used to perform biochemical analyses by Western blotting (pIκBα, IκBα, p-p65, translocation of NF-κB and/or EMSA) to demonstrate effects on TCR/CD28 and PMA/Iono stimulation. Also, activation of MAPKs can be monitored.

It is likely that this Reviewer meant “Fig. 5C” here. However, we do not think there are any advantages of immunoblotting over flow cytometry/image stream analysis of signaling in the cases where validated antibodies are available (such as IκB, NF-κB p65), since immunoblotting or EMSA are definitely less quantitative. In order to present the data in a less confusing manner, we quantified the data in Fig. 5B as geometric mean fluorescence intensity. We also changed the presentation of Fig. 5C to show the median nuclear translocation from individual experiments to increase clarity.

Any feedback regulations of the signaling intermediates would manifest on the bulk level (immunoblotting) as well. The bulk analysis is nothing more than a sum/average of single cells. Because the OT-I T cells were stimulated with antigen-presenting cells loaded with the cognate peptide, the immunoblotting (or any analysis based on the lysate of these cells) would inevitably combine material from both OT-I cells and T2-Kb antigen presenting cells, which would complicate the proper analysis. We are also not sure, why this Reviewer suggests to use freshly isolated CD4⁺ T cells here, since the experiments in Figure 5 focus on CD8⁺ T cells.

It is true that the difference between WT and GTKO in the activation of the NFκB pathway is small. In some assays even negligible and/or not statistically significant. The observed differences between WT and GTKO in the in vivo and ex vivo activation are larger than the analysis of the signaling intermediates, perhaps because the signaling intermediates are transient and the cells are not perfectly synchronized. The other explanation is that ABIN1 is not much involved in regulating the initiation of NFκB activation in T cells. To address these issues, we performed the analysis of the transcription response using our RNAseq data. We found out that ABIN1 does not regulate NFκB-target genes, which are rapidly and transiently activated after antigenic stimulation, but regulates those with slower kinetics (Fig. EV4D). Thus, ABIN1 probably

controls the duration of NFkB signaling rather than its initiation, as newly discussed in the revised manuscript.

5. Along the same line, the authors start by identifying ABIN1 in the context of the GITR and OX40 co-receptor complexes. With the exception of Fig. 6D, all subsequent analyses on signaling pathways were performed using CD3, CD3/CD28 or OVA stimulation. Given the opening of the manuscript, bit stronger evidence would be helpful, if GITR and OX40 signaling are controlled by ABIN1. Is it possible to monitor NF-kB or MAPK translocation in response to GITR or OX40 stimulation using image stream? Is PMA/Iono-induced expression of GITR and OX40 changed in ABIN1 GTKO CD4 T cells?

We originally identified ABIN1 as a component of the GITR and OX40 signaling complexes. We confirmed that ABIN1 negatively regulates GITR signaling in CD4⁺ T cells. Because the results in polyclonal CD8⁺ T cells were not conclusive, we compared GITR signaling in WT and *Abin1*^{GTKO/GTKO} CD8⁺ OT-I T cells during the revisions, which reduced the role of extrinsic factors in comparison to the analysis of CD8⁺ T cells from polyclonal mice. The analysis of NFkB and p38 pathways, including the NFkB translocation by Image Stream, showed that ABIN1 regulates GITR signaling also in CD8⁺ T cells (Fig. 2E, Fig. EV2E-F).

In the middle of the project, we realized that ABIN1 also regulates TCR signaling, which was later published by Yin et al. (PMID: 35099607). Because of the prominence of this pathway in T-cell biology, we steered the project to study the role of ABIN1 in TCR signaling in primary T cells. We also analyzed the in vivo role of ABIN1 using multiple models, where it probably regulates both the antigenic signaling and the co-stimulation signaling, which is virtually impossible to dissect.

After the PMA/ionomycine treatment, the expression of GITR is comparable (Figure 2 below).

Figure 1. GITR expression on PMA/ionomycine activated T cells from WT and *Abin1*^{GTKO/GTKO} mice.

Specific points

6. Fig. 2B: In PMID 22011580, the authors show a significant reduction in CD4 and CD8 T cell numbers in the spleen. What is different to ABIN1 GTKO mice used in this study, which shows increased levels of CD4 and CD8 T cells?

The absolute T-cell numbers in the spleens of WT and GTKO mice are comparable (Appendix 4). This corresponds to ABIN1[D485N] knock-in mice (Nada et al. 2011, PMID: 21606507). In our and their study, quite normal thymic development in polyclonal mice was observed.

It is unclear, why the mentioned study by Zhou et al. shows the reduction in splenic T cells in ABIN1-deficient mice, although the overall splenocyte counts are significantly elevated. This study does not show thymic development or even lymphocytes from lymph nodes. They also do not show any data on lymphocytes from the bone marrow chimeras. Although not explicitly stated, this study probably used mice backcrossed to the C57BL/6J background for at least 5 generations, which is the same background we used. Potentially, this difference can be caused by different housing conditions (microbial colonization) or by the difference in the modification of the *Abin1* allele (as discussed above).

As already mentioned, we discuss this paper in the revised version of the manuscript.

7. Fig. 3A and text: The authors write that no enhanced expansion and differentiation of T cells was noted in the T cells of ABIN1 GTKO chimeras, but an increase in CD44^{hi} effector points to effects on differentiation. The authors should show effects on naive, effector-memory, central-memory like T cell populations.

This Reviewer is correct that there are increased frequencies of CD44⁺ CD4⁺ T cells and marginally increased antigen-experienced CD8⁺ T cells of the GTKO origin in the mixed bone-marrow chimeras. We believe that the CD44⁺ CD4⁺ T cells at least partially represent Tregs, especially in the *Abin1*^{GTKO/GTKO} mice, although we did not co-stain CD44 with FOXP3 in the bone marrow experiments. However, the data from the steady-state show an increased frequency of Tregs among CD44⁺ CD4⁺ T cells in the GTKO mice (Fig. 3 below).

Figure 2. Percentage of *Foxp3*⁺ T cells among *CD44*⁺ *CD4*⁺ T cells in WT and *Abin1* GTKO mice in the steady state. The color of the data point represent individual experiments. The color of the data points indicate individual experiments.

In the revised manuscript, we show the frequencies and absolute numbers of naïve, AIMT (central-memory like), and antigen experienced cells (AgE) *CD8*⁺ T cells in the spleen and lymph nodes of the bone marrow chimeras in Fig. 3A and Fig. EV3A, which are the main populations we monitor also in the steady-state mice (Fig. 2B, Fig. EV2D). We made the labeling in the revised figures clearer to indicate the markers of these subsets.

8. Fig. 3C: The authors show that *ABIN1* GTKO have more Tregs and that Treg cells are functional. The authors identified *ABIN1* as a component of the *OX40* co-stimulatory complex, which serves critical functions on *CD4*⁺ Treg cells. Is *OX40* expression changed on *ABIN1* KO Treg cells? Have the authors closely looked by titrating Treg and conv *CD4* T ratios, if *ABIN1* GTKO Treg cells are even stronger in suppressing proliferation compared to WT Treg cells?

We stained *GITR* and *OX40* on Treg cells and observed slightly increased expression of both on *Abin1*^{GTKO/GTKO} Tregs than on WT Tregs (Fig. EV3C). Although this difference was not significant with the limited number of mice we could use, it is plausible that *ABIN1*-deficient Tregs have a stronger suppressive activity as suggested by the reviewer. We did not perform titrations of Treg numbers in the ex vivo suppression assay as we mostly focused on the role of *ABIN1* in cytotoxic T cells. However, the *GITRL* and *OX40L* would not be probably present in this simplistic ex vivo assay, so the increased levels of *GITR* and *OX40* should not play any role. Anyway, this is an interesting idea for a follow-up experiments focusing on the role of *ABIN1* in regulatory and conventional *CD4*⁺ T cells.

9. Fig. 3D: *IκBα* is prone to degradation and strong re-synthesis, which makes interpretation of FACS experiments in the re-stimulation difficult to interpret, especially since *IκBα* expressed at lower levels in *ABIN1* GTKO. Alternative assays may be used, e.g. p-p65 or induction of 'classical' NF-κB target genes like p-p65 staining or induction of 'classical' NF-κB target genes like *NFBIA*, *TNFAIP3* and *TNFA* by qPCR to support that *ABIN1* does or does not affect NF-κB signaling.

We are thankful for this comment. It motivated us to analyze a group NFκB target genes in the aCD3/aCD28-activated WT and GTKO CD8⁺ OTI T cells using our RNAseq data. We found out that *ABIN1* does not regulate NFκB-target genes, which are rapidly and transiently activated after antigenic stimulation, but regulates those with slower kinetics (Fig. EV4D). Thus, *ABIN1* probably controls the duration of NFκB signaling rather than the initiation, which is more difficult to observe in assays focused on signaling intermediates. Out of the three canonical NFκB targets mentioned by this Reviewer, *Tnfaip3* and *Nfkbia* were on the list of rapidly and transiently activated NFκB targets (PMID: 37524800) and were not upregulated in *Abin1*^{GTKO/GTKO} cells. TNF was not on the list of NFκB targets in lymphocytes, probably because it is not expressed in B cells (which were used in the study PMID: 3752480, which we used as a reference). However, TNF was not upregulated in activated *ABIN1*-deficient CD8⁺ T cells as shown in Fig. 4 below, similar to the other two early NFκB-responsive genes.

Figure 4. The expression of TNF is triggered by activation, but it is not upregulated in *Abin1* deficient T cells. Analysis of RNAseq data from both experiments.

As shown and discussed above, we did not see upregulation of these genes in *ABIN1*-deficient T cells upon antigenic signaling. For this reason, we find it unlikely that they would be strongly induced by the GITR co-stimulation signaling in these cells. We discuss the role of *ABIN1* in the NFκB signaling in T cells in the revised manuscript.

10. Fig. 5C: The authors note 'observed that *ABIN1*-deficient T cells showed slightly augmented nuclear translocation of NF-κB, but only very minor, if any, effect on NFAT nuclear translocation'. Despite being

highly significant, it is difficult to detect any difference between WT and ABIN1 KO and why is the change in NFAT less important.

We agree with this Reviewer that the previous representation of the data was confusing. To fix this, we only show the medians for individual experiments in the revised version, which demonstrates the reproducibility and clear differences in NFkB, but not NFAT, translocation in WT and GTKO mice. Concerning the NFkB pathway, we added the analysis of the expression of NFkB target genes based on their kinetics, which suggests that ABIN1 does not regulate the expression of the rapid and transient NFkB targets, but rather the expression of NFkB genes with slower kinetics (Fig. EV4D). This explains why the observed differences in the activation of the NFkB pathway are relatively minor. As mentioned above, we changed the representation of this experiment to make it clearer and revised the text accordingly.

11. Fig. 5D: The proliferation effects should be quantified. The authors should also use an IKK/NF- κ B pathway inhibitor (e.g. MLN120B) to determine the effects of the NF- κ B axis.

We tried to quantify the proliferation in Fig. 5D, but it is extremely difficult, because of assay-to-assay variability, which always gives a slightly different division pattern. We are showing here the quantification calculated as the Replication index (Fig. 5 below). Given the small number of replicate experiments and the variability, it does not reach the level of significance. For this reason, we are showing only the proliferation peaks of a representative experiment, which is a common practice in the field. More quantitative analyses of the role of p38 in WT and GTKO T cells were performed using the RNAseq, flow cytometry, and metabolic profiling (Fig. 5F-I, Fig. EV5C-E).

We agree that using an IKK/NF- κ B pathway inhibitor might provide interesting data. However, we decided to focus on the p38 pathway in this manuscript, since we observed more reproducible effects on the p38 than on the NFkB pathway. The second reason was that ABIN1 has already been connected to the NFkB pathway (PMID: 35099607), but its connection to the p38 pathway was novel. We re-addressed the NFkB axis by reanalyzing the RNAseq data as described above, which showed that ABIN1 probably regulates the duration rather than the initiation of the NFkB pathway (Fig. EV4D). We also had a limited number of mice for the revision experiments and we prioritized other experiments, which we found more important.

Figure 5. Replication indexes from four independent experiments as described in Fig. 5D in the revised manuscript.

12. Fig. 6: Overall, it is not so clear if there are indeed strong effects on cytotoxic T cells in the in vivo model as suggested by the title. A, B: Was there any effects on LM clearance by ABIN1 GTKO OT-I cells? C: What markers are expressed on tumor-infiltrating ABIN1 KO OT-1? Is there evidence for T cell exhaustion? E, F: The effect of ABIN1 KO OT-1 on inducing autoimmune diabetes looks impressive, but F measures glucose in urine and G in blood (day 7 only). It seems that the result in blood could look quite different, if measurements have been done one or two days later. How do the authors explain that there is almost no difference in blood glucose between 250 and 1000 OT-1 cells, while the difference in urine is only strongly evident with 250 OT-1 cells. With the few OT-1 cells it is of course difficult to track if the ABIN1 KO cells are indeed more activated in vivo, but can this be done ex vivo by incubating OT-1 cells with OVA-expressing pancreas cells? Such ex vivo data could strongly the model that ABIN1 deficiency enhances cytotoxic T cell responses.

We did not observe a large reproducible effect on Listeria clearance. However, the Listeria clearance is very rapid in WT mice and it is well known that the contribution of even high numbers of transferred naïve antigen-specific T cells to Listeria clearance is usually not apparent (in contrast to the transfer of memory T cells).

We measured selected surface markers on the tumor-infiltrating cells, including PD-1, but we did not observe any clear differences. However, this could be caused by the relatively early time-point of this analysis. As the p38 activity has been linked to T-cell senescence and weak anti-tumor activity (PMID: 25151490, PMID: 32516591), it is very likely that the initial robust infiltration of ABIN1-deficient cells into the tumor is later compensated by their inability to maintain their stemness, which leads to practically no differences in the tumor clearance in the longer term. We discuss this point in the revised version of the manuscript. As we could not see a clear role of ABIN1 in the tumor clearance, we did not follow this further, e.g. to see whether these infiltrating cells get exhausted faster in later time points. We still believe that ABIN1 deficient T cells might be able to efficiently kill tumor cells in certain conditions (e.g., when preactivated), but we decided not to perform additional experiments in this project.

The diabetic model is very well established in our hands (PMID: 36705564, PMID: 36275704, PMID: 27188212). The glucose levels in the blood almost perfectly correlate with the glucose levels in the urine (Tsyklauri et al. PMID:6705564, Figure 1 – Figure Supplement 1A; also Fig. 6 below). For this reason, we measure the blood only once (typically on day 7) to reduce the stress of the animals. The reviewer is correct that the blood glucose would recapitulate the urine levels on day 8 and 9. However, we do not find any potential issue in this. The blood measurement was planned to be taken on day 7 post-immunization prior to the experiments, so there is no bias on our side.

In this assay, we have previously titrated the number of transferred OT-I cells. We know that 1000 OTI T-cells is the threshold number to induce diabetes in the WT settings. Thus, it is not surprising that the mice receiving 1000 WT OT-I T cells do develop diabetes, although it takes them longer than for those receiving 1000 GTKO OT-I T cells. It is also not surprising that 250 WT OT-I T cells are not sufficient. Thus, the most important result is that 250 GTKO OT-I T cells are able to induce diabetes in this setup. It is even more striking that they are able to do it slightly faster than 1000 WT OT-I T cells.

We decided to perform additional experiments in this model. We do not believe that ex vivo co-culture of OT-I T cells and the OVA-expressing pancreatic beta cells would give us much valuable data. Moreover, it would be an extremely technically difficult experiment, which is not established in our lab. However, we observed robust expansion and effector cell formation of OT-I T cells primed by OVA-loaded DCs, which is

the way of OT-I priming in the diabetic model (Fig. 6E) and also slightly increased expansion of OT-I_s in the spleen in the actual diabetic model on day 6 (Fig. EV6G). We also added H&E and IF stained histology of the pancreas showing enhanced lymphocyte infiltration, insulinitis, and pancreatic islets destruction in mice that had received *Abin1*^{GTKO/GTKO} OT-I cells (Fig. 6I, EV6H). We believe that these additional data make the manuscript more convincing.

Figure 6. Blood glucose (mmol/l) in RIP.OVA mice stratified according to urine glucose levels (mg/dl). n= 427 mice in 23 experiments done in our laboratory in different projects.

Minor points:

13. A20 is introduced as a deubiquitinase (page 2). Indeed, the function of A20 DUB activity remains obscure and recent work indicates that A20 binding to ubiquitin chains may be more relevant to counterbalance immune and inflammatory signaling.

We agree with the Reviewer. We adjusted the text.

14. CMB complex page 10 should be CBM complex.

We thank the Reviewer for spotting this. We corrected it.

15. Fig. 6F and G have been mixed up legend/figure. Diabetes-free Kaplan Meyer curve in Fig. 6F is not a survival curve as mentioned in legend.

We thank the Reviewer for spotting this. We corrected it.

Dear Dr. Stepanek,

Thank you for the submission of your revised manuscript to our editorial offices. I have already forwarded to you the reports I have received from the two referees that I asked to re-evaluate the study, you will find again below. As you know, referee #1 indicates that s/he is not yet satisfied with the revision and has remaining concerns, whereas referee #2 states that all questions and concerns have been adequately addressed. I now went through your rebuttal letter and feel that your responses adequately address the remaining concerns of referee #1. I thus ask you to address the remaining points of referee #1 (as indicated in your rebuttal letter) in a final revised manuscript. Please also provide a final p-b-p-response with your final revision, addressing the remaining referee points.

- Please reduce the number of keywords to 5 and provide the abstract written in present tense throughout. Please also order the manuscript sections like this, using these names:

Title page - Abstract - Keywords - Introduction - Results - Discussion - Methods - Data availability section - Acknowledgements - Disclosure and Competing Interests Statement - References - Figure legends - Expanded View Figure legends

- Please add the EV figure legends to the main manuscript text file.

- Please make sure that the number "n" for how many independent experiments were performed, their nature (biological versus technical replicates), the bars and error bars (e.g. SEM, SD) and the test used to calculate p-values is indicated in the respective figure legends (also for potential EV figures and all those in the final Appendix). Please also check that all the p-values are explained in the legend, and that these fit to those shown in the figure. Please provide statistical testing where applicable. Please avoid the phrase 'independent experiment', but clearly state if these were biological or technical replicates. Please also indicate (e.g. with n.s.) if testing was performed, but the differences are not significant. In case n=2, please show the data as separate datapoints without error bars and statistics. See also:

<http://www.embopress.org/page/journal/14693178/authorguide#statisticalanalysis>

If $n < 5$, please show single datapoints for diagrams. Presently, some diagrams seem to miss the 'n.s.'. Please check. Moreover:

- Please indicate the statistical test used for data analysis in the legends of figures 4c; 5h; EV 4c-d; EV 6a-d.

- Please note that information related to n is missing in the legend of figure 6e.

- Please note that the error bars are not defined in the legends of figures 5a-c; EV 4a; EV 6a-d, g.

- Please add to each legend (main, EV and Appendix figures, where applicable) a 'Data Information' section explaining the statistics used or providing information regarding replicates and scales. See:

- Please provide a completed the author checklist by adding responses to the pulldown menus (column D).

- Please add scale bars of similar style and thickness to all the microscopic images, using clearly visible black or white bars (depending on the background). Please place these in the lower right corner of the images themselves. Please do not write on or near the bars in the image but define the size in the respective figure legend. Presently, there are still some scale bars with text nearby.

- The additional supplementary material should be supplied as a single pdf file labelled Appendix. The Appendix should have page numbers and needs to include a table of content on the first page (with page numbers) and legends for all content. Please follow the nomenclature Appendix Figure Sx, Appendix Table Sx etc. throughout the text, and also label the figures and tables according to this nomenclature.

- The three Appendix tables uploaded are datasets. Please upload these as dataset files, best as excel files with a legend and a title on the first TAB. Please name these Dataset EVx and also change their callouts accordingly.

- Please make sure that all the funding information is also entered into the online submission system and that it is complete and similar to the one in the acknowledgement section of the manuscript text file. These grants are presently only mentioned in the acknowledgements section:

- the Czech Centre for Phenogenomics, OP RDI BIOCEV CZ.1.05/1.1.00/02.0109),

- core funding provided by the Institute of Molecular Genetics of the Czech Academy of Sciences (RVO 68378050)

- Czech Ministry of Education, Youth and Sports (LM2023050 and RVO - 68378050-KAV-NPUI)

- Thanks for providing the source data (SD). Could also SD for Fig. 1B be provided? Moreover, I would like to ask you to organise the SD differently. Please upload the SD as one folder per figure (ZIPed up together) including subfolders for each panel, as there are sometimes image files and numerical data files for the same panel.

In addition, I would need from you:

Referee #1:

This revised manuscript has incorporated a number of issues raised with the original manuscript. The revised manuscript does provide stronger data that ABIN1 is a negative regulator of CD8+ T cells. While the authors state that their primary aim is to characterize CD8+ T cells, there is quite a mixing of analysis of CD4+ and CD8+ T cells which makes the updated manuscript still difficult to follow and confusing- particularly as the role of ABIN1 appears more complex in CD4+ when compared to CD8+ T cells. If the authors believe that their focus is on CD8+ T cells as described in the "comments to reviewers", they should consider re-organizing the manuscript with this focus. Nonetheless, there remain issues with the revised manuscript that the authors should address.

Figure 2B. Analysis of GTKO mice is missing analysis of splenic T cells. This should also be shown so as to interpret the thymus data here as well as the subsequent bone marrow chimera spleen and signaling data (Figures 3A and D).

Figure 3D. The enhanced p38 signal observed in the GTKO has a significant caveat. Here, the authors appear to have used bulk T cells rather than purified naïve or activated T cells. Since there appear to be significantly more CD44hi CD4 T cells (Fig 2B), at least in lymph nodes, the difference in p38 signaling could be due to the different T cell subsets. Hence, additional experiments using purified T cell subsets is critical, especially when the p38 differences observed with OT-I T cells is minimal.

Figure 3E. There appears to be a discordance between the FACS data shown and the quantitation. The WT cell shown in the FACS plot quantifies 23% of cells being p38+, while there is no 23% data point in the graph.

Minor issues:

Figure 1D. The authors should provide an explanation of why there is such a discordant amount of FLAG-GITRL in lane 1 (FLAG blot) in unstimulated cells when compared to the stimulated cells in lanes 2-4.

Referee #2:

The authors have adequately addressed all my questions and concerns. It remains a very complex and intense manuscript combining many different in vivo model systems. Nevertheless, The study is of high value, because it represents the most comprehensive in vivo analyses of ABIN1 function in CD8 T cells.

Point-by-point response to Referees' comments

Referee #1:

This revised manuscript has incorporated a number of issues raised with the original manuscript. The revised manuscript does provide stronger data that ABIN1 is a negative regulator of CD8+ T cells. While the authors state that their primary aim is to characterize CD8+ T cells, there is quite a mixing of analysis of CD4+ and CD8+ T cells which makes the updated manuscript still difficult to follow and confusing- particularly as the role of ABIN1 appears more complex in CD4+ when compared to CD8+ T cells. If the authors believe that their focus is on CD8+ T cells as described in the "comments to reviewers", they should consider re-organizing the manuscript with this focus. Nonetheless, there remain issues with the revised manuscript that the authors should address.

The major focus of our work is on CD8⁺ T cells. We find the flow of the manuscript quite logical. We first identified ABIN1 as a component of the GITR/OX40 signaling complexes. Then we characterized the lymphocyte compartment of the polyclonal ABIN1-deficient mice. The analysis of Tregs was there mostly for completeness. And then we continued with the functional analysis of CD8+ T cells using the monoclonal model. Additional experiments with CD4+ T cells were added during the revision only to address the comments of all three reviewers. We are satisfied with how the manuscript reads at the moment. We offered to the handling editor that we were open to changes in the manuscript organization according to his eventual recommendations. Since we did not get any specific instructions from the editorial office along this line, we did not re-organize the manuscript.

Three out of four specific comments by this Referee in this second round of reviews are related to figure panels which were already included in the original version of the manuscript and were not changed during the revisions. However, these results were not addressed by this reviewer in the first round of review. As we are not sure whether this complies with the good practice in peer review, we highlight such comments in the text below.

1. Figure 2B. Analysis of GTKO mice is missing analysis of splenic T cells. This should also be shown so as to interpret the thymus data here as well as the subsequent bone marrow chimera spleen and signaling data (Figures 3A and D).

This concern should have been raised during the first round of review as this figure was not changed during the revision.

The phenotyping of splenic T cells of GTKO mice was shown in the Appendix 4 in both submitted versions of the manuscript, i.e., before and after the revisions. The reason why it is not shown in a main or EV Figure is a lack of space and the fact that the analysis of the spleen (Appendix Figure 4) and lymph nodes (Fig. 2B, Fig. EV2D) have very similar results leading to identical conclusions. Since the Appendix

Figure 4 is correctly referenced in the same sentence as the Fig. 2B, we do not see any potential issue here.

2. Figure 3D. The enhanced p38 signal observed in the GTKO has a significant caveat. Here, the authors appear to have used bulk T cells rather than purified naïve or activated T cells. Since there appear to be significantly more CD44^{hi} CD4 T cells (Fig 2B), at least in lymph nodes, the difference in p38 signaling could be due to the different T cell subsets. Hence, additional experiments using purified T cell subsets is critical, especially when the p38 differences observed with OT-I T cells is minimal.

This concern should have been raised during the first round of review as this figure was not changed during the revision.

We admit that using sorted naïve T cells might be a bit cleaner experiment. We performed this experiment in this way to analyze CD8 and CD4 T cells side by side in bulk. Anyway, we do not believe that using sorted T cells would change the results significantly as the T cells are anyway activated in vitro by PMA/ionomycin (to induce the expression of GITR), All these in vitro activated cells are then CD44⁺. As we showed in the previous Response to Reviewers' comments (Figure 2 there) the expression of GITR was comparable on both WT and GTKO cells after PMA/ionomycin activation. We also do not agree that the difference in p38 phosphorylation between WT and GTKO OT-I T cells is minimal. The GITRL-induced increase in % pp38⁺ T cells is approximately doubled in the absence of ABIN1 (Fig. 3E). Moreover, other experiments in the manuscript clearly point to the regulation of p38 phosphorylation by ABIN1 in T cells (Fig. 5A-B, D-H, Fig. EV5A-E). Although Fig. 5 shows TCR-induced activation and Fig. 3 shows GITR-induced activation, these pathways converge on p38 and it is very probable that ABIN1 regulates p38 in T cells independently of the actual primary upstream stimulus. Based on the limited expected benefit and relatively high costs (ethical, economic, time), we decided not to repeat this experiment in the new setup. However, we offered this option to the handling editor, who did not request this experiment.

3. Figure 3E. There appears to be a discordance between the FACS data shown and the quantitation. The WT cell shown in the FACS plot quantifies 23% of cells being p38⁺, while there is no 23% data point in the graph.

There is no discordance. The results of this experiment were quantified as the increase in the frequency of phospho-p38⁺ cells after activation (i.e., %pp38 in the activated sample - %pp38 in the non-activated sample). The "raw" data - the actual %pp38 in all samples (WT and GTKO, activated and non-activated) is shown in Fig. EV3E, including the value of 23% (or 22.7% when not rounded). We adjusted the legend to Fig. 3E to make it clear.

4. Minor issues:

Figure 1D. The authors should provide an explanation of why there is such a discordant amount of FLAG-GITRL in lane 1 (FLAG blot) in unstimulated cells when compared to the stimulated cells in lanes 2-4.

This concern should have been raised during the first round of review as this figure was not changed during the revision.

In the activated samples, we add the recombinant GITRL to the cells, which are then centrifuged and the pellet is resuspended in the lysis buffer. Only those GITRL molecules which actually bind to the cells are then present in the lysate/WB (the unbound GITRL is aspirated after centrifugation). In the control, the same amount of ligand is added to the lysate of non-activated cells (it is probably not necessary, but we believe it is better to do it to exclude potential direct interactions between the GITRL and the members of the GTR signaling complex, A20 and ABIN1, in the lysate). We revised the Figure legends to avoid any confusions.

Referee #2:

The authors have adequately addressed all my questions and concerns. It remains a very complex and intense manuscript combining many different in vivo model systems. Nevertheless, The study is of high value, because it represents the most comprehensive in vivo analyses of ABIN1 function in CD8 T cells.

We are happy and thankful that this reviewer and the third reviewer (who reviewed only the original version of the manuscript) appreciated our work.

Dr. Ondrej Stepanek
Institute of Molecular Genetics of the Czech Academy of Sciences
Adaptive Immunity
Videnska 1083
Prague 14220
Czech Republic

Dear Dr. Stepanek,

Thank you for the submission of your final revised manuscript to our editorial offices. I now went through the files and your final p-b-p-response and consider the remaining points of referee #1 as adequately addressed. I am thus very pleased to accept your manuscript for publication in the next available issue of EMBO reports. Thank you for your contribution to our journal.

Yours sincerely,
